# Asymptotic Behaviors of Projected Stochastic Approximation: A Jump Diffusion Perspective

**Jiadong Liang**
School of Mathematical Sciences
Peking University
jdliang@pku.edu.cn

**Yuze Han**
School of Mathematical Sciences
Peking University
hanyuze97@pku.edu.cn

**Xiang Li**
School of Mathematical Sciences
Peking University
lx10077@pku.edu.cn

**Zhihua Zhang**
School of Mathematical Sciences
Peking University
zhzhang@math.pku.edu.cn

## Abstract

In this paper we consider linearly constrained stochastic approximation problems with federated learning as a special case. We propose a loopless projection stochastic approximation algorithm (LPSA) to ensure feasibility by performing the projection with probability $p_n$ at the $n$-th iteration. Considering a specific family of the probability $p_n$ and step size $\eta_n$, we analyze our algorithm from an asymptotic and continuous perspective. Using a novel jump diffusion approximation, we show that the trajectories connecting those properly rescaled last iterates weakly converge to the solution of specific stochastic differential equations (SDEs). By analyzing SDEs, we identify the asymptotic behaviors of LPSA for different choices of $(p_n, \eta_n)$. We find the algorithm presents an intriguing asymptotic bias-variance trade-off according to the relative magnitude of $p_n$ w.r.t. $\eta_n$. It brings insights on how to choose appropriate $\{(p_n, \eta_n)\}_{n\geq 1}$ to minimize the projection complexity.

## 1 Introduction

Recently, a novel distributed computing paradigm that called *Federated Learning* (FL) has been proposed for collaboratively training a global model from data that remote *clients* hold [31]. As a standard optimization algorithm in FL, *Local SGD* alternates between running stochastic gradient descent (SGD) independently in parallel on different clients and averaging the sequences only once in a while. Put simply, it learns a shared global model via infrequent communication. Empirical investigation finds its superior performance in communication efficiency [30] and theoretical analysis toward it has already provided a complete picture [26, 1, 15, 42, 43, 15]. Among them, Li et al. [27] establishes a functional CLT that Local SGD with Polyak-Ruppert averaging simultaneously achieves the optimal asymptotic variance and diminishing average communication frequency. However, they are all derived from a discrete perspective.

The use of a continuous-time stochastic process to characterize the entire trajectory of a discrete stochastic algorithm has been witnessed progresses in recent years, and we call it diffusion approximation. The continuous approach has advantages in its rich toolbox and can provide intuitive explanation for uncanny phenomena that are intractable to analyze in discrete cases. It can also motivate new optimization algorithms and statistical inference methods. Current works applying diffusion approximation to stochastic optimization algorithms can be roughly divided into two classes. The first one is to interest the optimization algorithm as a numerical discretization of a specific stochastic differential equation (SDE) [14] in a finite time interval $[0, T]$. When the step size $\eta$ is

sufficiently small and the length $T(=n\eta)$ of the interval is fixed ($n$ is the total iterations), such approximation is of high accuracy, and it is easy to analyze the geometric properties of our target algorithms [41, 23, 13, 7, 38, 34, 8]. However, this avenue is difficult to capture the convergence behaviors around the optimal point due to the fixed $T$.[1] The second class comes up to solve the issue. It instead considers the iterates divided by a proper power function of step sizes. Under certain conditions, as $n$ goes to infinity, the rescaled iterates would weakly converge to the stationary solution of corresponding SDEs [21, 35, 6, 9, 10]. In FL, to the best of our knowledge, no work considers analyzing Local SGD via the aspect, which is our focus here.

However, it is not easy to serialize Local SGD iterates due to its double-loop nature. Recent researchers developed a new technique named as 'loopless' to simplify the two-loop structure for SVRG and Katyusha [16]. The key is to replace the hard loop with a probabilistic loop. Specifically, we will independently toss a (possibly biased) coin $\omega_n$ with head probability $p_n$ at iteration $n$. When getting the head $\omega_n = 1$, we start a new loop and update the outer-loop intermediate variables; when getting the tail $\omega_n = 0$, we stay in the same loop and keep the intermediate variables. In this way, we obtain a loopless counterpart algorithm and do not need to distinguish inner and outer loops anymore. It facilitates theoretical analysis and typically does not deteriorate the convergence rate [12, 29, 28, 11]. It is worth mentioning that Hanzely and Richtárik [12] first introduced the loopless technique to FL and obtained many efficient FL algorithms. Li [28] used a dynamic $p_n$ (which varies with $n$) to generalize the scope of original methods. We are then motivated to analyze a loopless version of Local SGD with decreasing $p_n$, but from an asymptotic and continuous perspective.

## 1.1 Contribution

Our work is motivated by Local SGD but beyond it. In particular, for a general optimization problems with linear constrains (of which FL is a special case), we develop a loopless projection stochastic approximation method (LPSA) as a generalization of Local SGD (see Appendix A for more details). Such generality renders us the possibility to transfer our techniques and results to other linearly constrained problems. LPSA is affected by two important hyperparameters, namely the step size $\{\eta_n\}$ and the projection probability $\{p_n\}$. For the choices of $\eta_n \propto n^{-\alpha}$ and $p_n \propto \min\{\eta_n^\beta, 1\}$, we derive a non-asymptotic convergence rate for different $\alpha \in (0, 1]$ and $\beta \in (0, 1)$ in Theorem 3.1. We observe a phase transition for the convergence rate $\mathcal{O}(n^{-\alpha \min\{1, 2-2\beta\}})$ when $\beta$ crosses 0.5.

To derive asymptotic results, we obtain two sequences $\{u_n\}$ and $\{v_n\}$ by orthogonal decomposition for the optimized sequence of LPSA. We then construct two sequences of processes which pass through the appropriately rescaled $u_n$ and $v_n$, respectively. We show rigorously, when the iteration goes to infinity, these two sequences of stochastic processes weakly converge to the solutions of specific SDEs that are driven by either a Brownian motion or a Poisson process. As a corollary, the rescaled last iterate of $u_n$ (which we mainly care about) has a known asymptotic distribution (either Gaussian distribution in Theorem 3.3 or Dirac in Corollary 1). And the phase transition we mentioned above evolves into a trade-off between the bias caused by the low frequency projection and the fluctuation resulting from the gradient noise (see Section 3.2.3).

Moreover, according to different convergence rate for every $\{(\eta_n, p_n)\}$ pair, we consider a selection scheme at the end of Section 3.2.3, which makes the algorithm have the same nonasymptotic convergence order as the conventional stochastic approximation and spend as little as possible on the projection operation which is usually expensive in practice. At the end, we conduct numerical experiments to confirm the theoretical results.

From a technical level, we propose a novel proof technique to analyze the discontinuity brought by probabilistic projection. In particular, we borrow tools from jump diffusion and verify necessary conditions (e.g., stochastic tightness) to apply it. See the paragraph after Theorem 3.4 for a main idea. We believe our technique can extend to and help analyze other stochastic approximation algorithms which can be approximated by a jump diffusion.

---

[1]A finite $T$ implies not only the algorithm but also its corresponding SDE do not converge to the optimum.

## 2 Problem Formulation

### 2.1 Loopless Projected Stochastic Approximation

Notice that distributed optimization such as FL can be formulated as a global consensus problem which is a linearly constrained problem [5]. For the sake of simplicity and generality, we aim to solve the following problem

$$\min_{\boldsymbol{x}} \mathbb{E}_{\zeta \sim \mathcal{D}} f(\boldsymbol{x}, \zeta) \text{ subject to } \boldsymbol{A}^\top \boldsymbol{x} = \boldsymbol{0} \tag{1}$$

via a randomly (and infrequently) projected stochastic approximation algorithm. In particular, at iteration $n$, we first perform one step of SGD via

$$\boldsymbol{x}_{n+\frac{1}{2}} = \boldsymbol{x}_n - \eta_n \nabla f(\boldsymbol{x}_n) + \eta_n \xi_n, \tag{2}$$

where $f(\boldsymbol{x}) = \mathbb{E}_{\zeta \sim \mathcal{D}} f(\boldsymbol{x}, \zeta)$ and $\xi_n = \nabla f(\boldsymbol{x}_n) - \nabla f(\boldsymbol{x}_n, \zeta_n)$. Here $\{\xi_n\}$ is a martingale difference sequence (m.d.s.) under the natural filtration $\mathcal{F}_{n+1} := \sigma(\zeta_k, \omega_k; k \leq n+1)$. We then use the loopless trick introduced in the introduction, i.e., we independently cast a coin with the head probability $p_n$ and obtain the result $\omega_n \sim \text{Bernoulli}(p_n)$. If $\omega_n = 1$, we perform one step of projection to ensure $\boldsymbol{x}_{n+1}$ fall into the feasible region: $\boldsymbol{x}_{n+1} = \mathcal{P}_{\boldsymbol{A}^\perp}(\boldsymbol{x}_{n+\frac{1}{2}})$ where $\mathcal{P}_{\boldsymbol{A}^\perp}$ denotes the projection onto the null space of $\boldsymbol{A}^\top$. If $\omega_n = 0$, we assign $\boldsymbol{x}_{n+1}$ as the same value of $\boldsymbol{x}_{n+\frac{1}{2}}$, i.e., $\boldsymbol{x}_{n+1} = \boldsymbol{x}_{n+\frac{1}{2}}$. It is clear this algorithm (2) mimics the behavior of Local SGD in FL settings (see Appendix A for the equivalence).

### 2.2 Assumptions

For the linearly constrained convex optimization problem (1), we make the following assumptions which are quite common in the literature. Without special clarification, $\|\cdot\|$ denotes the Euclidean norm for vectors and the spectral norm for matrices.

**Assumption 1** (Smoothness)**.** *We assume that $f\colon \mathbb{R}^d \to \mathbb{R}$ is $L$-smooth, that is,*

$$\|\nabla f(\boldsymbol{x}) - \nabla f(\boldsymbol{y})\| \leq L\|\boldsymbol{x} - \boldsymbol{y}\|, \quad \forall \boldsymbol{x}, \boldsymbol{y} \in \mathbb{R}^d.$$

**Assumption 2** (Strong convexity)**.** *We assume that $f\colon \mathbb{R}^d \to \mathbb{R}$ is $\mu$-strongly convex, that is,*

$$f(\boldsymbol{x}) - f(\boldsymbol{y}) \geq \langle \nabla f(\boldsymbol{y}), \boldsymbol{x} - \boldsymbol{y} \rangle + \frac{\mu}{2}\|\boldsymbol{x} - \boldsymbol{y}\|^2, \quad \forall \boldsymbol{x}, \boldsymbol{y} \in \mathbb{R}^d.$$

**Assumption 3** (Continuous Hessian matrix)**.** *We assume that $f\colon \mathbb{R}^d \to \mathbb{R}$ is Hessian Lipschitz, that is, there is a constant $\tilde{L}$ such that*

$$\left\|\nabla^2 f(\boldsymbol{x}) - \nabla^2 f(\boldsymbol{y})\right\| \leq \tilde{L}\|\boldsymbol{x} - \boldsymbol{y}\|, \quad \forall \boldsymbol{x}, \boldsymbol{y} \in \mathbb{R}^d.$$

**Assumption 4** (Continuous covariance matrix)**.** *Given an m.d.s. $\{\xi_t\}$, we denote the conditional covariance as $\mathbb{E}[\xi_t \xi_t^\top | \mathcal{F}_t] = \Sigma(\boldsymbol{x}_t)$ and assume it is $L$-Lipschitz continuous in the sense that*

$$\|\Sigma(\boldsymbol{x}) - \Sigma(\boldsymbol{y})\|_2 \leq L\|\boldsymbol{x} - \boldsymbol{y}\|, \quad \forall \boldsymbol{x}, \boldsymbol{y} \in \mathbb{R}^d.$$

**Assumption 5.** *For the m.d.s. $\{\xi_n\}$, we assume there exists a $p > 2$ such that the $p$-th moment of every element in $\{\xi_n\}$ is uniformly bounded, that is,*

$$\sup_{n \geq 0} \mathbb{E}\|\xi_n\|^p < \infty.$$

The first three assumptions imply we consider the strongly convex case. The last two assumptions help us identify the asymptotic variance. Especially, the assumption of uniformly bounded $p$ $(p > 2)$ moments is typically required to establish central limit theorems [9, 10, 27]. Finally, we want to emphasize that the stationary condition for the problem (1) is different from unconstrained ones; $\nabla f(\boldsymbol{x}^\star)$ is not necessarily zero, however, its projection into the null space of $\boldsymbol{A}^\top$ must be zero.

**Proposition 1** ([25], Corollary 2.1)**.** *Let $\mathcal{P}_{\boldsymbol{A}}$ be the projection onto the column space of $\boldsymbol{A}$ and $\mathcal{P}_{\boldsymbol{A}^\perp}$ the projection onto the null space of $\boldsymbol{A}^\top$. Under Assumption 2, the solution of (1) is unique (denoted $\boldsymbol{x}^\star$). Moreover, we have $\mathcal{P}_{\boldsymbol{A}^\perp}(\nabla f(\boldsymbol{x}^\star)) = \boldsymbol{0}$.*

## 2.3 Jump Diffusion

Jump diffusion is a stochastic Lévy process that involves jumps and diffusion. Typically, the former is modeled by a Poisson process, while the latter is modeled as a Brownian motion. It has wide and important applications in physics, finance[36], and computer vision.

We say a function $f$ defined on $\mathbb{R}$ is càdlàg when $f$ is right-continuous and has left limits everywhere. For a càdlàg process $(\mathbf{V}_s)_{s \geq 0}$, we denote $\mathbf{V}_{t-}$ as the left limit of $\mathbf{V}$. at time $t$. Let $\mathbf{N}_\gamma(t)$ denote the Poisson process with $\gamma$ the intensity, which quantifies the number of jumps up to the time $t$ and is clearly càdlàg. We use $\mathbf{N}_\gamma(dt) = \mathbf{N}_\gamma(t) - \mathbf{N}_\gamma(t-) \in \{0, 1\}$ to indicate whether $\mathbf{N}_\gamma$ jumps at time $t$ and $\int_0^T g(t-)\mathbf{N}(dt) = \sum_{\{t:\mathbf{N}_\gamma(t) \neq \mathbf{N}_\gamma(t-)\}} g(t-)$ to denote the integral that drives for a measurable function $g(\cdot)$. We will consider a special class of jump diffusion in the following form

$$d\mathbf{X}_t = \alpha(t, \mathbf{X}_t)dt + \beta(t, \mathbf{X}_t)d\mathbf{W}_t + \varphi(t, \mathbf{X}_{t-})\mathbf{N}_\gamma(dt). \tag{3}$$

When the coefficient functions $\alpha(t, \mathbf{X}_t)$ and $\beta(t, \mathbf{X}_t)$ satisfy conditions like linear growth and Lipschitz continuity, there exists a solution for the jump diffusion (3) (e.g., Theorem 1.19 in [33]).

# 3 Main Results

In the section, we are going to capture the convergence behaviors of our projected stochastic approximation method (2) from both non-asymptotic and asymptotic perspectives. We consider a specific family of step size $\eta_n$ and projection probability $p_n$, namely, $\eta_n = \eta_0 n^{-\alpha}$ and $p_n = \min\{\eta_n^\beta, 1\}$ indexed by $0 < \alpha \leq 1$ and $0 \leq \beta < 1$, respectively. The choice of step sizes $\eta_n$ has been used to establish CLTs [37, 27], while the choice of $p_n$ is quite novel. To provide a complete picture of convergence, we will consider almost all combinations of $\alpha$ and $\beta$.

## 3.1 Non-asymptotic Analysis

To provide the convergence rate, it is natural to focus on the projection of $\boldsymbol{x}_n$ into the column space of $\boldsymbol{A}$ (since it is the easiest feasible solution one can obtain from $\boldsymbol{x}_n$). Hence, we decompose the iterated $\boldsymbol{x}_n$ into two orthogonal components $\boldsymbol{x}_n := \boldsymbol{u}_n + \boldsymbol{v}_n$ where $\boldsymbol{u}_n = \mathcal{P}_{\boldsymbol{A}^\perp}(\boldsymbol{x}_n)$ and $\boldsymbol{v}_n = \mathcal{P}_{\boldsymbol{A}}(\boldsymbol{x}_n)$.[2] We specify the the convergence rate of $\mathbb{E}\|\boldsymbol{u}_n - \boldsymbol{x}^\star\|^2$ in terms of $\alpha, \beta$ and $n$ in the following theorem.

**Theorem 3.1.** *Suppose that Assumptions 1, 2 and 4 hold. Let $\eta_n = \eta_0 n^{-\alpha}$ and $p_n = \min\{\eta_n^\beta, 1\}$ with $0 \leq \beta < 1$. Then for (i) $0 < \alpha < 1$ or (ii) $\alpha = 1$ with $\eta_0 > 2/\mu$ ($\mu$ is the strong convexity parameter of the objective function $f$), we have*

$$\mathbb{E}\|\boldsymbol{u}_n - \boldsymbol{x}^\star\|^2 = \mathcal{O}(n^{-\alpha\min\{1, 2-2\beta\}}).$$

From Theorem 3.1, as $\beta$ decreases, that is, the projection happens more frequently, $\mathbb{E}\|\boldsymbol{u}_n - \boldsymbol{x}^\star\|^2$ converges faster. The rate is $\mathcal{O}(n^{-\alpha})$ when $\beta < 0.5$, while the rate is $\mathcal{O}(n^{-2\alpha(1-\beta)})$ when $\beta > 0.5$. Thus there exists a phase transition when $\beta$ goes across $0.5$, which implies we should analyze asymptotic performances for these two phases respectively. As an extreme, when $\beta = 1$, the algorithm is possible to disconverge in an artifact quadratic loss with a specific $\boldsymbol{A}$ (see Theorem 3.2). Though for a specific $\boldsymbol{A}$, it could apply to FL (see Corollary 2 in Appendix A.2.1 for the detail).

**Theorem 3.2.** *If $\eta_n = \eta_0 n^{-\alpha}$ and $p_n = \min\{p_0\eta_n, 1\}$ with $0 < \alpha \leq 1$, for a specific $\boldsymbol{A}$, there exists a quadratic function $f(\boldsymbol{x})$ so that $\nabla^2 f(\boldsymbol{x}) \succeq \mathbf{I}_d$ and $\mathbb{E}\|\boldsymbol{u}_n - \boldsymbol{x}^\star\|^2$ does not converge to 0. Here $\mathbf{I}_d \in \mathbb{R}^{d \times d}$ is the identity matrix, and $\nabla^2 f(\boldsymbol{x}) \succeq \mathbf{I}_d$ means $\nabla^2 f(\boldsymbol{x}) - \mathbf{I}_d$ is positive semidefinite.*

## 3.2 Asymptotic Behavior of the Rescaled Trajectory

In this section, we want to derive an asymptotic convergence for (2). Recall that there exists a phase transition for the convergence rate of $\mathbb{E}\|\boldsymbol{u}_t - \boldsymbol{x}^\star\|^2$ when $\beta$ crosses $0.5$, when the projection probability is set as $p_n = \eta_n^\beta$. In the following, we will analyze the asymptotic behaviors of LSPA for the two cases $\beta \in [0, 1/2)$ and $\beta \in (1/2, 1)$.

---

[2]One can check Proposition 4 in Appendix B to see why $\boldsymbol{u}_n$ is orthogonal to $\boldsymbol{v}_n$.

### 3.2.1 Case 1: Frequent Projection where $\beta \in [0, 1/2)$

From an asymptotic perspective, the typical central limit theorem (CLT) claims $\check{\boldsymbol{u}}_n := \frac{\boldsymbol{u}_n - \boldsymbol{x}^*}{\sqrt{\eta_{n-1}}}$ would weakly converge to a rescaled standard distribution [21]. It helps us capture the large-sample convergence behaviors and provide ways for future statistical inference. However, we can provide a stronger result that captures the asymptotic behavior of the whole trajectory. In particular, we serialize the sequence $\{\check{\boldsymbol{u}}_n\}$ by constructing a continuous random function (denoted $\bar{\boldsymbol{u}}_t^{(n)}$) such that it starts from $\bar{\boldsymbol{u}}_0^{(n)} = \check{\boldsymbol{u}}_n$, and as $t$ increases it will pass through $\check{\boldsymbol{u}}_{n+1}, \check{\boldsymbol{u}}_{n+2}$ and so on. We will show that such a random function $\bar{\boldsymbol{u}}_t^{(n)}$ will weakly converge to the solution of a specific SDE. From the SDE, we can derive asymptotic variance of $\check{\boldsymbol{u}}_n$ and the whole trajectory evolution.

Since $\bar{\boldsymbol{u}}_t^{(n)}$ should pass all $\{\check{\boldsymbol{u}}_k\}_{k \geq n}$, we can connect these discrete points by piecewise linear functions. To that end, we first derive the one-step relation between $\check{\boldsymbol{u}}_n$ and $\check{\boldsymbol{u}}_{n+1}$. In particular,

$$\check{\boldsymbol{u}}_{n+1} = \check{\boldsymbol{u}}_n - \eta_n \boldsymbol{b}_n + \sqrt{\eta_n} \xi_n^{(1)}, \tag{4}$$

$$\boldsymbol{b}_n := \mathcal{P}_{\boldsymbol{A}^\perp} \left( \nabla^2 f(\boldsymbol{x}^\star) - \frac{1}{2\eta_0} \mathbb{1}_{\{\alpha=1\}} \mathbf{I}_d \right) \check{\boldsymbol{u}}_n + \frac{1}{\eta_n} \mathcal{R}_n, \tag{5}$$

where $\mathcal{R}_n$ stands for a high-order residual error, and $\xi_n^{(1)}$ denotes the component of noise $\xi_n$ on the null space of $\boldsymbol{A}$. One can find the derivation of (4) in Appendix C.1. Roughly speaking, (4) can be viewed as a one-step Euler Maruyama discretization with timescale $\eta_n$ for an SDE, which starts at $\check{\boldsymbol{u}}_n$ with local drift coefficient $\boldsymbol{b}_n$ and local diffusion coefficient $\text{var}(\xi_n^{(1)})$.

**Definition 1** (**Time interpolation**). *Let a positive sequence $\gamma = \{\gamma_n\}_n^\infty$ decrease to zero. For $n \in \mathbb{N}$, $t \geq 0$, define*

$$N(n, t, \gamma) = \min_{m \in \mathbb{N}} \left\{ m \geq n : \sum_{k=n}^m \gamma_k > t \right\}, \Gamma_n(\gamma) = \sum_{k=1}^{n-1} \gamma_k, \text{ and } \underline{t}_n(\gamma) = \Gamma_{N(n,t,\gamma)} - \Gamma_n.$$

We introduce a time interpolation for the formal description of the continuous function and further analysis. Intuitively, $N(n, t, \gamma)$ is the number of iterations $m$ at which the sum of step sizes $\sum_{k=n+1}^{m+1} \eta_k$ is just larger than $t$ and $\underline{t}_n(\gamma)$ is the approximation of $t$ when we only use step sizes $\{\gamma_k\}_{k \geq n}$. Since $\gamma_n \to 0$, $\underline{t}_n(\gamma) \to t$ as $n$ goes to infinity. A property of Definition 1 is that $N(n, \Gamma_m(\gamma) - \Gamma_n(\gamma), \gamma) = m$ for any $m \geq n$. By now, we are ready to construct $\bar{\boldsymbol{u}}_t^{(n)}$. For a given $n \in \mathbb{N}$, let $\bar{\boldsymbol{u}}_0^{(n)} = \check{\boldsymbol{u}}_n$ and define for $t \geq 0$,

$$
\begin{aligned}
\bar{\boldsymbol{u}}_t^{(n)} = \check{\boldsymbol{u}}_n &+ \left\{ \sum_{k=n}^{N(n,t,\eta)-1} \eta_k \boldsymbol{b}_k + (t - \underline{t}_n(\eta)) \boldsymbol{b}_{N(n,t,\eta)} \right\} \\
&+ \left\{ \sum_{k=n}^{N(n,t,\eta)-1} \sqrt{\eta_k} \xi_k^{(1)} + \sqrt{t - \underline{t}_n(\eta)} \xi_{N(n,t,\eta)}^{(1)} \right\}.
\end{aligned}
\tag{6}
$$

From the construction, we can see that $\bar{\boldsymbol{u}}_{\underline{t}_k(\eta)}^{(n)} = \check{\boldsymbol{u}}_{n+k}$.

**Theorem 3.3** (**Diffusion Approximation**). *Let Assumptions 1-5 hold. The following family of continuous stochastic processes $\{\bar{\boldsymbol{u}}_t^{(n)} : t \geq 0\}_{n=1}^\infty$ weakly converges to the stationary weak solution of the following SDE:*

$$d\mathbf{X}_t = -\mathcal{P}_{\boldsymbol{A}^\perp} \left( \nabla^2 f(\boldsymbol{x}^\star) - \frac{1}{2\eta_0} \mathbb{1}_{\{\alpha=1\}} \mathbf{I}_d \right) \mathbf{X}_t dt + \mathcal{P}_{\boldsymbol{A}^\perp} \Sigma(\boldsymbol{x}^\star)^{\frac{1}{2}} d\mathbf{W}_t. \tag{7}$$

*Further, the rescaled sequence $\{\check{\boldsymbol{u}}_n\}_{n=1}^\infty$ converges weakly to the invariant distribution of the dynamics (7), i.e., $\mathcal{N}(\mathbf{0}, \tilde{\Sigma})$. Here the variance $\tilde{\Sigma}$ satisfies the Lyapunov equation*

$$\mathcal{P}_{\boldsymbol{A}^\perp} \left( \nabla^2 f(\boldsymbol{x}^\star) - \frac{1}{2\eta_0} \mathbb{1}_{\{\alpha=1\}} \mathbf{I}_d \right) \tilde{\Sigma} + \tilde{\Sigma} \left( \nabla^2 f(\boldsymbol{x}^\star) - \frac{1}{2\eta_0} \mathbb{1}_{\{\alpha=1\}} \mathbf{I}_d \right) \mathcal{P}_{\boldsymbol{A}^\perp} = \mathcal{P}_{\boldsymbol{A}^\perp} \Sigma(\boldsymbol{x}^\star) \mathcal{P}_{\boldsymbol{A}^\perp}.$$

**Remark 1.** *By using the continuous time version of the Lyapunov theorem (Lemma 1 in [40]), the Lyapunov equation has a unique positive semidefinite solution (denoted $\tilde{\Sigma}$). From Theorem 3.3 and Theorem 4.1.1 in [9], we can tell that when $\beta \in [0, \frac{1}{2})$ our algorithm LPSA achieves the same asymptotic variance as SGD that also uses the same step size. The typical projected SGD corresponds to the case $\beta = 0$, while LPSA allows $\beta$ to vary in $[0, \frac{1}{2})$. One can reduce the projection frequency by increasing $\beta$ (equivalently decreasing the probability $p_n$). Hence, when projection is expensive, LPSA is more efficient in performing projections due to its flexible and moderate projection frequency.*

**Proof Idea of Theorem 3.3.** We shed light on the proof idea of Theorem 3.3. From a high level, we leverage the general theory for operator semigroups, which are developed by Trotter and Kurtz [39, 17–19] and are used to analyze stochastic optimization algorithms in [9]. Our diffusion approximation results are built on it, but generalize it in the sense that we use the celebrated Prokhorov's theorem to extend to the whole trajectory. One difficulty is to prove the stochastic tightness of $\{\boldsymbol{u}_t^{(n)}\}$. To that end, we make use of a classic result (e.g. Theorem 7.3 of [4]).

### 3.2.2 Case 2: Occasional Projection where $\beta \in (1/2, 1)$

When we step into the low-frequency regime where $\beta \in \left(\frac{1}{2}, 1\right)$, the situation totally changes. Intuitively, when LPSA performs much less frequent projection, we will frequently use infeasible $\boldsymbol{x}_t$ to update parameters, which accumulates residual errors. These errors would not only dominate and slow down the non-asymptotic convergence rate (see Theorem 3.1), but also change the asymptotic behavior. In this case, we should not only find the right timescale, but also need to figure out how these errors are accumulated. To solve the issue, we develop a new analysis routine. In the following, we consider $p_t = \gamma \eta_t^\beta$ with $\gamma > 0$.

Our solution is to monitor another random process that is related with $\{\boldsymbol{v}_n\}$, which serves as a bridge to derive the asymptotic behavior of $\{\boldsymbol{u}_n\}$. The right scale should make the scaled sequence have non-vanishing expected $L_2$ norm. From Theorem 3.1, it should be $\check{\boldsymbol{v}}_n = \eta_{n-1}^{\beta-1}\boldsymbol{v}_n$. In addition, given $\check{\boldsymbol{v}}_n$, the candidate value of $\check{\boldsymbol{v}}_{n+1}$ before tossing the coin $\omega_n$, can be derived from LPSA's Algorithm 1 in Appendix A, and we denote this candidate as $\check{\boldsymbol{v}}_{(n+1)-}$.

$$\check{\boldsymbol{v}}_{(n+1)-} := \check{\boldsymbol{v}}_n - \eta_n^\beta \boldsymbol{d}_n + \eta_n^\beta \xi_n^{(2)}, \tag{8}$$

where $\boldsymbol{d}_n = \nabla f(\boldsymbol{x}^*) + \eta_n^{-\beta}\mathcal{S}_n$ with $\mathcal{S}_n$ a residual error which satisfies $\eta_n^{-\beta}\mathcal{S}_n = o_{\mathbb{P}}(1)$ (see Appendix C.2 for more details) and $\xi_n^{(2)}$ stands for the component of noise $\xi_n$ on the orthogonal complementary space $A^\perp$. Due to the probabilistic projection, $\check{\boldsymbol{v}}_{n+1}$ takes value $\check{\boldsymbol{v}}_{(n+1)-}$ with probability $1 - \gamma\eta_n^\beta$ and takes value zero with probability $\gamma\eta_n^\beta$. Similar to the previous section, we then construct a càdlàg random process $\bar{\boldsymbol{v}}_t^{(n)}$ which starts from $\check{\boldsymbol{v}}_n$ and will pass through $\{\check{\boldsymbol{v}}_{(k)-}\}_{k \geq n}$. We can connect these discrete points with a step function. It results in the following construction

$$\bar{\boldsymbol{v}}_t^{(n)} = \bar{\boldsymbol{v}}_{\underline{t}_n(\eta^\beta)}^{(n)} - \left(t - \underline{t}_n(\eta^\beta)\right)\left(\boldsymbol{d}_{N(n,t,\eta^\beta)} - \xi_{N(n,t,\eta^\beta)}^{(2)}\right) \text{ if } t \in \left(\underline{t}_n(\eta^\beta), \underline{t}_n(\eta^\beta) + \eta_{\underline{t}_n(\eta^\beta)}^\beta\right),$$
$$\bar{\boldsymbol{v}}_{\underline{t}_n(\eta^\beta)}^{(n)} = \check{\boldsymbol{v}}_{N(n,t,\eta^\beta)}. \tag{9}$$

From (9), we can claim that $\bar{\boldsymbol{v}}_{\underline{t}_n(\eta^\beta)-}^{(n)} = \check{\boldsymbol{v}}_{N(n,t,\eta^\beta)-}$ for any $t \geq 0$. With probability $p_{N(n,t,\eta^\beta)}$, $\check{\boldsymbol{v}}_{N(n,t,\eta^\beta)}$ takes value zero, which causes the process $\bar{\boldsymbol{v}}_\cdot^{(n)}$ to change abruptly at the time $\underline{t}_n(\eta^\beta)$. These discontinuities about $\bar{\boldsymbol{v}}_\cdot^{(n)}$ prevent the diffusion process from working on $\bar{\boldsymbol{v}}_\cdot^{(n)}$ as the result of Theorem 3.3. Even so, the following theorem shows that we can still find a suitable process in the broader *jump* diffusion class to approximate $\bar{\boldsymbol{v}}_\cdot^{(n)}$.

**Theorem 3.4 (Jump Approximation).** *Let Assumptions 1, 2, 4 and 5 hold. The following family of càdlàg stochastic processes $\{\bar{\boldsymbol{v}}_t^{(n)} : t \geq 0\}_{n=1}^\infty$ weakly converges to the stationary weak solution of the following SDE*

$$d\mathbf{Y}_t = -\nabla f(\boldsymbol{x}^\star)dt - \mathbf{Y}_{t-} \cdot \mathbf{N}_\gamma(dt). \tag{10}$$

*Here $\mathbf{N}_\gamma(t)$ represents Poisson process with intensity $\gamma$, and $\mathbf{N}_\gamma(dt) = \mathbf{N}_\gamma(t) - \mathbf{N}_\gamma(t-)$. Further, the rescaled sequence $\{\check{\boldsymbol{v}}_n\}_{n=1}^\infty$ weakly converges to the invariant distribution of the dynamics (10), i.e., $-\frac{\nabla f(\boldsymbol{x}^\star)}{\|\nabla f(\boldsymbol{x}^\star)\|} \cdot \mathcal{E}\left(\frac{\|\nabla f(\boldsymbol{x}^\star)\|}{\gamma}\right)$. Here $\mathcal{E}(\theta)$ represents the exponential distribution with intensity $\frac{1}{\theta}$.*

Theorem 3.4 shows that the sequence $\{\bar{\boldsymbol{v}}_t^{(n)}\}$ constructed by shifting initial points will finally approximate a jump process with a constant drift as $n$ goes to infinity. The SDE (10) sheds light on how $\bar{\boldsymbol{v}}_t^{(n)}$ (equivalently a rescaled version of $\boldsymbol{v}_n$) move as $t$ increases. As the error incurred by infrequent projections, $\bar{\boldsymbol{v}}_t^{(n)}$ will move towards the direction of $\nabla f(\boldsymbol{x}^*)$ (due to the drift term $-\nabla f(\boldsymbol{x}^*)dt$) and be periodically forced to set as zero vector ( due to the correcting term $-\mathbf{Y}_{t-} \cdot \mathbf{N}_\gamma(dt)$ ). From a qualitative perspective, the SDE (10) captures the periodical behavior of $\boldsymbol{v}_n$, hence it shows without projection the residual error will accumulate along the direction of $\nabla f(\boldsymbol{x}^\star)$. As argued in Proposition 1, $\nabla f(\boldsymbol{x}^\star)$ is unlikely to be zero in our constrained problems.

The remaining issue is how to link $\{\bar{\boldsymbol{v}}_t^{(n)}\}$ to our target $\{\boldsymbol{u}_n\}$. Similarly, we should consider a rescaled $\boldsymbol{u}_n$, that is, $\hat{\boldsymbol{u}}_n := (\boldsymbol{u}_n - \boldsymbol{x}^\star)/\eta_{n-1}^{1-\beta}$. The following corollary, which is based on Theorem 3.4, shows when $\beta \in \left(\frac{1}{2}, 1\right)$, $\hat{\boldsymbol{u}}_n$ converges to a non-zero vector. Recall that $\breve{\boldsymbol{u}}_n = \frac{\boldsymbol{u}_n - \boldsymbol{x}^*}{\sqrt{\eta_{n-1}}} = \eta_{n-1}^{0.5-\beta} \hat{\boldsymbol{u}}_n$. The equation together with Corollary 1 implies $\|\mathbb{E}\breve{\boldsymbol{u}}_n\| = \eta_{n-1}^{0.5-\beta} \|\mathbb{E}\hat{\boldsymbol{u}}_n\| \to \infty$. As a result, the bias in Corollary 1 instead of the Gaussian fluctuation in Theorem 3.3 becomes the leading term hindering the convergence.

**Corollary 1.** *Let Assumptions 1- 3 hold. Then* $\hat{\boldsymbol{u}}_n := \frac{1}{\eta_{n-1}^{1-\beta}}(\boldsymbol{u}_n - \boldsymbol{x}^\star)$ *converges to a non-zero vector* $\frac{1}{\gamma} \left\{ \mathcal{P}_{\boldsymbol{A}^\perp} \left( \nabla^2 f(\boldsymbol{x}^\star) - \frac{1-\beta}{\eta_0} \mathbb{1}_{\{\alpha=1\}} \mathbf{I} \right) \mathcal{P}_{\boldsymbol{A}^\perp} \right\}^\dagger \left( \mathcal{P}_{\boldsymbol{A}^\perp} \nabla^2 f(\boldsymbol{x}^\star) \nabla f(\boldsymbol{x}^\star) \right)$ *in the $L_2$ as $n \to \infty$.* *Where $\mathbf{G}^\dagger$ denotes the pseudoinverse of the symmetric matrix $\mathbf{G}$.*

**Proof Idea of Theorem 3.4**   The main proof idea is similar to that of Theorem 3.4 except that we need to handle the jump diffusion which introduces additional discontinuity. As a result, for each $n \geq 0$, $\{\bar{\boldsymbol{v}}_t^{(n)}\}_{t \geq n}$ is càdlàg rather than continuous. We then use the approximation result for jump diffusions developed by Kushner [20] instead of Trotter and Kurtz's theories. Furthermore, the tool for proving tightness also needs to change. We replace the classic tool in [4] with a generalized determination method, the latter used to establish the stochastic tightness for càdlàg processes (e.g., Theorem 4.1 in [19]). The remaining issue is to figure out properties (e.g., the mixing nature) of (10). To that end, we establish the geometric ergodicity of Eq. (10) by combining the coupling method with the Itô's formula for jump diffusions, and show that its invariant distribution exists uniquely.

### 3.2.3   Summary and Discussion

From Sections 3.2.1 and 3.2.2, for the choice $p_n \propto \eta_n^\beta$, when $\beta$ varies, our algorithm has an interesting bias-variance tradeoff. In fact, Theorems 3.3 and 3.4 reveal that the fluctuation of $\boldsymbol{u}_n$ is of order $\mathcal{O}(\eta_n^{\frac{1}{2}})$ and the bias is of order $\mathcal{O}(\eta_n^{1-\beta})$. When $\beta \in [0, 1/2)$ the fluctuation caused by the randomness of gradient queries in every iteration dominates the optimization accuracy. And when $\beta \in (1/2, 1)$, this indicator is manipulated by the biases formed by the accumulation of skewed updates in the unconstrained state within each 'inner loop'.

In practice, projection is expensive to perform. Hence, it is important to tune $\alpha$ and $\beta$ so that the projection complexity is minimized as much as possible. We use the average projection complexity (APC) to quantify the projection efficiency. For a target accuracy $\epsilon > 0$, APC is defined as the number of projections required to obtain an $\epsilon$-accuracy feasible solution. We summarize the derived results and the corresponding APC in Table 1. We can see that APC is minimized when $\alpha \to 1$ and $\beta \to 0.5$. In this case, APC is approaching $\frac{1}{\sqrt{\epsilon}}$.

We find an interesting parallelism between LPSA and Local SGD. In the case of FL, projection complexity corresponds to communication complexity, because a synchronization in FL is essentially a projection in linearly constrained problems (see Appendix A for the equivalence). In [27], the authors analyzed the averaged communication complexity (ACC) for Local SGD with Polyak-Ruppert averaging.[3] They considered a general case where the length of the $m$-th inner loop could be up to $E_m := m^\nu$ with $\nu \in [0, 1)$. After $E_m$ steps of inner loop, communication would perform to synchronize local models. Hence, $E_m$ plays a role similar to $p_n$ in our paper. Li et al. [27] found that when $\nu \in [0, 1)$, the averaged Local SGD iterates enjoy an optimal asymptotic normality up

---

[3]For a target accuracy $\epsilon > 0$, ACC is defined as the number of communication required to obtain a $\epsilon$-accuracy global parameter.

Table 1: (Non-)Asymptotic results and projection complexity under different choice of $\eta_n$ and $p_n$. The first two columns list the non-asymptotic and asymptotic results respectively, and the last column characterizes projection complexity.

| $(\alpha, \beta)$ | $\mathbb{E}\|\boldsymbol{u}_n - \boldsymbol{x}^\star\|^2$ | Asymptotic behavior | APC |
|---|---|---|---|
| $(0,1] \times [0,1/2)$ | $\mathcal{O}\left(\frac{1}{n^\alpha}\right)$ 3.1 | Normal 3.3 | $\mathcal{O}\left(\epsilon^{\beta - \frac{1}{\alpha}}\right)$ |
| $(0,1] \times (1/2,1)$ | $\mathcal{O}\left(\frac{1}{n^{2\alpha(1-\beta)}}\right)$ 3.1 | Biased 1 | $\mathcal{O}\left(\epsilon^{\frac{\alpha\beta-1}{2\alpha(1-\beta)}}\right)$ |

to a known constant scale and its ACC is $\left(\frac{1}{\epsilon}\right)^{\frac{1}{1+\nu}}$. When $\nu \to 1$, ACC is approaching $\frac{1}{\sqrt{\epsilon}}$, similar to our case where APC converges to $\frac{1}{\sqrt{\epsilon}}$ when $\alpha \to 1$ and $\beta \to 0.5$. Actually, the $\frac{1}{\sqrt{\epsilon}}$ average communication complexity is actually optimal for any first-order oracle distributed algorithms, as shown in [43]. Hence, it implies our LSPA is efficient and near optimal in projection, because we can always reduce FL as a special of (1).

## 4 Experiments

In this section, we validate our theoretical results through comprehensive experiments. Due to space limitations, we only show some representative results on synthetic datasets under FL settings. For the results on general linearly constrained problems, please refer to Appendix D.

**Experimental Setup** We focus on classification problems with cross entropy loss, and $\ell_2^2$ regularization is imposed to ensure the strong convexity of the objective function. The synthetic datasets are generated by following [24]. There are $K$ clients and the sample $(\boldsymbol{x}_k, z_k)$ on the $k$-th client is modeled as $\boldsymbol{x}_k \sim \mathcal{N}(\nu_k, \Lambda)$ and $z_k = \operatorname{argmax}(\operatorname{softmax}(\boldsymbol{W}_k \boldsymbol{x}_k + \boldsymbol{b}_k))$ where $\Lambda \in \mathbb{R}^{d \times d}$ is diagonal with the entry $(j, j)$ equal to $j^{-1.2}$, $\boldsymbol{W}_k \in \mathbb{R}^{C \times d}$ and $\boldsymbol{b}_k \in \mathbb{R}^C$. We consider two specific datasets. The first one is denoted by `IID`, where all the clients share the same $\boldsymbol{W}_k$ and $\boldsymbol{b}_k$, and $\nu_k \sim \mathcal{N}(\mathbf{0}, \mathbf{I}_d)$. For this one, we set $K = 100$, $d = 60$ and $C = 10$. The second one is denoted by `Synthetic` $(a, b)$, where $a$ and $b$ control the heterogeneity across clients. Specifically, each entry of $\boldsymbol{W}_k$ and $\boldsymbol{b}_k$ is modeled as $\mathcal{N}(\mu_k, 1)$ with $\mu_k \sim \mathcal{N}(0, a)$ and $\nu_k \sim \mathcal{N}(\zeta_k, \boldsymbol{I})$ with $\zeta_k \sim \mathcal{N}(\mathbf{0}, b\mathbf{I}_d)$. For this dataset, we set $K = 20$, $d = 10$ and $C = 5$.

We find that the results on `IID` are intuitive enough to demonstrate the convergence rates of the mean squared error (MSE) $\mathbb{E}\|\boldsymbol{u}_n - \boldsymbol{x}^\star\|^2$ and the asymptotic behavior of $\check{\boldsymbol{u}}_n$ for $\beta \in [0, 1/2)$. The results on `Synthetic` $(a, b)$, a dataset with fewer parameters and more heterogeneity, are more appropriate to illustrate the asymptotic biased of $\hat{\boldsymbol{u}}_n$ for $\beta \in (1/2, 1)$. The full results on both the datasets are deferred to Appendix D.

**Convergence Rate** We plot the log-log scale graphs of averaged MSEs over 5 repetitions on `IID` vs iterations in Figure 1. The value of $\alpha$ is set as $\{1, 0.8, 0.6\}$ and the value of $\beta$ is from $\{0, 0.2, 0.4, 0.6, 0.8\}$. For each repetition, we run 2000 steps of LPSA. By Theorem 3.1, the slope of the line in the log-log scale graph should be $-\alpha \min\{1, 2 - 2\beta\}$. This is in accordance with Figure 1 when the iteration is larger than 100. For $\beta \in [0, 1/2)$, the value of $\beta$ does not affect the slope, while for $\beta \in (1/2, 1)$, larger $\beta$ and smaller $\alpha$ both lead to smoother lines.

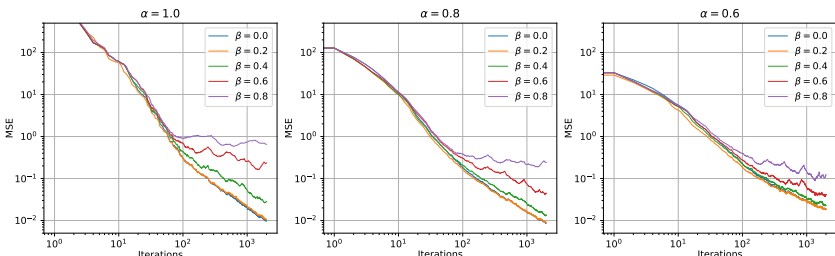

Figure 1: The log-log scale graphs of averaged MSE on `IID` over 5 repetition vs iterations.

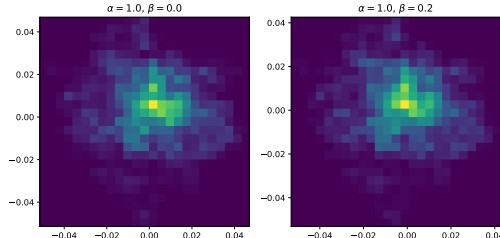
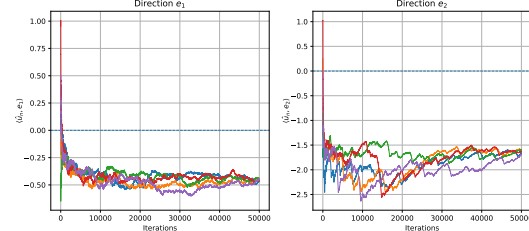

Figure 2: The heatmaps of $\check{u}_n$ across two orthogonal directions over 100 repetitions on `IID`.

Figure 3: Trajectories of $\hat{u}_n$ along two random directions over 5 repetitions on `Synthetic` $(1, 1)$.

**Frequent Projection**   For $\alpha = 1$ and $\beta \in \{0, 0.2\}$, we run 2000 steps of LPSA over 100 repetitions on `IID` and pick up the last 200 iterates. For these iterates, we compute the rescaled vectors $\check{u}_n$ and project them into a two-dimensional random subspace. Then we plot the heatmaps across the two dimensions in Figure 2. We observe that the cells near the origin have the lightest colors, and as we move away from the origin the cell color becomes darker. Since the cells with lighter colors imply more frequencies, these phenomenons agree with Theorem 3.3, where the limiting distribution of $\check{u}_n$ is Gaussian. The results with other values of $\alpha$ and $\beta$ are deferred to Appendix D.4.

**Occasional Projection**   For $\alpha = 0.8$ and $\beta = 0.6$, we run 50000 steps of LPSA over 5 repetition on `Synthetic` $(1, 1)$. Then we compute the rescaled sequence $\hat{u}_n$ and project them along two random directions $e_1$ and $e_2$. The trajectories depicted in Figure 3 show that the limits of $\langle \hat{u}_n, e_1 \rangle$ and $\langle \hat{u}_n, e_2 \rangle$ are nonzero and verify the asymptotic biased of $\hat{u}_n$ mentioned in Corollary 1. The results with other values of $\alpha$ and $\beta$ are deferred to Appendix D.6.

## 5   Concluding Remarks

In this paper we study the linearly constrained optimization problem. We propose the LPSA algorithm that is inspired by Local SGD. The probabilistic projection in LPSA follows the spirit of loopless methods [16, 12, 28] and simplifies the double-loop structure of original Local SGD, facilitating theoretical analysis. We thoroughly analyze the (non-)asymptotic properties of properly scaled trajectories obtained from $\{u_n\}$ and discover an interesting phase transition where $\{u_n\}$ changes from asymptotically normal to asymptotically biased as the projection frequency decreases. From a technical level, we generalize jump diffusion approximations to accommodate the particularity and discontinuity of LPSA.

There are also some open problems. It is unclear about the asymptotic behavior of $u_n$ when $\beta = 0.5$, i.e., $p_n = \Theta(\sqrt{\eta}_n)$. The jump diffusion approach fails because we can't analyze $\{u_n\}$ via the length of $\{v_n\}$ anymore. It accounts for failure that $\{\check{u}_n\}$ and $\{\check{v}_n\}$ are incompatible in the sense that they use different time scales and the time interpolation. However, we speculate $\check{u}_n$ would finally converge weakly to a non-centred Gaussian distribution. In addition, it is also interesting to analyze the performance of projection complexity of LPSA. From Corollary 1, to achieve a better convergence rate at lower projection frequencies, we must overcome the asymptotically biased nature of $u_n$. One feasible approach is to build a 'de-biasing' algorithm which attenuates the effect of $v_n$ during the update of $u_n$. We leave them as future work.

## Acknowledgments and Disclosure of Funding

This work has been supported by the National Natural Science Foundation of China (No. 12271011).

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
