## A  Special Condition: Federated Learning

In this section, we focus on the specific case of Federated Learning (FL). We first present the FL problem and establish the equivalence between LPSA and local SGD. Then we restate our main results under the context of FL. We also discuss related works on distributed optimization.

### A.1  The Problem and Reduction

In this subsection, we formulate our algorithm in federated settings and show that it is equivalent to local SGD. Before we proceed, we first give the formal statement of our algorithm

---

**Algorithm 1:** Loopless Projected Stochastic Approximation (LPSA)

> **Input:** function $f$, data distribution $\mathcal{D}$, initial point $\boldsymbol{x}_0$, step size $\eta_n$, projection probability $p_n$.
> **Initialization:** let $\mathbf{x}_0^{(k)} = \mathbf{x}_0$ for all $k$.
> **for** $n = 0$ **to** $T - 1$ **do**
>   Sample $\zeta_n \sim \mathcal{D}$ and $\omega_n \sim \mathrm{Bernoulli}(p_n)$
>   $\boldsymbol{x}_{n+\frac{1}{2}} = \boldsymbol{x}_n - \eta_n \nabla f(\boldsymbol{x}_n, \zeta_n)$
>   **if** $\omega_n = 1$ **then**
>     $\boldsymbol{x}_{n+1} = \mathcal{P}_{\boldsymbol{A}^\perp} \boldsymbol{x}_{n+\frac{1}{2}}$
>   **else**
>     $\boldsymbol{x}_{n+1} = \boldsymbol{x}_{n+\frac{1}{2}}$
>   **end if**
> **end for**
> **Return:** $\mathcal{P}_{\boldsymbol{A}^\perp} \boldsymbol{x}_T$.

---

For typical distributed optimization problems, we can rewrite them as a global consensus problem,

$$\min_{\boldsymbol{x}^{(1)}, \boldsymbol{x}^{(2)}, \cdots, \boldsymbol{x}^{(N)}} \frac{1}{N} \sum_{k=1}^{N} \mathbb{E}_{\zeta^{(k)} \sim \mathcal{D}_k} g(\boldsymbol{x}^{(k)}, \zeta^{(k)}) \quad \text{s.t. } \boldsymbol{x}^{(1)} = \cdots = \boldsymbol{x}^{(N)}, \tag{11}$$

where there are $N$ clients, $\boldsymbol{x}^{(k)}$ is the local parameter at the $k$-th client and $\zeta^{(k)}$ represents the randomness from this client. If we concatenate all the local parameters as $\boldsymbol{x} = \left[(\boldsymbol{x}^{(1)})^\top, (\boldsymbol{x}^{(2)})^\top, \cdots, (\boldsymbol{x}^{(N)})^\top\right]^\top \in \mathbb{R}^{Nd}$ and $\zeta = (\zeta^{(1)}, \zeta^{(2)}, \cdots, \zeta^{(N)})^\top$, we can rewrite the equation (11) as the form of equation (1), where $f(\boldsymbol{x}, \zeta) = \frac{1}{N} \sum_{k=1}^{N} g(\boldsymbol{x}^{(k)}, \zeta^{(k)})$, $\mathcal{D} = \mathcal{D}_1 \times \mathcal{D}_2 \times \cdots \times \mathcal{D}_N$ and $\boldsymbol{A}^\top$ is equipped with a particular structure

$$\boldsymbol{A}^\top = \begin{bmatrix} \mathbf{I}_d & -\mathbf{I}_d & \mathbf{0}_d & \cdots & \mathbf{0}_d & \mathbf{0}_d \\ \mathbf{0}_d & \mathbf{I}_d & -\mathbf{I}_d & \cdots & \mathbf{0}_d & \mathbf{0}_d \\ \vdots & \vdots & \vdots & \ddots & \vdots & \vdots \\ \mathbf{0}_d & \mathbf{0}_d & \mathbf{0}_d & \cdots & \mathbf{I}_d & -\mathbf{I}_d \end{bmatrix} \in \mathbb{R}^{(N-1)d \times Nd}. \tag{12}$$

In the expression of $\boldsymbol{A}^\top$, $\mathbf{I}_d \in \mathbb{R}^{d \times d}$ is the identity matrix and $\mathbf{0}_d \in \mathbb{R}^{d \times d}$ is the zero matrix. For such an $\boldsymbol{A}$, the operators $\mathcal{P}_{\boldsymbol{A}^\perp}$ and $\mathcal{P}_{\boldsymbol{A}}$ are easy to compute. One can check that

$$\mathcal{P}_{\boldsymbol{A}^\perp}(\boldsymbol{x}) = \left[\bar{\boldsymbol{x}}^\top, \bar{\boldsymbol{x}}^\top, \cdots, \bar{\boldsymbol{x}}^\top\right]^\top \tag{13}$$

and

$$\mathcal{P}_{\boldsymbol{A}}(\boldsymbol{x}) = \left[(\boldsymbol{x}^{(1)} - \bar{\boldsymbol{x}})^\top, (\boldsymbol{x}^{(2)} - \bar{\boldsymbol{x}})^\top, \cdots, (\boldsymbol{x}^{(N)} - \bar{\boldsymbol{x}})^\top\right]^\top,$$

where $\bar{\boldsymbol{x}} = \frac{1}{N} \sum_{i=1}^{N} \boldsymbol{x}^{(i)}$. Then we can establish the equivalent between LPSA and local SGD. At iteration $n$, the step $\boldsymbol{x}_{n+\frac{1}{2}} = \boldsymbol{x}_n - \eta_n \nabla f(\boldsymbol{x}_n, \zeta_n)$ represents a step of local update, that is $\boldsymbol{x}_{n+\frac{1}{2}}^{(k)} = \boldsymbol{x}_n^{(k)} - \eta_n \nabla g(\boldsymbol{x}_n^{(k)}, \zeta_n^{(k)})$ for each $k$. If $\omega_n = 1$, the projection step $\boldsymbol{x}_{n+1} = \mathcal{P}_{\boldsymbol{A}^\perp}(\boldsymbol{x}_{n+\frac{1}{2}})$ becomes a round of communication such that all the local parameters share the same value, i.e.,

$x_{n+1}^{(k)} = \frac{1}{N}\sum_{i=1}^{N} x_{n+\frac{1}{2}}^{(i)}$ for each $k$; if $\omega = 0$, no communication happens and $x_{n+1}^{(k)} = x_{n+\frac{1}{2}}^{(k)}$.

Finally, Algorithm 1 returns the average of all local parameters $\mathcal{P}_{A^\perp} x_T = \frac{1}{N}\sum_{k=1}^{N} x_T^{(k)}$.

The above reduction analysis implies that under the context of FL, Algorithm 1 becomes a loopless version of local SGD. The main difference between LPSA and original Local SGD is that LPSA has a stochastic length of local updates, which is determined by how frequent we observe $\omega_n = 1$. Since $p_n$ gradually decreases, the expectation of local updates would gradually increase. Such a difference does not deteriorate the convergence under certain conditions, as shown in Theorem 3.1. Moreover, the probabilistic loop also facilitates theoretical analysis.

## A.2 Restatement of Theoretical Results

In this subsection, we examine the theoretical results and give a revision of Theorem 3.2 under the FL condition.

For simplicity, we define the population loss function on client $k$ as $g_k(x^{(k)}) := \mathbb{E}_{\zeta^{(k)} \sim \mathcal{D}_k} g(x^{(k)}, \zeta^{(k)})$. Then we have $f(x) = \mathbb{E}_{\zeta \sim \mathcal{D}} f(x, \zeta) = \frac{1}{N}\sum_{k=1}^{N} g_k(x^{(k)})$,

$$\nabla f(x) = \frac{1}{N}\begin{bmatrix}\nabla g_1(x^{(1)}) \\ \nabla g_2(x^{(2)}) \\ \vdots \\ \nabla g_n(x^{(N)})\end{bmatrix}$$

and

$$\nabla^2 f(x) = \frac{1}{N}\begin{bmatrix}\nabla^2 g_1(x^{(1)}) & \mathbf{0}_d & \cdots & \mathbf{0}_d \\ \mathbf{0}_d & \nabla^2 g_2(x^{(2)}) & \cdots & \mathbf{0}_d \\ \vdots & \vdots & \ddots & \vdots \\ \mathbf{0}_d & \mathbf{0}_d & \cdots & \nabla^2 g_N(x^{(N)})\end{bmatrix}.$$

We first focus on Proposition 1. Note that the solution to (11) must be of the form

$$x^\star = \left[(x^{(\star)})^\top, (x^{(\star)})^\top, \cdots, (x^{(\star)})^\top\right]^\top \in \mathbb{R}^{Nd}.$$

Proposition 1 implies that the solution satisfies $\frac{1}{N}\sum_{k=1}^{N}\nabla g_k(x^{(\star)}) = \mathbf{0}$. This equation does not imply that the $\nabla g_k(x^\star)$ are all equal to zero. In fact, under the heterogeneous setting, where the $g_k$ are different due to the diversity across the clients, we typically have $\nabla f(x^{(\star)}) \neq \mathbf{0}$. This is crucial for the validity of Theorem 3.4 and Corollary 1. As for the homogeneous setting where the $g_k$ share the same form, $\frac{1}{N}\sum_{k=1}^{N}\nabla g_k(x^{(\star)}) = \mathbf{0}$ does imply $\nabla g_k(x^{(\star)}) = \mathbf{0}$ and consequently $\nabla f(x^\star) = \mathbf{0}$. In this case, Theorem 3.4 and Corollary 1 do not hold any more. However, the homogeneous setting is beyond the scope of our paper and is left for future work. Thus, we assume $\nabla f(x^\star) \neq \mathbf{0}$ from now on.

Now we turn to the results in Section 3.1. With $\mathcal{P}_{A^\perp}$ described in (13), we have

$$u_n = \left[(\bar{x}_n)^\top, (\bar{x}_n)^\top, \cdots, (\bar{x}_n)^\top\right]^\top,$$

where $\bar{x}_n = \frac{1}{N}\sum_{k=1}^{N} x_n^{(k)}$. As a result, $\mathbb{E}\|u_n - x^\star\|^2 = N\mathbb{E}\|\bar{x}_n - x^{(\star)}\|^2$. Then Theorem 3.1 guarantees that $\mathbb{E}\|\bar{x}_n - x^{(\star)}\|^2 = \mathcal{O}(n^{-\alpha\min\{1, 2-2\beta\}})$, which is what we desire. The revision of Theorem 3.2 is deferred to the last part of this subsection.

As for the results in Section 3.2, we first take a glance at Theorem 3.3. Since the expression of $\mathcal{P}_{A^\perp}$ implies we can just focus on the first $d$ dimensions of (7), Theorem 3.3 actually characterize the asymptotic behavior of $\frac{\bar{x}_n - x^{(\star)}}{\sqrt{\eta_{n-1}}}$ when $\beta \in [0, \frac{1}{2})$. Then we consider the bias vector mentioned in Corollary 1. Direct computation shows

$$\mathcal{P}_{A^\perp}\nabla^2 f(x^\star)\nabla f(x^\star) = \frac{1}{N^2}\mathcal{P}_{A^\perp}\begin{bmatrix}\nabla^2 g_1(x^{(\star)})\nabla g_1(x^{(\star)}) \\ \nabla^2 g_2(x^{(\star)})\nabla g_2(x^{(\star)}) \\ \vdots \\ \nabla^2 g_N(x^{(\star)})\nabla g_N(x^{(\star)})\end{bmatrix}$$

$$= \frac{1}{N^3} \begin{bmatrix} \mathbf{I}_d \\ \mathbf{I}_d \\ \vdots \\ \mathbf{I}_d \end{bmatrix} \sum_{k=1}^{N} \nabla^2 g_k(\boldsymbol{x}^{(\star)}) \nabla g_k(\boldsymbol{x}^{(\star)})$$

Even if $\nabla g_k(\boldsymbol{x}^{(\star)}) \neq 0$ for any $k$, the bias vector could still be equal to the zero vector. For example, $\nabla^2 g_k(\boldsymbol{x}^{(\star)})$ are all the same and $\sum_{k=1}^{N} \nabla g_k(\boldsymbol{x}^{(\star)}) = \mathbf{0}$. For such a special case, the convergence of $\mathbb{E} \left\| \bar{\boldsymbol{x}}_n - \boldsymbol{x}^{(\star)} \right\|_2^2$ could be faster, since the leading term hindering the convergence vanishes.

### A.2.1 Revision of the Lower Bound

Finally, we present a revised version of Theorem 3.2. Recall that the Hessian matrix $\nabla^2 f(\boldsymbol{x})$ is a block diagonal matrix. Although Theorem 3.1 provides a counter example for the general case, it does not specify the form of $\nabla^2 f(\boldsymbol{x})$. Fortunately, with $\boldsymbol{A}$ defined in (12), we can find a counter example such that $\nabla^2 f(\boldsymbol{x})$ is a diagonal matrix.

**Corollary 2.** *Consider the problem* (11). *If* $\eta_n = \eta_0 n^{-\alpha}$ *and* $p_n = \min\{p_0 \eta_n, 1\}$ *with* $0 < \alpha \leq 1$, *then there exists a quadratic function* $f(\boldsymbol{x})$ *such that* $\nabla^2 f(\boldsymbol{x})$ *is a diagonal matrix,* $\nabla^2 f(\boldsymbol{x}) \succeq \mathbf{I}_d$ *and* $\mathbb{E} \left\| \boldsymbol{u}_n - \boldsymbol{x}^\star \right\|^2$ *does not converge to* 0.

The proof of Corollary 2 is deferred to Appendix B.3.

### A.3 Related Work

In this section, we focus on several works that investigate the asymptotic and dynamical nature of distributed optimization. We can trace this line of research back from the classical work [22] by Kushner et al. Unlike the prevailing federated learning algorithm (multi-step local computation between adjacent communications), Kushner et al. [22] consider a random, incomplete decentralized communication within each iteration. And the randomness of these communications are characterized by a sequence of random gossip matrices $\{\mathbf{W}_n\}$. In particular, the algorithm has the following form,

$$\text{Local step: } \boldsymbol{x}_{n+\frac{1}{2},i} = \boldsymbol{x}_{n,i} + \epsilon \mathbf{Y}_{n,i}$$

$$\text{Gossip step: } \boldsymbol{x}_{n+1,i} = \sum_{j=1}^{N} \omega_{n+1}(i,j) \boldsymbol{x}_{n+\frac{1}{2},j} \tag{14}$$

where $\mathbf{W}_n = [\omega_n(i,j)]_{i,j=1}^{N}$ and $N$ is the number of nodes.

For the algorithm, Kushner et al. [22] proved that the trajectories of the final iteration converge weakly to the solution of the particular ODE as the step size $\epsilon$ converges to zero, and formally discussed the weak convergence of the rescaled sequences to the solution of a specific linear SDE (i.e. the diffusion approximation result). However, there are several limitations to this work. First of all, the most critical point is that the above theoretical results are discussed in the case of fixed step sizes. As the iteration increases, the variance term of a stochastic approximation begins to dominate the rate of convergence, and the use of a constant step size at this point will make the effect of variance never fall to zero. So a fixed step size means a fixed and finite total iteration (depending on the constant step size and the required estimation accuracy). This makes all the asymptotic results in [22] less practical. In addition, the article assumes (without proof) some intermediate results such as tightness and weak convergence at the initial point, making his theoretical results incomplete.

Thereafter, Bianchi et al. [2] consider the same random gossip stochastic approximation algorithm. Unlike [22], Bianchi et al. replace the fixed step size with a decreasing step size and obtain the asymptotic normality of rescaled final iteration and Polyak-Ruppert averaging sequence. But note that Bianchi et al. [2] assume that all gossip matrices have the same distribution, implying that their asymptotic results hold only if the communication frequency does not decrease as the iteration increases. This is equivalent to the case in LPSA where the projection probability is set as a constant. In particular, they assume the step size $\gamma_n \sim \frac{1}{n^\alpha}$ satisfies that $\alpha \in (\frac{1}{2}, 1]$. In the end, neither of the above two works analyzes the effect of communication frequency on the asymptotic performance of the distributed stochastic approximation algorithm, which is explicitly reflected in our analysis in the form of bias-variance tradeoff.

# B  Proof of Section 3.1

In this section, we give the proof of Theorems 3.1 and 3.2.

## B.1  Useful Propositions and Lemmas

In this subsection, we present some existing results and auxiliary lemmas useful for our later analysis.

**Proposition 2** ([32], Theorem 2.1.9, property of strong convexity). *If $f(\boldsymbol{x})$ is $\mu$-strongly convex, then we have*

$$\langle \nabla f(\boldsymbol{x}) - \nabla f(\boldsymbol{y}), \boldsymbol{x} - \boldsymbol{y} \rangle \geq \mu \|\boldsymbol{x} - \boldsymbol{y}\|^2, \ \forall \boldsymbol{x}, \boldsymbol{y} \in \mathbb{R}^d.$$

**Proposition 3** (Cauchy–Schwarz Inequality). *For any vectors $\boldsymbol{a}, \boldsymbol{b} \in \mathbb{R}^d$ and positive number $\gamma$, it holds that*

$$2 \langle \boldsymbol{a}, \boldsymbol{b} \rangle \leq \gamma \|\boldsymbol{a}\|^2 + \frac{1}{\gamma} \|\boldsymbol{b}\|^2.$$

*Moreover, for any positive integer $n$ and any vectors $\boldsymbol{x}_1, \boldsymbol{x}_2, \ldots, \boldsymbol{x}_n \in \mathbb{R}^d$, it holds that*

$$\left\| \sum_{i=1}^n \boldsymbol{x}_i \right\|^2 \leq n \sum_{i=1}^n \|\boldsymbol{x}_i\|^2.$$

**Proposition 4** ([25], Proposition 2.1 and Lemma B.1, property of projection). *Suppose that $\boldsymbol{A}$ is a $p \times q$ matrix. Let $\mathcal{P}_{\boldsymbol{A}}$ be the projection onto the column space of $\boldsymbol{A}$ and $\mathcal{P}_{\boldsymbol{A}^\perp}$ the projection onto the null space of $\boldsymbol{A}^\top$. Then we have*

1. *Linearity: $\mathcal{P}_{\boldsymbol{A}}(\alpha \boldsymbol{x} + \beta \boldsymbol{y}) = \alpha \mathcal{P}_{\boldsymbol{A}}(\boldsymbol{x}) + \beta \mathcal{P}_{\boldsymbol{A}}(\boldsymbol{y})$ for any $\boldsymbol{x}, \boldsymbol{y} \in \mathbb{R}^p$ and $\alpha, \beta \in \mathbb{R}$.*

2. *Non-expansiveness: $\max\{\|\mathcal{P}_{\boldsymbol{A}}(\boldsymbol{x}) - \mathcal{P}_{\boldsymbol{A}}(\boldsymbol{y})\|, \|\mathcal{P}_{\boldsymbol{A}^\perp}(\boldsymbol{x}) - \mathcal{P}_{\boldsymbol{A}^\perp}(\boldsymbol{y})\|\} \leq \|\boldsymbol{x} - \boldsymbol{y}\|$ for any $\boldsymbol{x}, \boldsymbol{y} \in \mathbb{R}^p$.*

3. *Orthogonality: any $\boldsymbol{x} \in \mathbb{R}^p$ can be decomposed uniquely into $\boldsymbol{x} = \boldsymbol{u} + \boldsymbol{v}$ where $\boldsymbol{u} = \mathcal{P}_{\boldsymbol{A}^\perp}(\boldsymbol{x})$ and $\boldsymbol{v} = \boldsymbol{P}_{\boldsymbol{A}}(\boldsymbol{x})$ satisfying $\langle \boldsymbol{u}, \boldsymbol{v} \rangle = 0$.*

*More specifically, we have $\mathcal{P}_{\boldsymbol{A}}(\boldsymbol{x}) = \boldsymbol{A}(\boldsymbol{A}^\top \boldsymbol{A})^\dagger \boldsymbol{A}^\top \boldsymbol{x} = (\boldsymbol{A}^\top)^\dagger \boldsymbol{A}^\top \boldsymbol{x}$ and $\mathcal{P}_{\boldsymbol{A}^\perp}(\boldsymbol{x}) = \boldsymbol{I}_p - \mathcal{P}_{\boldsymbol{A}}(\boldsymbol{x}) = \left(\boldsymbol{I}_p - \boldsymbol{A}(\boldsymbol{A}^\top \boldsymbol{A})^\dagger \boldsymbol{A}^\top\right) \boldsymbol{x} = \left(\boldsymbol{I}_p - (\boldsymbol{A}^\top)^\dagger \boldsymbol{A}^\top\right) \boldsymbol{x}$ with $\dagger$ the pseudo inverse.*

**Proposition 5** (Stolz–Cesàro theorem). *Let $\{a_n\}$ and $\{b_n\}$ be two sequences of real numbers such that*

1. *$0 < b_1 < b_2 < \cdots < b_n < \ldots$ and $\lim_{t \to \infty} b_t = \infty$.*

2. *$\lim_{n \to \infty} \frac{a_{n+1} - a_n}{b_{n+1} - b_n} = l \in \mathbb{R}$.*

*Then, $\lim_{n \to \infty} \frac{a_n}{b_n}$ exists and is equal to $l$.*

**Lemma 1.** *Let $\{r_n\} \subset (0, 1)$ be a sequence of positive numbers that decays to zero monotonically. If $\frac{r_n}{r_{n+1}} - 1 = o(r_n)$, for $p \geq 1$, we have that*

$$\lim_{T \to \infty} \frac{\sum_{n=1}^T r_n^p \prod_{s=n+1}^T (1 - r_s)}{r_T^{p-1}} = 1.$$

**Lemma 2.** *Let $\{r_n\} \subset (0, 1)$ be a sequence of positive numbers that decays to zero monotonically and $a$ is a positive number. If $\frac{r_n}{r_{n+1}} - 1 = ar_n + o(r_n)$, for $p \geq 1$ and $1/a > p - 1$, we have*

$$\lim_{T \to \infty} \frac{\sum_{n=1}^T r_t^p \prod_{s=n+1}^T (1 - r_s)}{r_T^{p-1}} = \frac{1}{1 - a(p-1)}.$$

**Lemma 3.** *Let $\{r_n\} \subset (0, 1)$ be a sequence of positive numbers that decays to zero monotonically and $\{s_n\}$ is a sequence of positive numbers. If $\frac{r_n}{r_{n+1}} - 1 = ar_n + o(r_n)$ for $a \geq 0$ and $s_{n+1} = (1 - r_n)s_n + o(r_n)$. Then we have $s_n = o(1)$.*

The proof of the three lemmas are deferred to Appendix B.4.

## B.2 Proof of Theorem 3.1

In this subsection, we give the formal statement of Theorem 3.1 and its proof. Before that, we first present the one-step descent lemmas of $\mathbb{E} \|\boldsymbol{u}_n - \boldsymbol{x}^\star\|^2$ and $\mathbb{E} \|\boldsymbol{v}_n\|^2$, whose proof is deferred to Appendix B.5.

**Lemma 4** (One-step descent of $\mathbb{E} \|\boldsymbol{u}_n - \boldsymbol{x}^\star\|^2$). *Suppose that Assumptions 1, 2 and 4 hold. Then there exists a $n_0$ such that for any $n \geq n_0$,*

$$\mathbb{E}\|\boldsymbol{u}_{n+1} - \boldsymbol{x}^\star\|^2 \leq (1 - \mu\eta_n)\mathbb{E}\|\boldsymbol{u}_n - \boldsymbol{x}^\star\|^2 + \frac{3L^2}{\mu}\eta_n\mathbb{E}\|\boldsymbol{v}_n\|^2 + 2\eta_n^2\Sigma_\star^{(1)}, \tag{15}$$

*where $\Sigma_\star^{(1)} := \mathbb{E} \|\mathcal{P}_{\boldsymbol{A}^\perp}\xi^\star\|^2$ with $\xi^\star = \nabla f(\boldsymbol{x}^\star) - \nabla f(\boldsymbol{x}^\star, \zeta), \zeta \sim \mathcal{D}$.*

**Lemma 5** (One-step descent of $\mathbb{E} \|\boldsymbol{v}_n\|^2$). *Suppose that Assumptions 1, 2 and 4 hold. Then there exists a $n_0$ such that for any $n \geq n_0$*

$$\mathbb{E}\|\boldsymbol{v}_{n+1}\|^2 \leq \left(1 - \frac{p_n}{2}\right)\mathbb{E}\|\boldsymbol{v}_n\|^2 + \frac{7L^2\eta_n^2}{p_n}\mathbb{E}\|\boldsymbol{u}_n - \boldsymbol{x}^\star\|^2 + \frac{7L^2\eta_n^2}{p_n}\|\nabla f(\boldsymbol{x}^\star)\|^2 + 2\eta_n^2\Sigma_\star^{(2)}, \tag{16}$$

*where $\Sigma_\star^{(2)} := \mathbb{E} \|\mathcal{P}_{\boldsymbol{A}}\xi^\star\|^2$ with $\xi^\star = \nabla f(\boldsymbol{x}^\star) - \nabla f(\boldsymbol{x}^\star, \zeta), \zeta \sim \mathcal{D}$.*

Now we are prepared to give the formal statement of Theorem 3.1.

**Theorem B.1** (Formal statement of Theorem 3.1). *Suppose that Assumptions 1, 2 and 4 hold. Let $\eta_n = \eta_0 n^{-\alpha}$ and $p_n = \min\{p_0\eta_n^\beta, 1\}$ with $0 \leq \beta < 1$. Then for (i) $0 < \alpha < 1$ or (ii) $\alpha = 1$ with $\eta_0 > 2/\mu$, we have*

$$\mathbb{E}\|\boldsymbol{u}_n - \boldsymbol{x}^\star\|^2 = \mathcal{O}\left(\eta_n + \eta_n^{2-2\beta}\right)$$
$$\mathbb{E}\|\boldsymbol{v}_n\|^2 = \mathcal{O}\left(\eta_n^{2-2\beta}\right)$$

*Proof.* (**Proof of Theorem B.1**)

Let $z_n = \|\boldsymbol{u}_n - \boldsymbol{x}^\star\|^2 + c_0\sqrt{\frac{p_n}{\eta_n}}\|\boldsymbol{v}_n\|^2$ with $c_0 = \sqrt{3/(7\mu)}$. By Lemmas 4 and 5, there exists a $n_0$ such that for any $n \geq n_0$, we have

$$\mathbb{E}z_{n+1} \leq \left(1 - \min\left\{\mu\eta_n, \frac{p_n}{2}\right\} + 7c_0L^2\frac{\eta_n^{3/2}}{p_n^{1/2}}\right)\mathbb{E}z_n + 2\eta_n^2\Sigma_\star^{(1)}$$
$$+ 7c_0L^2\frac{\eta_n^{3/2}}{p_n^{1/2}}\|\nabla f(\boldsymbol{x}^\star)\|^2 + 2c_0\eta_n^{3/2}p_n^{1/2}\Sigma_\star^{(2)}.$$

With $p_n = \min\{p_0\eta_n^\beta, 1\}$ for some $0 \leq \beta < 1$, there exists a $n_1 \geq n_0$ such that for any $n \geq n_1$, we have $p_n = p_0\eta_n^\beta$ and

$$\mathbb{E}z_{n+1} \leq \left(1 - \frac{\mu\eta_n}{2}\right)\mathbb{E}z_n + 2\eta_n^2\Sigma_\star^{(1)} + \frac{7c_0L^2}{\sqrt{p_0}}\eta_n^{3/2-\beta/2}\|\nabla f(\boldsymbol{x}^\star)\|^2 + 2c_0\sqrt{p_0}\eta_n^{3/2+\beta/2}\Sigma_\star^{(2)}. \tag{17}$$

For any $T \geq n_1$, applying the recursion (17) $(T - n_1)$ times yields

$$\mathbb{E}z_T \leq \mathbb{E}z_{n_1}\prod_{n=n_1}^{T-1}\left(1 - \frac{\mu\eta_n}{2}\right) + 2\Sigma_\star^{(1)}\sum_{n=n_1}^{T-1}\eta_n^2\prod_{s=n+1}^{T-1}\left(1 - \frac{\mu\eta_s}{2}\right)$$
$$+ \frac{7c_0L^2}{\sqrt{p_0}}\|\nabla f(\boldsymbol{x}^\star)\|^2\sum_{n=n_1}^{T-1}\eta_n^{3/2-\beta/2}\prod_{s=n+1}^{T-1}\left(1 - \frac{\mu\eta_s}{2}\right) \tag{18}$$
$$+ 2c_0\sqrt{p_0}\Sigma_\star^{(2)}\sum_{n=n_1}^{T-1}\eta_n^{3/2+\beta/2}\prod_{s=n+1}^{T-1}\left(1 - \frac{\mu\eta_s}{2}\right).$$

For case (i) where $0 < \alpha < 1$, we have $\eta_n = \eta_0 n^{-\alpha}$. Thus, for the first term, we have

$$\mathbb{E}z_{n_1} \prod_{n=n_1}^{T-1} \left(1 - \frac{\mu\eta_n}{2}\right) \leq \mathbb{E}z_{n_1} \exp\left(-\frac{\mu}{2}\sum_{n=n_1}^{T-1}\eta_n\right)$$

$$\leq \mathbb{E}z_{n_1} \exp\left(-\frac{\mu\eta_0(T^{1-\alpha} - n_1^{1-\alpha})}{2(1-\alpha)}\right).$$

For other terms, one can check that $\frac{\eta_n}{\eta_{n+1}} - 1 = o(\eta_n)$. Then by Lemma 1, we have

$$\mathbb{E}\|\boldsymbol{u}_n - \boldsymbol{x}^\star\|^2 \leq \mathbb{E}z_n = \mathcal{O}\left(\frac{c_0 L^2 \|\nabla f(\boldsymbol{x}^\star)\|^2}{\sqrt{p_0}\,\mu}\eta_n^{1/2-\beta/2}\right) = \mathcal{O}\left(\frac{L^2 \|\nabla f(\boldsymbol{x}^\star)\|^2}{\sqrt{p_0}\mu}\eta_n^{1/2-\beta/2}\right). \quad (19)$$

This implies that there exists a positive number $c_1$ such that $\mathbb{E}\|\boldsymbol{u}_n - \boldsymbol{x}^\star\|^2 \leq c_1\frac{L^2\|\nabla f(\boldsymbol{x}^\star)\|^2}{\sqrt{p_0}\mu}\eta_n^{1/2-\beta/2}$ for any $n \geq n_1$. Substituting this into (16) yields that

$$\mathbb{E}\|\boldsymbol{v}_{n+1}\|^2 \leq \left(1 - \frac{p_0\eta_n^\beta}{2}\right)\mathbb{E}\|\boldsymbol{v}_n\|^2 + \frac{7c_1 L^4 \|\nabla f(\boldsymbol{x}^\star)\|^2}{p_0^{3/2}\sqrt{\mu}}\eta_n^{5/2-3\beta/2}\mathbb{E}\|\boldsymbol{u}_n - \boldsymbol{x}^\star\|^2$$

$$+ \frac{7L^2}{p_0}\eta_n^{2-\beta}\|\nabla f(\boldsymbol{x}^\star)\|^2 + 2\eta_n^2\Sigma_\star^{(2)}$$

hold for any $n \geq n_1$. Following the same argument as before, we can prove

$$\mathbb{E}\|\boldsymbol{v}_n\|^2 = \mathcal{O}\left(\frac{L^2 \|\nabla f(\boldsymbol{x}^\star)\|^2}{p_0^2}\eta_n^{2-2\beta}\right). \quad (20)$$

Then there exists $c_2 > 0$ such that $\mathbb{E}\|\boldsymbol{v}_n\|^2 \leq \frac{c_2 L^2\|\nabla f(\boldsymbol{x}^\star)\|_2^2}{p_0^2}\eta_n^{2-2\beta}$ for any $n \geq n_1$. Substituting this into (15) yields that

$$\mathbb{E}\|\boldsymbol{u}_{n+1} - \boldsymbol{x}^\star\|^2 \leq (1 - \mu\eta_n)\mathbb{E}\|\boldsymbol{u}_n - \boldsymbol{x}^\star\|^2 + \frac{3c_2 L^4 \|\nabla f(\boldsymbol{x}^\star)\|_2^2}{\mu p_0^2}\eta_n^{3-2\beta} + 2\eta_n^2\Sigma_\star^{(1)}$$

hold for any $n \geq n_1$. Following the same procedure again, we can obtain

$$\mathbb{E}\|\boldsymbol{u}_n - \boldsymbol{x}^\star\|^2 = \mathcal{O}\left(\frac{\Sigma_\star^{(1)}}{\mu}\eta_n + \frac{L^4 \|\nabla f(\boldsymbol{x}^\star)\|^2}{\mu^2 p_0^2}\eta_n^{2-2\beta}\right). \quad (21)$$

For case (ii) where $\alpha = 1$ with $\eta_0 > 2/\mu$, we can still obtain (18). Since $\eta_n = \eta_0 t^{-1}$, for the first term on the right-hand side of (18), we have

$$\mathbb{E}z_{n_1} \prod_{n=n_1}^{T-1} \left(1 - \frac{\mu\eta_n}{2}\right) \leq \mathbb{E}z_{n_1} \exp\left(-\frac{\mu}{2}\sum_{n=n_1}^{T-1}\eta_n\right)$$

$$\leq \mathbb{E}z_{n_1} \exp\left(-\frac{\mu\eta_0(\ln T - \ln n_1)}{2}\right)$$

$$= \mathcal{O}\left(T^{-\mu\eta_0/2}\right).$$

For other terms, one can check that $\frac{\eta_n}{\eta_{t+1}} - 1 = \frac{2}{\mu\eta_0}\cdot\frac{\mu\eta_n}{2} + o(\eta_n)$. Then by Lemma 2, we have (19) holds for $\eta_0 > 2/\mu$. Following the same procedure as before, we can also obtain (20). Substituting this into (15) yields that

$$\mathbb{E}\|\boldsymbol{u}_{n+1} - \boldsymbol{x}^\star\|^2 \leq (1 - \mu\eta_n)\mathbb{E}\|\boldsymbol{u}_n - \boldsymbol{x}^\star\|^2 + \frac{3c_2 L^2}{\mu}\eta_n^{3-2\beta} + 2\eta_n^2\Sigma_\star^{(1)}$$

holds for any $n \geq n_1$. Since $\frac{\eta_n}{\eta_{t+1}} - 1 = \frac{1}{\mu\eta_0}\cdot\mu\eta_n + o(\eta_n)$, following the same procedure as before, we can obtain (21) for $\eta_0 > 2/\mu > \max\{2 - 2\beta, 1\}/\mu$. $\qquad\square$

## B.3 Proof of Theorem 3.2

We first give a formal statement of Theorem 3.2 that can combines Theorem 3.2 and Corollary 2.

**Theorem B.2.** *If $\eta_n = \eta_0 n^{-\alpha}$ and $p_n = \min\{p_0\eta_n, 1\}$ with $0 < \alpha \leq 1$, for a specific $\boldsymbol{A} \in \mathbb{R}^{p \times r}$ with $r < p$, there exists a quadratic function $f(\boldsymbol{x})$ defined on $\mathbb{R}^p$ so that $\nabla^2 f(\boldsymbol{x}) \succeq \mathbf{I}_p$ and $\mathbb{E}\|\boldsymbol{u}_n - \boldsymbol{x}^\star\|^2$ does not converge to 0. Here $\mathbf{I}_p \in \mathbb{R}^{p \times p}$ is the identity matrix, and $\nabla^2 f(\boldsymbol{x}) \succeq \mathbf{I}_p$ means $\nabla^2 f(\boldsymbol{x}) - \mathbf{I}_p$ is positive semidefinite. Moreover, if $\mathcal{P}_{\boldsymbol{A}}$ is not of the form $\mathcal{P}_{\boldsymbol{A}} = \sum_{i \in I} \boldsymbol{e}_i \boldsymbol{e}_i^\top$, where $I \subseteq \{1, 2, \ldots, p\}$ and $\boldsymbol{e}_i$ is the unit vector in $\mathbb{R}^p$ with the $i$-th element equal to 1, $\nabla^2 f(\boldsymbol{x})$ can be chosen as a diagonal matrix such that $\nabla^2 f(\boldsymbol{x}) \succeq \mathbf{I}_p$.*

Before give the proof of Theorem B.2, we first give the proof of Corollary 2 based on Theorem B.1.

*Proof.* (**Proof of Corollary 2**)

With $\boldsymbol{A}$ defined in (12), we have $p = Nd$ and $r = (N-1)d$. Recall that for $\boldsymbol{x} = \left[(\boldsymbol{x}^{(1)})^\top, (\boldsymbol{x}^{(2)})^\top, \cdots, (\boldsymbol{x}^{(N)})^\top\right]^\top \in \mathbb{R}^{Nd}$, we have

$$\mathcal{P}_{\boldsymbol{A}^\perp}(\boldsymbol{x}) = \left[\bar{\boldsymbol{x}}^\top, \bar{\boldsymbol{x}}^\top, \cdots, \bar{\boldsymbol{x}}^\top\right]^\top$$

where $\bar{\boldsymbol{x}} = \frac{1}{N}\sum_{k=1}^N \boldsymbol{x}^{(k)}$. As a result, we have $\mathcal{P}_{\boldsymbol{A}^\perp}\boldsymbol{e}_1 = \frac{1}{N}\sum_{k=0}^{N-1}\boldsymbol{e}_{1+kd}$. This implies that $\mathcal{P}_{\boldsymbol{A}^\perp}$ can not be of the form $\mathcal{P}_{\boldsymbol{A}^\perp} = \sum_{i \in I}\boldsymbol{e}_i\boldsymbol{e}_i^\top$. Thus, $\nabla^2 f(\boldsymbol{x})$ can be chosen as a diagonal matrix. $\square$

Now we present the proof of Theorem B.2.

*Proof.* (**Proof of Theorem B.2**)

Consider the quadratic function $f(\boldsymbol{x}) = \frac{1}{2}\boldsymbol{x}^\top \boldsymbol{B}\boldsymbol{x} + \boldsymbol{c}^\top \boldsymbol{x}$ where the positive definite matrix $\boldsymbol{B} \in \mathbb{R}^{p \times p}$ and the vector $\boldsymbol{c} \in \mathbb{R}^p$ are specified later.

**The exact solution to problem** (1)    We first compute the exact solution to problem (1), where $\boldsymbol{A} \in \mathbb{R}^{p \times r}$ for some positive integer $r < p$. With out loss of generalization, we assume $\text{rank}(\boldsymbol{A}) = r$.

Suppose that the singular value decomposition (SVD) of $\boldsymbol{A}$ is $\boldsymbol{A} = \boldsymbol{U}\boldsymbol{D}_{\boldsymbol{A}}\boldsymbol{V}^\top$ where $\boldsymbol{U} \in \mathbb{R}^{p \times p}$ and $\boldsymbol{V} \in \mathbb{R}^{r \times r}$ are orthogonal matrices and $\boldsymbol{D}_A \in \mathbb{R}^{p \times r}$ is a rectangular diagonal matrix with diagonal entries in descending order. One can check that the solution to $\boldsymbol{A}^\top \boldsymbol{x} = \boldsymbol{0}$ has the form $\boldsymbol{x} = \left(\mathbf{I}_p - (\boldsymbol{A}^\top)^\dagger \boldsymbol{A}^\top\right)\boldsymbol{w} = \mathcal{P}_{\boldsymbol{A}^\perp}(\boldsymbol{w})$ where $\boldsymbol{w}$ is an arbitrary vector in $\mathbb{R}^p$ and $(\boldsymbol{A}^\top)^\dagger$ is the pseudo inverse of $\boldsymbol{A}^\top$. From the SVD of $\boldsymbol{A}$, we have

$$\mathbf{I}_p - (\boldsymbol{A}^\top)^\dagger \boldsymbol{A}^\top = \boldsymbol{U}\begin{bmatrix} \boldsymbol{0}_r & \boldsymbol{0}_{r \times (p-r)} \\ \boldsymbol{0}_{(p-r) \times r} & \mathbf{I}_{p-r} \end{bmatrix}\boldsymbol{U}^\top,$$

where $\boldsymbol{0}_{m \times n} \in \mathbb{R}^{m \times n}$ denote the zero matrix and reduces to $\boldsymbol{0}_n \in \mathbb{R}^{n \times n}$ for $m = n$. We denote the first $r$ columns of $\boldsymbol{U}$ by $\boldsymbol{U}_1$ and last $p - r$ columns of $\boldsymbol{U}$ by $\boldsymbol{U}_2$ for simplicity, Then the problem (1) becomes the following unconstrained problem

$$\min_{\boldsymbol{w} \in \mathbb{R}^p} \frac{1}{2}\boldsymbol{w}^\top \left(\mathbf{I}_p - (\boldsymbol{A}^\top)^\dagger \boldsymbol{A}^\top\right)^\top \boldsymbol{B}\left(\mathbf{I}_p - (\boldsymbol{A}^\top)^\dagger \boldsymbol{A}^\top\right)\boldsymbol{w} + \boldsymbol{w}^\top \left(\mathbf{I}_p - (\boldsymbol{A}^\top)^\dagger \boldsymbol{A}^\top\right)^\top \boldsymbol{c}.$$

$$= \min_{\boldsymbol{w}_2 \in \mathbb{R}^{p-r}} \frac{1}{2}\boldsymbol{w}_2^\top \boldsymbol{B}_2 \boldsymbol{w}_2 + \boldsymbol{w}_2^\top \boldsymbol{c}_2,$$

where $\boldsymbol{w}_2 = \boldsymbol{U}_2^\top \boldsymbol{w}$, $\boldsymbol{B}_2 = \boldsymbol{U}_2^\top \boldsymbol{B}\boldsymbol{U}_2$ and $\boldsymbol{c}_2 = \boldsymbol{U}_2^\top \boldsymbol{c}$. The solution is $\boldsymbol{w}_2^\star = -\boldsymbol{B}_2^{-1}\boldsymbol{c}_2$. From the expression of $\mathbf{I}_p - (\boldsymbol{A}^\top)^\dagger \boldsymbol{A}^\top$, we know that the first $r$ elements of $\boldsymbol{U}^\top \boldsymbol{w}$ will not affect the value of $\boldsymbol{x}$. Thus, the solution to the original problem (1) is $\boldsymbol{x}^\star = -\boldsymbol{U}_2 \boldsymbol{B}_2^{-1}\boldsymbol{c}_2$.

Moreover, one can check

$$\mathcal{P}_{\boldsymbol{A}} = \boldsymbol{U}\begin{bmatrix} \mathbf{I}_r & \boldsymbol{0}_{r \times (p-r)} \\ \boldsymbol{0}_{(p-r) \times r} & \boldsymbol{0}_{(p-r) \times (p-r)]} \end{bmatrix}\boldsymbol{U}^\top = \boldsymbol{U}_1 \boldsymbol{U}_1^\top$$

and

$$\mathcal{P}_{\boldsymbol{A}^\perp} = \boldsymbol{U}\begin{bmatrix} \boldsymbol{0}_{r \times r} & \boldsymbol{0}_{r \times (p-r)} \\ \boldsymbol{0}_{(p-r) \times r} & \mathbf{I}_{p-r} \end{bmatrix}\boldsymbol{U}^\top = \boldsymbol{U}_2 \boldsymbol{U}_2^\top.$$

**Recursions of $\mathbb{E}u_n$ and $\mathbb{E}v_n$**   From the definition of $u_n$ and the linearity of $\mathcal{P}_{A^\perp}$, we have

$$
\begin{aligned}
u_{n+1} - x^\star &= \mathcal{P}_{A^\perp}(x_n - \eta_n B x_n - \eta_n c + \eta_n \xi_n) - x^\star \\
&= u_n - x^\star - \eta_n \mathcal{P}_{A^\perp}(B x_n + c) + \eta_n \mathcal{P}_{A^\perp}\xi_n \\
&= u_n - x^\star - \eta_n \mathcal{P}_{A^\perp} B(u_n - x^\star) - \eta_n \mathcal{P}_{A^\perp} B v_n - \eta_n \mathcal{P}_{A^\perp}(B x^\star + c) + \eta_n \mathcal{P}_{A^\perp}\xi_n \\
&= u_n - x^\star - \eta_n \mathcal{P}_{A^\perp} B \mathcal{P}_{A^\perp}(u_n - x^\star) - \eta_n \mathcal{P}_{A^\perp} B v_n - \eta_n \mathcal{P}_{A^\perp}(B x^\star + c) + \eta_n \mathcal{P}_{A^\perp}\xi_n.
\end{aligned}
$$

The optimality of $x^\star$ implies that $\mathcal{P}_{A^\perp}(B x^\star + c) = 0$. Taking expectation yields

$$
\mathbb{E}u_{n+1} - x^\star = (\mathbf{I}_p - \eta_n \mathcal{P}_{A^\perp} B \mathcal{P}_{A^\perp})(\mathbb{E}u_n - x^\star) - \eta_n \mathcal{P}_{A^\perp} B\, \mathbb{E}v_n. \tag{22}
$$

As for the iteration of $\mathbb{E}v_t$. From the definition of $v_n$, with probability $1 - p_n$ we have

$$
\begin{aligned}
v_{t+1} &= \mathcal{P}_A(x_n - \eta_n B x_n - \eta_n c + \eta_n \xi_n) \\
&= v_n - \eta_n \mathcal{P}_A(B x_n + c) + \eta_n \mathcal{P}_A \xi_n \\
&= v_n - \eta_n \mathcal{P}_A B(u_n - x^\star) - \eta_n \mathcal{P}_A B v_n - \eta_n \mathcal{P}_A(B x^\star + c) + \eta_n \mathcal{P}_A \xi_n \\
&= (\mathbf{I}_p - \eta_n \mathcal{P}_A B \mathcal{P}_A)v_n - \eta_n \mathcal{P}_A B(u_n - x^\star) - \eta_n \mathcal{P}_A(B x^\star + c) + \eta_n \mathcal{P}_A \xi_n,
\end{aligned}
$$

and with probability $p_n$ we have $v_{n+1} = 0$. Taking expectation yields

$$
\mathbb{E}v_{n+1} = (1 - p_n)(\mathbf{I}_p - \eta_n \mathcal{P}_A B \mathcal{P}_A)\mathbb{E}v_n - (1 - p_n)\eta_n \left[\mathcal{P}_A B(\mathbb{E}u_n - x^\star) + \mathcal{P}_A(B x^\star + c)\right]. \tag{23}
$$

**Simultaneous diagonalization of $\mathcal{P}_A B \mathcal{P}_A$ and $\mathcal{P}_{A^\perp} B \mathcal{P}_{A^\perp}$**   We first express the two matrices as follows:

$$
\begin{aligned}
\mathcal{P}_A B \mathcal{P}_A &= U \begin{bmatrix} \mathbf{I}_r & \mathbf{0}_{r \times (n-r)} \\ \mathbf{0}_{(n-r) \times r} & \mathbf{0}_{n-r} \end{bmatrix} U^\top B U \begin{bmatrix} \mathbf{I}_r & \mathbf{0}_{r \times (n-r)} \\ \mathbf{0}_{(n-r) \times r} & \mathbf{0}_{n-r} \end{bmatrix} U^\top \\
&= U \begin{bmatrix} B_1 & \mathbf{0}_{r \times (n-r)} \\ \mathbf{0}_{(n-r) \times r} & \mathbf{0}_{n-r} \end{bmatrix} U^\top, \\
\mathcal{P}_{A^\perp} B \mathcal{P}_{A^\perp} &= U \begin{bmatrix} \mathbf{0}_r & \mathbf{0}_{r \times (n-r)} \\ \mathbf{0}_{(n-r) \times r} & \mathbf{I}_{n-r} \end{bmatrix} U^\top B U \begin{bmatrix} \mathbf{0}_r & \mathbf{0}_{r \times (n-r)} \\ \mathbf{0}_{(n-r) \times r} & \mathbf{I}_{n-r} \end{bmatrix} U^\top \\
&= U \begin{bmatrix} \mathbf{0}_r & \mathbf{0}_{r \times (n-r)} \\ \mathbf{0}_{(n-r) \times r} & B_2 \end{bmatrix} U^\top,
\end{aligned}
$$

where $B_1 = U_1^\top B U_1$ and $B_2 = U_2^\top B U_2$ are positive definite. We suppose the eigenvalue decomposition of $B_1$ and $B_2$ is $B_1 = Q_1 D_{B_1} Q_1^\top$ and $B_2 = Q_2 D_{B_2} Q_2^\top$. With $Q := \begin{bmatrix} Q_1 & \mathbf{0}_{r \times (n-r)} \\ \mathbf{0}_{(n-r) \times r} & Q_2 \end{bmatrix}$ and $P := UQ$, we obtain the eigenvalue decomposition of $\mathcal{P}_{A^\perp} B \mathcal{P}_{A^\perp}$ and $\mathcal{P}_A B \mathcal{P}_A$ as follows

$$
\mathcal{P}_A B \mathcal{P}_A = P \begin{bmatrix} D_{B_1} & \mathbf{0}_{r \times (n-r)} \\ \mathbf{0}_{(n-r) \times r} & \mathbf{0}_{n-r} \end{bmatrix} P^\top =: P \widetilde{D}_{B_1} P^\top
$$

and

$$
\mathcal{P}_{A^\perp} B \mathcal{P}_{A^\perp} = P \begin{bmatrix} \mathbf{0}_r & \mathbf{0}_{r \times (n-r)} \\ \mathbf{0}_{(n-r) \times r} & D_{B_2} \end{bmatrix} P^\top =: P \widetilde{D}_{B_2} P^\top.
$$

**Proof by contradiction**   Left multiplication of (23) by $P^\top$ yields

$$
\mathbb{E}\tilde{v}_{n+1} = (\mathbf{I}_p - \widetilde{D}_n)\mathbb{E}\tilde{v}_n - \eta_n(1 - p_n)B_0(\mathbb{E}u_n - x^\star) - (1 - p_n)\eta_n c_0. \tag{24}
$$

where $\tilde{v}_n := P^\top v_n$, $B_0 := P^\top \mathcal{P}_A B$, $\widetilde{D}_n := \eta_n \widetilde{D}_{B_1} + p_n \mathbf{I}_p - \eta_n p_n \widetilde{D}_{B_1}$ and $c_0 := P^\top \mathcal{P}_A(B x^\star + c)$. Adding $(p_0 \mathbf{I}_p + \widetilde{D}_{B_1})^{-1} c_0$ to both sides of (24), we obtain

$$
\begin{aligned}
&\mathbb{E}\tilde{v}_{n+1} + (p_0 \mathbf{I}_p + \widetilde{D}_{B_1})^{-1} c_0 \\
&= (\mathbf{I}_p - \widetilde{D}_n)\mathbb{E}\tilde{v}_n - \eta_n(1 - p_n)B_0(\mathbb{E}u_n - x^\star) - (1 - p_n)\eta_n c_0 + (p_0 \mathbf{I}_p + \widetilde{D}_{B_1})^{-1} c_0 \\
&= (\mathbf{I}_p - \widetilde{D}_n)[\mathbb{E}\tilde{v}_n + (p_0 \mathbf{I}_p + \widetilde{D}_{B_1})^{-1} c_0] - \eta_n(1 - p_n)B_0(\mathbb{E}u_n - x^\star)
\end{aligned}
$$

$$+ [p_n - p_0\eta_n(1 - p_n)](p_0\mathbf{I}_p + \widetilde{\boldsymbol{D}}_{B_1})^{-1}\boldsymbol{c}_0. \tag{25}$$

Suppose $\mathbb{E}\|\boldsymbol{u}_n - \boldsymbol{x}^\star\|^2 = o(1)$, which implies $\mathbb{E}\boldsymbol{u}_n - \boldsymbol{x}^\star = o(1)$. Let $\widetilde{\boldsymbol{D}}_n = \mathrm{diag}\left\{\tilde{d}_{n,1}, \tilde{d}_{n,2}, \ldots, \tilde{d}_{n,p}\right\}$ amd $\widetilde{\boldsymbol{D}}_{B_1} = \mathrm{diag}\left\{\tilde{d}_{B_1,1}, \tilde{d}_{B_1,2}, \ldots, \tilde{d}_{B_1,p}\right\}$. Left multiplication of (25) by $\boldsymbol{e}_i^\top$ gives

$$\left|\mathbb{E}\boldsymbol{e}_i^\top \tilde{\boldsymbol{v}}_{n+1} + \frac{1}{p_0 + \tilde{d}_{B_1,i}}\boldsymbol{e}_i^\top \boldsymbol{c}_0\right| \le (1 - \tilde{d}_{n,i})\left|\mathbb{E}\boldsymbol{e}_i^\top \tilde{\boldsymbol{v}}_n + \frac{1}{p_0 + \tilde{d}_{B_1,i}}\boldsymbol{e}_i^\top \boldsymbol{c}_0\right| + o(\eta_n),$$

where $\boldsymbol{e}_i$ is the unit vector with the $i$-th element equal to 1. Since $\tilde{d}_{n,i} = \eta_n d_{B_1,i}(1 - p_n) + p_n$ and $p_n = \min\{p_0\eta_n, 1\}$, $o(\eta_n) = o(\tilde{d}_{n,i})$. Lemma 3 implies $\mathbb{E}\boldsymbol{e}_i^\top \tilde{\boldsymbol{v}}_{n+1} = -\frac{1}{p_0 + \tilde{d}_{B_1,i}}\boldsymbol{e}_i^\top \boldsymbol{c}_0 + o(1)$. It follows that $\mathbb{E}\tilde{\boldsymbol{v}}_n = -(p_0\mathbf{I}_p + \widetilde{\boldsymbol{D}}_{B_1})^{-1}\boldsymbol{c}_0 + o(1)$. Thus we have

$$\mathbb{E}\boldsymbol{v}_n = -\boldsymbol{P}(p_0\mathbf{I}_p + \widetilde{\boldsymbol{D}}_{B_1})^{-1}\boldsymbol{P}^\top \mathcal{P}_{\boldsymbol{A}}(\boldsymbol{B}\boldsymbol{x}^\star + \boldsymbol{c}) + o(1). \tag{26}$$

Denote the limit of $\mathbb{E}\boldsymbol{v}_n$ by $\boldsymbol{v}_\infty$ and we come back to the iteration (22). Left multiplication of (22) by $\boldsymbol{P}^\top$ yields

$$\mathbb{E}\tilde{\boldsymbol{u}}_{n+1} = (\mathbf{I}_p - \eta_n\widetilde{\boldsymbol{D}}_{B_2})\mathbb{E}\tilde{\boldsymbol{u}}_n - \eta_n\boldsymbol{P}^\top \mathcal{P}_{\boldsymbol{A}^\perp}\boldsymbol{B}\boldsymbol{v}_\infty + o(\eta_n),$$

where $\tilde{\boldsymbol{u}}_n = \boldsymbol{P}^\top(\boldsymbol{u}_n - \boldsymbol{x}^\star)$. Similar to the above argument, adding $\boldsymbol{P}^\top \mathcal{P}_{\boldsymbol{A}^\perp}\boldsymbol{B}\boldsymbol{v}_\infty$ to both sides and using Lemma 3, we can obtain

$$\mathbb{E}\tilde{\boldsymbol{u}}_n = -\boldsymbol{P}^\top \mathcal{P}_{\boldsymbol{A}^\perp}\boldsymbol{B}\boldsymbol{v}_\infty + o(1)$$
$$= \boldsymbol{P}^\top \mathcal{P}_{\boldsymbol{A}^\perp}\boldsymbol{B}\boldsymbol{P}(p_0\mathbf{I}_p + \widetilde{\boldsymbol{D}}_{B_1})^{-1}\boldsymbol{P}^\top \mathcal{P}_{\boldsymbol{A}}(\boldsymbol{B}\boldsymbol{x}^\star + \boldsymbol{c}) + o(1).$$

It remains to prove that there exists a positive definite matrix $\boldsymbol{B} \in \mathbb{R}^{p \times p}$ and a vector $\boldsymbol{c} \in \mathbb{R}^p$ such that the limit is nonzero.

**Specification of $B$ and $c$**   From the expression of $\boldsymbol{x}^\star$, we have

$$\boldsymbol{B}\boldsymbol{x}^\star + \boldsymbol{c} = \boldsymbol{c} - \boldsymbol{B}\boldsymbol{U}_2(\boldsymbol{U}_2^\top \boldsymbol{B}\boldsymbol{U}_2)^{-1}\boldsymbol{U}_2^\top \boldsymbol{c} = (\mathbf{I}_p - \boldsymbol{B}\boldsymbol{U}_2(\boldsymbol{U}_2^\top \boldsymbol{B}\boldsymbol{U}_2)^{-1}\boldsymbol{U}_2^\top)\boldsymbol{c}.$$

Define $\widetilde{\boldsymbol{B}} := \mathbf{I}_p - \boldsymbol{B}\boldsymbol{U}_2(\boldsymbol{U}_2^\top \boldsymbol{B}\boldsymbol{U}_2)^{-1}\boldsymbol{U}_2^\top$ for short. We examine the column space of $\widetilde{\boldsymbol{B}}$, which is denoted by $\mathcal{R}(\widetilde{\boldsymbol{B}})$. We can easily find $\boldsymbol{U}_2^\top \widetilde{\boldsymbol{B}} = \boldsymbol{0}_{(p-r) \times r}$. Thus $\mathcal{R}(\widetilde{\boldsymbol{B}}) \subseteq \mathcal{R}(\boldsymbol{U}_1)$. On the other hand, we have $\widetilde{\boldsymbol{B}}\boldsymbol{U}_1 = \boldsymbol{U}_1$, which implies $\mathrm{rank}(\widetilde{\boldsymbol{B}}) \ge \mathrm{rank}(\boldsymbol{U}_1)$. As a result, $\mathcal{R}(\widetilde{\boldsymbol{B}}) = \mathcal{R}(\boldsymbol{U}_1)$. Then for any $\boldsymbol{z} \in \mathbb{R}^r$, there exists a $\boldsymbol{c} \in \mathbb{R}^p$ such that $\widetilde{\boldsymbol{B}}\boldsymbol{c} = \boldsymbol{U}_1\boldsymbol{z}$. It suffices to prove that there exists a positive definite matrix $\boldsymbol{B} \in \mathbb{R}^{p \times p}$ and a vector $\boldsymbol{z} \in \mathbb{R}^r$ such that $\boldsymbol{P}^\top \mathcal{P}_{\boldsymbol{A}^\perp}\boldsymbol{B}\boldsymbol{P}(p_0\mathbf{I}_p + \widetilde{\boldsymbol{D}}_{B_1})^{-1}\boldsymbol{P}^\top \mathcal{P}_{\boldsymbol{A}}\boldsymbol{U}_1\boldsymbol{z}$ is nonzero.

Since $\boldsymbol{P} = \boldsymbol{U}\boldsymbol{Q}$, $\mathcal{P}_{\boldsymbol{A}} = \boldsymbol{U}_1\boldsymbol{U}_1^\top$, $\mathcal{P}_{\boldsymbol{A}^\perp} = \boldsymbol{U}_2\boldsymbol{U}_2^\top$, $\boldsymbol{Q} = \begin{bmatrix} \boldsymbol{Q}_1 & \boldsymbol{0}_{r \times (n-r)} \\ \boldsymbol{0}_{(n-r) \times r} & \boldsymbol{Q}_2 \end{bmatrix}$, $\widetilde{\boldsymbol{D}}_{B_1} = \begin{bmatrix} \boldsymbol{D}_{B_1} & \boldsymbol{0}_{r \times (n-r)} \\ \boldsymbol{0}_{(n-r) \times r} & \boldsymbol{0}_{n-r} \end{bmatrix}$ and $\boldsymbol{B}_1 = \boldsymbol{Q}_1\boldsymbol{D}_{B_1}\boldsymbol{Q}_1^\top$, we have

$$\boldsymbol{P}^\top \mathcal{P}_{\boldsymbol{A}^\perp}\boldsymbol{B}\boldsymbol{P}(p_0\mathbf{I}_p + \widetilde{\boldsymbol{D}}_{B_1})^{-1}\boldsymbol{P}^\top \mathcal{P}_{\boldsymbol{A}}\boldsymbol{U}_1\boldsymbol{z}$$
$$= \boldsymbol{Q}^\top \boldsymbol{U}^\top \boldsymbol{U}_2\boldsymbol{U}_2^\top \boldsymbol{B}\boldsymbol{U}\boldsymbol{Q}(p_0\mathbf{I}_p + \widetilde{\boldsymbol{D}}_{B_1})^{-1}\boldsymbol{Q}^\top \boldsymbol{U}^\top \boldsymbol{U}_1\boldsymbol{z}$$
$$= \boldsymbol{Q}^\top \begin{bmatrix} \boldsymbol{0} \\ \boldsymbol{U}_2^\top \end{bmatrix} \boldsymbol{B}[\boldsymbol{U}_1 \ \boldsymbol{U}_2] \begin{bmatrix} (p_0\mathbf{I}_r + \boldsymbol{B}_1)^{-1} & \boldsymbol{0}_{r \times (p-r)} \\ \boldsymbol{0}_{(p-r) \times r} & \frac{1}{p_0}\mathbf{I}_{p-r} \end{bmatrix} \begin{bmatrix} \boldsymbol{z} \\ \boldsymbol{0} \end{bmatrix}$$
$$= \boldsymbol{Q}^\top \begin{bmatrix} \boldsymbol{0} \\ \boldsymbol{U}_2^\top \boldsymbol{B}\boldsymbol{U}_1(p_0\mathbf{I}_p + \widetilde{\boldsymbol{D}}_{B_1})^{-1}\boldsymbol{z} \end{bmatrix}.$$

Then it suffices to prove that there exist a positive matrix $\boldsymbol{B} \in \mathbb{R}^{p \times p}$ such that $\boldsymbol{U}_2^\top \boldsymbol{B}\boldsymbol{U}_1$ is nonzero. Suppose that $\boldsymbol{U}_1 = (\boldsymbol{p}_1, \boldsymbol{p}_2, \ldots, \boldsymbol{p}_r) = (p_{ki})_{p \times r}$ and $\boldsymbol{U}_2 = (\boldsymbol{q}_1, \boldsymbol{q}_2, \ldots, \boldsymbol{q}_{p-r}) = (q_{kj})_{p \times (p-r)}$. Then the column vectors of $\boldsymbol{U}_1$ and $\boldsymbol{U}_2$ form an orthonormal basis of $\mathbb{R}^p$.

If there exist $i, j$ and $k_0$ such that $p_{k_0 i}q_{k_0 j} \ne 0$, Then we can take $\boldsymbol{B}$ as a diagonal matrix $\mathbf{I}_p + \boldsymbol{E}_{k_0 k_0}$ where $\boldsymbol{E}_{ij}$ is the $p \times p$ matrix with $(i, j)$ entry equal to 1 and others equal to 0. The $(j, i)$ entry of $\boldsymbol{U}_2^\top \boldsymbol{B}\boldsymbol{U}_1$ is $\sum_{k=1}^{p} p_{ki}q_{kj} + p_{k_0 i}q_{k_0 j} = p_{k_0 i}q_{k_0 j} \ne 0$. And one can check $\boldsymbol{B} \succeq \mathbf{I}_p$.

Otherwise, there must exist $i$, $j$, $k_0$ and $l_0$ such that $p_{k_0 i} q_{l_0 j} \neq 0$ and $k_0 \neq l_0$. Since in this case $p_{ki} q_{kj} = 0$ for any $k$, then we have $q_{k_0 j} = p_{l_0 i} = 0$. We take $\boldsymbol{B} = 2\mathbf{I}_p + \boldsymbol{E}_{k_0 l_0} + \boldsymbol{E}_{l_0 k_0}$. Then the $(j, i)$ entry of $\boldsymbol{U}_2^\top \boldsymbol{B} \boldsymbol{U}_1$ is $2\sum_{k=1}^p p_{ki} q_{kj} + p_{k_0 i} q_{l_0 j} + p_{l_0 i} q_{k_0 j} = p_{k_0 i} q_{l_0 j} \neq 0$. And one can check $\boldsymbol{B} \succeq \mathbf{I}_p$.

As a result, there always exists $\boldsymbol{B}$ and $\boldsymbol{c}$ such that the limit of $\|\mathbb{E}\tilde{\boldsymbol{u}}_n\|$ is nonzero. This implies $\mathbb{E}\|\boldsymbol{u}_n - \boldsymbol{x}^\star\|^2 \neq o(1)$, which induces a contradiction.

In the latter case, for any $\boldsymbol{e}_i \in \mathbb{R}^p$, either $\boldsymbol{e}_i^\top \boldsymbol{U}_1$ or $\boldsymbol{e}_i^\top \boldsymbol{U}_2$ is zero. Note that $\boldsymbol{U}_1$ and $\boldsymbol{U}_2$ are of full column rank. Then we have $\boldsymbol{U}_1 = \sum_{i \in I_1} \boldsymbol{e}_i \tilde{\boldsymbol{p}}_i^\top$ and $\boldsymbol{U}_2 = \sum_{j \in I_2} \boldsymbol{e}_j \tilde{\boldsymbol{q}}_j^\top$ where $|I_1| = r$, $|I_2| = p - r$, $I_1 \cup I_2 = \{1, 2, \ldots, p\}$, $I_1 \cap I_2 = \varnothing$, $\tilde{\boldsymbol{p}}_i$ ($i \in I_1$) are orthonormal basis of $\mathbb{R}^r$ and $\tilde{\boldsymbol{q}}_j$ ($j \in I_2$) are orthonormal basis of $\mathbb{R}^{p-r}$. As a consequence, $\mathcal{P}_{\boldsymbol{A}} = \boldsymbol{U}_1 \boldsymbol{U}_1^\top = \sum_{i \in I_1} \boldsymbol{e}_i \boldsymbol{e}_i^\top$ and $\mathcal{P}_{\boldsymbol{A}^\perp} = \boldsymbol{U}_2 \boldsymbol{U}_2^\top = \sum_{j \in I_2} \boldsymbol{e}_j \boldsymbol{e}_j^\top$. This implies that if $\mathcal{P}_{\boldsymbol{A}}$ is not of this form there must exist $i, j, k_0$ such that $p_{k_0 i} q_{k_0 j} \neq 0$. Then we can choose $\boldsymbol{B}$ as a diagonal matrix such that $\boldsymbol{B} \succeq \mathbf{I}_p$. $\qquad\square$

### B.4 Proof of Lemmas 1, 2 and 3

*Proof.* (**Proof of Lemma 1**)

Define $a_T = \sum_{n=1}^T r_n^p \prod_{s=1}^n \frac{1}{1-r_s}$ and $b_T = r_T^{p-1} \prod_{s=1}^T \frac{1}{1-r_s}$. We first prove that $b_{T+1} > b_T$ for sufficiently large $T$ and $\lim_{T \to \infty} b_T = \infty$. Since

$$
\begin{aligned}
\frac{b_{T+1}}{b_T} &= \left(\frac{r_{T+1}}{r_T}\right)^{p-1} \cdot \frac{1}{1 - r_{T+1}} \\
&= \frac{1}{(1 + o(r_T))^{p-1}} \cdot (1 + r_{T+1} + o(r_{T+1})) \\
&= (1 + o(r_{T+1}))(1 + r_{T+1} + o(r_{T+1})) \\
&= 1 + r_{T+1} + o(r_{T+1}),
\end{aligned}
$$

then we have $b_{T+1} > b_T$ for sufficiently large $T$. Besides, $\frac{r_n}{r_{n+1}} - 1 = o(r_n)$ implies $\frac{1}{r_n} - \frac{1}{r_{n+1}} = o(1)$. By Stolz–Cesàro theorem, we have $\lim_{n \to \infty} \frac{1}{nr_n} = 0$ and $\lim_{n \to \infty} \frac{\sum_{s=1}^n 1/s}{\sum_{s=1}^n r_s} = 0$. As a consequence,

$$
\begin{aligned}
b_T &\geq r_T^{p-1} \exp\left(\sum_{s=1}^T r_s\right) \\
&= r_T^{p-1} \exp\left(\frac{\sum_{s=1}^T r_s}{\sum_{s=1}^T 1/s} \sum_{s=1}^T 1/s\right) \\
&\geq r_T^{p-1} \exp\left(\frac{\sum_{s=1}^T r_s}{\sum_{s=1}^T 1/s} \log T\right) \\
&= (Tr_T)^{p-1} \exp\left[\left(\frac{\sum_{s=1}^T r_s}{\sum_{s=1}^T 1/s} - p + 1\right) \log T\right].
\end{aligned}
$$

Thus $\lim_{T \to \infty} b_T = \infty$. Now we use Stolz–Cesàro theorem to prove $\lim_{T \to \infty} \frac{a_T}{b_T} = 1$. With the definition of $a_T$ and $b_T$, we have

$$
a_{T+1} - a_T = r_{T+1}^p \prod_{s=1}^{T+1} \frac{1}{1 - r_s}
$$

and

$$
\begin{aligned}
b_{T+1} - b_T &= r_{T+1}^{p-1} \prod_{s=1}^{T+1} \frac{1}{1 - r_s} - r_T^{p-1} \prod_{s=1}^T \frac{1}{1 - r_s} \\
&= (r_{T+1}^{p-1} - r_T^{p-1}) \prod_{s=1}^{T+1} \frac{1}{1 - r_s} + r_T^{p-1} r_{T+1} \prod_{s=1}^{T+1} \frac{1}{1 - r_s}.
\end{aligned}
$$

It follows that

$$
\begin{aligned}
\frac{a_{T+1} - a_T}{b_{T+1} - b_T} &= \frac{r_{T+1}^p}{r_{T+1}^{p-1} - r_T^{p-1} + r_T^{p-1} r_{T+1}} \\
&= \frac{r_{T+1}}{1 - (r_T/r_{T+1})^{p-1} + r_{T+1}(r_T/r_{T+1})^{p-1}} \\
&= \frac{r_{T+1}}{1 - (1 + o(r_T))^{p-1} + r_{T+1}(1 + o(1))} \\
&= \frac{r_{T+1}}{o(r_T) + r_{T+1}(1 + o(1))} \\
&= \frac{1}{1 + o(1)},
\end{aligned}
$$

which implies $\lim_{T\to\infty} \frac{a_{T+1}-a_T}{b_{T+1}-b_T} = 1$. By Stolz–Cesàro theorem, we obtain what we want. $\qquad\square$

*Proof.* (**Proof of Lemma 2**)

Define $a_T = \sum_{n=1}^T r_t^p \prod_{s=1}^n \frac{1}{1-r_s}$ and $b_T = r_T^{p-1} \prod_{s=1}^T \frac{1}{1-r_s}$. We first prove that $b_{T+1} > b_T$ for sufficiently large $T$ and $\lim_{T\to\infty} b_T = \infty$. Since

$$
\begin{aligned}
\frac{b_{T+1}}{b_T} &= \left(\frac{r_{T+1}}{r_T}\right)^{p-1} \cdot \frac{1}{1 - r_{T+1}} \\
&= \frac{1}{(1 + ar_T + o(r_T))^{p-1}} (1 + r_{T+1} + o(r_{T+1})) \\
&= (1 - a(p-1)r_T + o(r_T)) \left(1 + \frac{r_T}{1 + ar_T + o(r_T)} + o(r_T)\right) \\
&= (1 - a(p-1)r_T + o(r_T)) (1 + r_T + o(r_T)) \\
&= 1 + [1 - a(p-1)]r_T + o(r_T),
\end{aligned}
$$

then we have $b_{T+1} > b_T$ for sufficiently large $T$. Besides, $\frac{r_n}{r_{n+1}} - 1 = ar_n + o(r_n)$ implies $\frac{1}{r_n} - \frac{1}{r_{n+1}} = a + o(1)$. By Stolz–Cesàro theorem, we have $\lim_{t\to\infty} \frac{1}{nr_n} = a$ and $\lim_{n\to\infty} \frac{\sum_{s=1}^n 1/s}{\sum_{s=1}^n r_s} = a$. As a consequence,

$$
\begin{aligned}
b_T &\geq r_T^{p-1} \exp\left(\sum_{s=1}^T r_s\right) \\
&= r_T^{p-1} \exp\left(\frac{\sum_{s=1}^T r_s}{\sum_{s=1}^T 1/s} \sum_{s=1}^T 1/s\right) \\
&\geq r_T^{p-1} \exp\left(\frac{\sum_{s=1}^T r_s}{\sum_{s=1}^T 1/s} \log T\right) \\
&= (Tr_T)^{p-1} \exp\left[\left(\frac{\sum_{s=1}^T r_s}{\sum_{s=1}^T 1/s} - p + 1\right) \log T\right] \\
&= (1/a + o(1))^{p-1} \exp\left[(1/a + o(1) - p + 1) \log T\right].
\end{aligned}
$$

Thus $\lim_{T\to\infty} b_T = \infty$.

Now we use Stolz–Cesàro theorem to prove $\lim_{T\to\infty} \frac{a_T}{b_T} = \frac{1}{1-a(p-1)}$. With the definition of $a_T$ and $b_T$, we have

$$
a_{T+1} - a_T = r_{T+1}^p \prod_{s=1}^{T+1} \frac{1}{1 - r_s}
$$

and

$$
b_{T+1} - b_T = r_{T+1}^{p-1} \prod_{s=1}^{T+1} \frac{1}{1 - r_s} - r_T^{p-1} \prod_{s=1}^T \frac{1}{1 - r_s}
$$

$$= (r_{T+1}^{p-1} - r_T^{p-1}) \prod_{s=1}^{T+1} \frac{1}{1-r_s} + r_T^{p-1} r_{T+1} \prod_{s=1}^{T+1} \frac{1}{1-r_s}.$$

It follows that

$$\begin{aligned}
\frac{a_{T+1} - a_T}{b_{T+1} - b_T} &= \frac{r_{T+1}^p}{r_{T+1}^{p-1} - r_T^{p-1} + r_T^{p-1} r_{T+1}} \\
&= \frac{r_{T+1}}{1 - (r_T/r_{T+1})^{p-1} + r_{T+1}(r_T/r_{T+1})^{p-1}} \\
&= \frac{r_{T+1}}{1 - (1 + ar_T + o(r_T))^{p-1} + r_{T+1}(1 + ar_T + o(r_T))^{p-1}} \\
&= \frac{r_{T+1}}{1 - 1 - a(p-1)r_T + o(r_T) + r_{T+1}(1 + o(1))} \\
&= \frac{1}{-a(p-1)r_T/r_{T+1} + o(1) + 1 + o(1)} \\
&= \frac{1}{1 - a(p-1) + o(1)},
\end{aligned}$$

which implies $\lim_{T \to \infty} \frac{a_{T+1} - a_T}{b_{T+1} - b_T} = \frac{1}{1 - a(p-1)}$. By Stolz–Cesàro theorem, we obtain what we want. $\qquad\square$

### Proof. (**Proof of Lemma 3**)

Suppose that $s_n = o(1)$ does not hold. Then for any positive number $\varepsilon > 0$, there exists a sequence of positive integers $\{n_i\}$ that increases to $\infty$ such that $s_{n_i} \geq \varepsilon$. From the recursion of $s_n$, there exists a positive integer $T$ such that

$$s_{n+1} \leq (1 - r_n)s_n + \frac{\varepsilon}{2} r_n \tag{27}$$

for any $n \geq T$. For $n_i > T$, we have

$$\varepsilon \leq s_{n_i} \leq (1 - r_{n_i-1})s_{n_i-1} + \frac{\varepsilon}{2} r_{n_i-1} \leq (1 - r_{n_i-1})s_{n_i-1} + \varepsilon r_{n_i-1}.$$

It follows that $s_{n_i-1} \geq \varepsilon$. Since $n_i$ increases to $\infty$, we have $s_n \geq \varepsilon$ for any $n \geq T$ by induction. For any $T_1 > T$, summing (27) from $T$ to $T_1 - 1$, we have

$$\sum_{n=T+1}^{T_1} s_n \leq \sum_{n=T}^{T_1-1} s_n - \sum_{n=T}^{T_1-1} s_n r_n + \frac{\varepsilon}{2} \sum_{n=n_T}^{T_1-1} r_n.$$

Rearranging the terms yields

$$s_T \geq s_{T_1} + \sum_{n=T}^{T_1-1} s_n r_n - \frac{\varepsilon}{2} \sum_{n=T}^{T_1-1} r_n \geq s_{T_1} + \frac{\varepsilon}{2} \sum_{n=T}^{T_1-1} r_n.$$

From the proofs of Lemmas 1 and 2, we have $\lim_{n \to \infty} \frac{\sum_{s=1}^n 1/s}{\sum_{s=1}^n r_s} = a$. Thus $\lim_{T_1 \to \infty} \sum_{n=T}^{T_1} r_n = \infty$, which induces a contradiction. As a consequence, we have $s_n = o(1)$. $\qquad\square$

### B.5 Proof of Lemmas 4 and 5

### Proof. (**Proof of Lemma 4**)

From the update rule and the linearity of $\mathcal{P}_{\boldsymbol{A}^\perp}$, we have

$$\begin{aligned}
\mathbb{E}\|\boldsymbol{u}_{n+1} - \boldsymbol{x}^\star\|^2 &= \mathbb{E}\|\mathcal{P}_{\boldsymbol{A}^\perp}(\boldsymbol{x}_n - \eta_n \nabla f(\boldsymbol{x}_n) + \eta_n \xi_n) - \boldsymbol{x}^\star\|^2 \\
&= \mathbb{E}\|\boldsymbol{u}_n - \boldsymbol{x}^\star - \eta_n \mathcal{P}_{\boldsymbol{A}^\perp}(\nabla f(\boldsymbol{x}_n) - \nabla f(\boldsymbol{x}^\star)) + \eta_n \mathcal{P}_{\boldsymbol{A}^\perp} \xi_n\|^2 \\
&= \mathbb{E}\|\boldsymbol{u}_n - \boldsymbol{x}^\star\|^2 + \eta_n^2 \mathbb{E}\|\mathcal{P}_{\boldsymbol{A}^\perp}(\nabla f(\boldsymbol{x}_n) - \nabla f(\boldsymbol{x}^\star))\|^2 \\
&\quad - 2\eta_n \mathbb{E}\langle \boldsymbol{u}_n - \boldsymbol{x}^\star, \mathcal{P}_{\boldsymbol{A}^\perp}(\nabla f(\boldsymbol{x}_n) - \nabla f(\boldsymbol{x}^\star))\rangle + \eta_n^2 \Sigma_n^{(1)}, \tag{28}
\end{aligned}$$

where the last equality is due to that $\{\xi_n\}$ is a m.d.s. and $\Sigma_n^{(1)} := \mathbb{E}\|\mathcal{P}_{\boldsymbol{A}^\perp}\xi_n\|^2$.

For the second term of (28), we have

$$\|\mathcal{P}_{\boldsymbol{A}^\perp}(\nabla f(\boldsymbol{x}_n) - \nabla f(\boldsymbol{x}^\star))\|^2 = \|\mathcal{P}_{\boldsymbol{A}^\perp}(\nabla f(\boldsymbol{x}_n) - \nabla f(\boldsymbol{u}_n)) + \mathcal{P}_{\boldsymbol{A}^\perp}(\nabla f(\boldsymbol{u}_n) - \nabla f(\boldsymbol{x}^\star))\|^2$$
$$\overset{(a)}{\leq} 2\|\mathcal{P}_{\boldsymbol{A}^\perp}(\nabla f(\boldsymbol{x}_n) - \nabla f(\boldsymbol{u}_n))\|^2 + 2\|\mathcal{P}_{\boldsymbol{A}^\perp}(\nabla f(\boldsymbol{u}_n) - \nabla f(\boldsymbol{x}^\star))\|^2$$
$$\overset{(b)}{\leq} 2L^2\|\boldsymbol{v}_n\|^2 + 2L^2\|\boldsymbol{u}_n - \boldsymbol{x}^\star\|^2,$$

where (a) is by Proposition 3 and (b) is due to non-expansiveness of $\mathcal{P}_{\boldsymbol{A}^\perp}$ and smoothness of $f$. For the third term of (28), we have

$$-\langle \boldsymbol{u}_n - \boldsymbol{x}^\star, \mathcal{P}_{\boldsymbol{A}^\perp}(\nabla f(\boldsymbol{x}_n) - \nabla f(\boldsymbol{x}^\star))\rangle$$
$$\overset{(a)}{=} -\langle \boldsymbol{u}_n - \boldsymbol{x}^\star, \nabla f(\boldsymbol{x}_n) - \nabla f(\boldsymbol{x}^\star)\rangle$$
$$= -\langle \boldsymbol{u}_n - \boldsymbol{x}^\star, \nabla f(\boldsymbol{x}_n) - \nabla f(\boldsymbol{u}_n)\rangle - \langle \boldsymbol{u}_n - \boldsymbol{x}^\star, \nabla f(\boldsymbol{u}_n) - \nabla f(\boldsymbol{x}^\star)\rangle$$
$$\overset{(b)}{\leq} \frac{\mu}{4}\|\boldsymbol{u}_n - \boldsymbol{x}^\star\|^2 + \frac{1}{\mu}\|\nabla f(\boldsymbol{x}_n) - \nabla f(\boldsymbol{u}_n)\|^2 - \mu\|\boldsymbol{u}_n - \boldsymbol{x}^\star\|^2$$
$$\overset{(c)}{\leq} -\frac{3\mu}{4}\|\boldsymbol{u}_n - \boldsymbol{x}^\star\|^2 + \frac{L^2}{\mu}\|\boldsymbol{v}_n\|^2,$$

where (a) follows from the orthogonality between $\mathcal{P}_{\boldsymbol{A}}$ and $\mathcal{P}_{\boldsymbol{A}^\perp}$, (b) is by Propositions 2 and 3 and (c) is due to the smoothness of $f$. For the last term of (28), we first show that $|\Sigma_n^{(1)} - \Sigma_\star^{(1)}| \leq dL\mathbb{E}\|\boldsymbol{x}_n - \boldsymbol{x}^\star\|$. From the definition of $\Sigma_n^{(1)}$ and $\Sigma_\star^{(1)}$, we have

$$|\Sigma_n^{(1)} - \Sigma_\star^{(1)}| = \left|\mathbb{E}\operatorname{trace}\left(\mathcal{P}_{\boldsymbol{A}^\perp}(\xi_n\xi_n^\top - \xi^\star(\xi^\star)^\top)\mathcal{P}_{\boldsymbol{A}^\perp}\right)\right|$$
$$= \left|\operatorname{trace}\left(\mathcal{P}_{\boldsymbol{A}^\perp}(\mathbb{E}\xi_n\xi_n^\top - \mathbb{E}\xi^\star(\xi^\star)^\top)\mathcal{P}_{\boldsymbol{A}^\perp}\right)\right|$$
$$\leq d\|\mathcal{P}_{\boldsymbol{A}^\perp}\mathbb{E}(\Sigma(\boldsymbol{x}_n) - \Sigma(\boldsymbol{x}^\star))\mathcal{P}_{\boldsymbol{A}^\perp}\|$$
$$\leq dL\mathbb{E}\|\boldsymbol{x}_n - \boldsymbol{x}^\star\|,$$

where the last inequality is due to the non-expansiveness of $\mathcal{P}_{\boldsymbol{A}^\perp}$ and Assumption 3. It follows that

$$\Sigma_n^{(1)} \leq \Sigma_\star^{(1)} + |\Sigma_n^{(1)} - \Sigma_\star^{(1)}|$$
$$\leq \Sigma_\star^{(1)} + dL\mathbb{E}\|\boldsymbol{x}_n - \boldsymbol{x}^\star\|$$
$$\leq \Sigma_\star^{(1)} + dL\mathbb{E}(\|\boldsymbol{u}_n - \boldsymbol{x}^\star\| + \|\boldsymbol{v}_n\|)$$
$$\overset{(a)}{\leq} \Sigma_\star^{(1)} + dL\left[\frac{\Sigma_\star^{(1)}}{dL} + \frac{dL}{2\Sigma_\star^{(1)}}\mathbb{E}(\|\boldsymbol{u}_n - \boldsymbol{x}^\star\|^2 + \|\boldsymbol{v}_n\|^2)\right]$$
$$= 2\Sigma_\star^{(1)} + \frac{d^2L^2}{2\Sigma_\star^{(1)}}\mathbb{E}\|\boldsymbol{u}_n - \boldsymbol{x}^\star\|^2 + \frac{d^2L^2}{2\Sigma_\star^{(1)}}\|\boldsymbol{v}_n\|^2,$$

where (a) follows from Proposition 3.

By substituting these inequalities, we obtain

$$\mathbb{E}\|\boldsymbol{u}_{n+1} - \boldsymbol{x}^\star\|^2 \leq \left(1 - \frac{3\mu}{2}\eta_n + 2L^2\eta_n^2 + \frac{d^2L^2}{2\Sigma_\star^{(1)}}\eta_n^2\right)\mathbb{E}\|\boldsymbol{u}_n - \boldsymbol{x}^\star\|^2$$
$$+ \left(\frac{2L^2}{\mu}\eta_n + 2L^2\eta_n^2 + \frac{d^2L^2}{2\Sigma_\star^{(1)}}\eta_n^2\right)\mathbb{E}\|\boldsymbol{v}_n\|^2 + 2\eta_n^2\Sigma_n^{(1)}$$
$$\overset{(a)}{\leq} (1 - \mu\eta_n)\mathbb{E}\|\boldsymbol{u}_n - \boldsymbol{x}^\star\|^2 + \frac{4L^2}{\mu}\eta_n\mathbb{E}\|\boldsymbol{v}_n\|^2 + 2\eta_n^2\Sigma_n^{(1)},$$

where (a) holds if $n$ is large enough. $\qquad\square$

*Proof.* (**Proof of Lemma 5**)

From the update rule and the linearity of $\mathcal{P}_A$, we have

$$
\begin{aligned}
\mathbb{E}\left\|v_{n+1}\right\|^2 &= (1-p_n)\mathbb{E}\left\|\mathcal{P}_A(x_n - \eta_n \nabla f(x_n)\eta_n \xi_n)\right\|^2 \\
&= (1-p_n)\mathbb{E}\left\|v_n - \eta_n \mathcal{P}_A \nabla f(x_n) + \eta_n \mathcal{P}_A \xi_n\right\|^2 \\
&= (1-p_n)\mathbb{E}\left\|v_n\right\|^2 + (1-p_n)\eta_n^2 \mathbb{E}\left\|\mathcal{P}_A \nabla f(x_n)\right\|^2 \\
&\quad - 2(1-p_n)\eta_n \mathbb{E}\left\langle v_n, \mathcal{P}_A \nabla f(x_n)\right\rangle + (1-p_n)\eta_n^2 \Sigma_n^{(2)},
\end{aligned}
\tag{29}
$$

where the last equality is due to that $\{\xi_n\}$ is a m.d.s. and $\Sigma_n^{(2)} := \mathbb{E}\left\|\mathcal{P}_A \xi_n\right\|^2$.
For the second term of (29), we have

$$
\begin{aligned}
\left\|\mathcal{P}_A \nabla f(x_n)\right\|^2 &= \left\|\mathcal{P}_A(\nabla f(x_n) - \nabla f(u_n) + \nabla f(u_n) - \nabla f(x^\star) + \nabla f(x^\star))\right\|^2 \\
&\overset{(a)}{\leq} 3\left\|\mathcal{P}_A(\nabla f(x_n) - \nabla f(u_n))\right\|^2 + 3\left\|\mathcal{P}_A(\nabla f(u_n) - \nabla f(x^\star))\right\|^2 + 3\left\|\mathcal{P}_A \nabla f(x^\star)\right\|^2 \\
&\overset{(b)}{\leq} 3L^2\left\|v_n\right\|^2 + 3L^2\left\|u_n - x^\star\right\|^2 + 3\left\|\nabla f(x^\star)\right\|^2,
\end{aligned}
$$

where (a) is by Proposition 3 and (b) follows from non-expansiveness of $\mathcal{P}_A$ and smoothness of $f$.
For the third term of (29), we have

$$
\begin{aligned}
&-\left\langle v_n, \mathcal{P}_A \nabla f(x_n)\right\rangle \\
&\overset{(a)}{=} -\left\langle v_n, \nabla f(x_n)\right\rangle \\
&= -\left\langle v_n, \nabla f(x_n) - \nabla f(u_n)\right\rangle - \left\langle v_n, \nabla f(u_n) - \nabla f(x^\star)\right\rangle - \left\langle v_n, \nabla f(x^\star)\right\rangle \\
&\overset{(b)}{\leq} -\mu\left\|v_n\right\|^2 + \frac{p_n}{8\eta_n}\left\|v_n\right\|^2 + \frac{2\eta_n}{p_n}\left\|\nabla f(u_n) - \nabla f(x^\star)\right\|^2 + \frac{p_n}{8\eta_n}\left\|v_n\right\|^2 + \frac{2\eta_n}{p_n}\left\|\nabla f(x^\star)\right\|^2 \\
&\overset{(c)}{\leq} -\mu\left\|v_n\right\|^2 + \frac{p_n}{4\eta_n}\left\|v_n\right\|^2 + \frac{2L^2\eta_n}{p_n}\left\|u_n - x^\star\right\|^2 + \frac{2\eta_n}{p_n}\left\|\nabla f(x^\star)\right\|^2,
\end{aligned}
$$

where (a) follows from the orthogonality between $\mathcal{P}_A$ and $\mathcal{P}_{A^\perp}$, (b) is by Propositions 2 and 3 and (c) is due to the smoothness of $f$. For the last term of (29), we can obtain

$$
\Sigma_n^{(2)} \leq 2\Sigma_\star^{(2)} + \frac{d^2 L^2}{2\Sigma_\star^{(2)}}\mathbb{E}\left\|u_n - x^\star\right\|^2 + \frac{d^2 L^2}{2\Sigma_\star^{(2)}}\left\|v_n\right\|^2
$$

by following similar procedure in the proof of Lemma 4. By substituting these inequalities, we obtain

$$
\begin{aligned}
\mathbb{E}\left\|v_{n+1}\right\|^2 &\leq (1-p_n)\left(1 - 2\mu\eta_n + 3L^2\eta_n^2 + \frac{p_n}{2} + \frac{d^2 L^2}{2\Sigma_\star^{(2)}}\eta_n^2\right)\mathbb{E}\left\|v_n\right\|^2 \\
&\quad + \left(\frac{4L^2\eta_n^2}{p_n} + 3L^2\eta_n^2 + \frac{d^2 L^2}{2\Sigma_\star^{(2)}}\eta_n^2\right)\mathbb{E}\left\|u_n - x^\star\right\|^2 \\
&\quad + \left(\frac{4\eta_n^2}{p_n} + 3\eta_n^2\right)\left\|\nabla f(x^\star)\right\|^2 + 2\eta_n^2 \Sigma_n^{(2)} \\
&\overset{(a)}{\leq} \left(1 - \frac{p_n}{2}\right)\mathbb{E}\left\|v_n\right\|^2 + \frac{7L^2\eta_n^2}{p_n}\mathbb{E}\left\|u_n - x^\star\right\|^2 + \frac{7\eta_n^2}{p_n}\left\|\nabla f(x^\star)\right\|^2 + 2\eta_n^2 \Sigma_n^{(2)},
\end{aligned}
$$

where (a) holds if $n$ is large enough. $\qquad\square$

## C    Proof of Section 3.2

### C.1    Proof of Case 1

We can deduce the recursive relationship of $\check{\boldsymbol{u}}_n$ by the definition of $\boldsymbol{u}_n$ and the update rule (2).

$$
\begin{aligned}
\check{\boldsymbol{u}}_{n+1} &= \mathcal{P}_{\boldsymbol{A}^\perp} \frac{\boldsymbol{x}_{n+\frac{1}{2}} - \boldsymbol{x}^\star}{\sqrt{\eta_n}} \\
&= \mathcal{P}_{\boldsymbol{A}^\perp} \frac{1}{\sqrt{\eta_n}} (\boldsymbol{x}_n - \eta_n \nabla f(\boldsymbol{x}_n) + \eta_n \xi_n - \boldsymbol{x}^\star) \\
&= \frac{\sqrt{\eta_{n-1}}}{\sqrt{\eta_n}} \check{\boldsymbol{u}}_n - \sqrt{\eta_n} \mathcal{P}_{\boldsymbol{A}^\perp} \left\{ (\nabla f(\boldsymbol{x}_n) - \nabla f(\boldsymbol{u}_n)) + (\nabla f(\boldsymbol{u}_n) - \nabla f(\boldsymbol{x}^\star)) \right\} + \sqrt{\eta_n} \mathcal{P}_{\boldsymbol{A}^\perp} \xi_n \\
&= \frac{\sqrt{\eta_{n-1}}}{\sqrt{\eta_n}} \check{\boldsymbol{u}}_n - \sqrt{\eta_n \eta_{n-1}} \mathcal{P}_{\boldsymbol{A}^\perp} \left\{ \int_0^1 \nabla^2 f(t\boldsymbol{x}^\star + (1-t)\boldsymbol{u}_n) \, dt \right\} \check{\boldsymbol{u}}_n + \mathcal{R}_n^{(1)} + \sqrt{\eta_n} \mathcal{P}_{\boldsymbol{A}^\perp} \xi_n \\
&= \check{\boldsymbol{u}}_n - \eta_n \mathcal{P}_{\boldsymbol{A}^\perp} \left( \nabla^2 f(\boldsymbol{x}^\star) - \frac{1}{2\eta_0} \mathbb{1}_{\{\alpha=1\}} \boldsymbol{I} \right) \check{\boldsymbol{u}}_n + \mathcal{R}_n^{(1)} + \mathcal{R}_n^{(2)} + \mathcal{R}_n^{(3)} + \sqrt{\eta_n} \xi_n^{(1)}
\end{aligned}
\tag{30}
$$

Where $\mathcal{R}_n^{(i)}$, $i = 1, 2, 3$ are higher order term with respect to $\eta_n$ with the form:

$$
\begin{aligned}
\mathcal{R}_n^{(1)} &= -\sqrt{\eta_n} \mathcal{P}_{\boldsymbol{A}^\perp} (\nabla f(\boldsymbol{x}_n) - \nabla f(\boldsymbol{u}_n)) \\
\mathcal{R}_n^{(2)} &= -\left( 1 - \sqrt{\frac{\eta_{n-1}}{\eta_n}} + \frac{\eta_n}{2\eta_0} \mathbb{1}_{\{\alpha=1\}} \right) \check{\boldsymbol{u}}_n + (\eta_n - \sqrt{\eta_n \eta_{n-1}}) \mathcal{P}_{\boldsymbol{A}^\perp} \nabla^2 f(\boldsymbol{x}^\star) \check{\boldsymbol{u}}_n \\
\mathcal{R}_n^{(3)} &= \sqrt{\eta_n \eta_{n-1}} \mathcal{P}_{\boldsymbol{A}^\perp} \left( \nabla^2 f(\boldsymbol{x}^\star) - \int_0^1 \nabla^2 f(t\boldsymbol{x}^\star + (1-t)\boldsymbol{u}_n) \, dt \right) \check{\boldsymbol{u}}_n
\end{aligned}
\tag{31}
$$

Where $\boldsymbol{\theta}_n$ is an entry-wise linear interpolation point from $\boldsymbol{u}_n$ to $\boldsymbol{x}^\star$ and Lemma 6 shows that $\frac{1}{\eta_n} \mathcal{R}_n^{(i)}$ are $o(1)$ in some sense.

**Lemma 6.** *When Assumptions 1, 2 and 4 hold, and let $p_t = \eta_t^\beta$ where $\beta \in [0, 1/2)$, then for any $i \in \{1, 2\}$, $\mathbb{E}\|\mathcal{R}_n^{(i)}\|^2 = o(\eta_n^2)$. For $\mathcal{R}_n^{(3)}$, we have $\mathbb{E}\|\mathcal{R}_n^{(3)}\|^2 = \mathcal{O}(\eta_n^2)$ and $\mathbb{E}\|\mathcal{R}_n^{(3)}\| = o(\eta_n)$.*

We first show the tightness of the rescaling sequence we built. Actually, we make use of a classical criterion (Theorem 7.3 in [3]) to prove this property of $\bar{\boldsymbol{u}}_t^{(n)}$.

**Proposition 6.** *The sequence $\bar{\boldsymbol{u}}^{(n)}$ is tight if these two conditions hold*

> 1. *For each positive $\eta$, there exists an $a$ and an $n_0$ such that*

$$
\mathbb{P}(\|\check{\boldsymbol{u}}_n\| \geq a) \leq \eta \quad \forall n \geq n_0
\tag{32}
$$

> 2. *For any $T > 0$, for any positive $\epsilon, \eta$, a $\delta$ exists and an integer $n_0$ exists such that:*

$$
\mathbb{P}\left( \sup_{s \in [t, t+\delta]} \left\| \bar{\boldsymbol{u}}_s^{(n)} - \bar{\boldsymbol{u}}_t^{(n)} \right\| \geq \varepsilon \right) \leq \eta\delta; \quad \forall t \in [0, T] \quad \forall n \geq n_0
\tag{33}
$$

*Proof.* (**Proof of Lemma 6**) When $\alpha < 1$,

$$
\begin{aligned}
\mathbb{E}\|\mathcal{R}_n^{(1)}\|^2 &\leq \eta_n \mathbb{E}\|\nabla f(\boldsymbol{x}_n) - \nabla f(\boldsymbol{u}_n)\|^2 \leq \eta_n L^2 \mathbb{E}\|\boldsymbol{v}_n\|^2 \precsim L^2 \eta_n \times \eta_n^{2-2\beta} = o(\eta_n^2) \\
\mathbb{E}\|\mathcal{R}_n^{(2)}\|^2 &\precsim \left(1 - \sqrt{\frac{\eta_{n-1}}{\eta_n}}\right)^2 + (\eta_n - \sqrt{\eta_n \eta_{n-1}})^2 = \left(\frac{1}{\eta_n} + \eta_n\right)(\sqrt{\eta_n} - \sqrt{\eta_{n-1}})^2 \\
&\precsim \frac{(\eta_n - \eta_{n-1})^2}{\eta_n^2} = [1 - (1 + o(\eta_n))]^2 = o(\eta_n^2) \\
\mathbb{E}\left\|\mathcal{R}_n^{(3)}\right\|^2 &\precsim \eta_n^2 \mathbb{E}\left\|\left\{\int_0^1 \nabla^2 f\left(t\boldsymbol{x}^\star + (1-t)\boldsymbol{u}_n\right) dt - \nabla^2 f(\boldsymbol{x}^\star)\right\}\check{\boldsymbol{u}}_n\right\|^2 \\
&\precsim \eta_n^2 \mathbb{E}\|\check{\boldsymbol{u}}_n\|^2 = \mathcal{O}(\eta_n^2) \\
\mathbb{E}\left\|\mathcal{R}_n^{(3)}\right\| &\precsim \eta_n \mathbb{E}\left\|\int_0^1 \nabla^2 f\left(t\boldsymbol{x}^\star + (1-t)\boldsymbol{u}_n\right) dt - \nabla^2 f(\boldsymbol{x}^\star)\right\| \cdot \|\check{\boldsymbol{u}}_n\| \\
&\precsim \eta_n \mathbb{E}\left\|\check{\boldsymbol{u}}_n\right\| \int_0^1 \left\|\nabla^2 f(t\boldsymbol{x}^\star - (1-t)\boldsymbol{u}_n) - \nabla^2 f(\boldsymbol{x}^\star)\right\| dt \\
&\precsim \eta_n \mathbb{E}\left\|\check{\boldsymbol{u}}_n\right\| \int_0^1 (1-t)\left\|\boldsymbol{u}_n - \boldsymbol{x}^\star\right\| dt \precsim \eta_n^{3/2} \mathbb{E}\left\|\check{\boldsymbol{u}}_n\right\|^2 = o(\eta_n).
\end{aligned}
\tag{34}
$$

And when $\alpha = 1$

$$
\mathbb{E}\left\|\mathcal{R}_n^{(2)}\right\|^2 \precsim \left(1 - \sqrt{1 + \frac{1}{n-1}} + \frac{1}{2n}\right)^2 + \frac{1}{n}\left(\frac{1}{\sqrt{n}} - \frac{1}{\sqrt{n-1}}\right)^2 = o\left(\frac{1}{n^2}\right).
$$

$\square$

**Lemma 7** (**Tightness of** $\bar{\boldsymbol{u}}^{(n)}$)**.** *Suppose that Assumptions 1, 2 and 4 holds, and assume that there exists a positive number $p > 2$ such that $\sup\limits_{n \geq 0} \mathbb{E}\|\xi_n\|^p < \infty$. Then the sequence of random processes $\{\bar{\boldsymbol{u}}^{(n)}\}$ is tight under the Skorokhod topology in finite interval.*

*Proof.* (**Proof of Lemma 7**)

From the construction of $\bar{\boldsymbol{u}}_t^{(n)}$ we know it is a continuous process. What remains we have to do is to verify two conditions (32) and (33).

For the first condition about initialization of the process, it is easy to check by the convergence rate result for $\boldsymbol{u}_n - \boldsymbol{x}^\star$.

For the condition (33), note that we have

$$
\begin{aligned}
\bar{\boldsymbol{u}}_s^{(n)} - \bar{\boldsymbol{u}}_t^{(n)} &= \left\{\sum_{k=N(n,t,\eta)}^{N(n,s,\eta)-1} \eta_k \boldsymbol{b}_k - \left[(t - \underline{t}_n(\eta))\boldsymbol{b}_{N(n,t,\eta)} - (s - \underline{s}_n(\eta)\boldsymbol{b}_{N(n,s,\eta)})\right]\right\} \\
&\quad + \left\{\sum_{k=N(n,t,\eta)}^{N(n,s,\eta)-1} \sqrt{\eta_k}\xi_k - \sqrt{t - \underline{t}_n(\eta)}\xi_{N(n,t,\eta)} + \sqrt{s - \underline{s}_n(\eta)}\xi_{N(n,s,\eta)}\right\} \\
&=: \mathbf{B} + \boldsymbol{\Xi}
\end{aligned}
\tag{35}
$$

From the discussion following Lemma 6, we can see that $\mathbb{E}\|\boldsymbol{b}_n\|^2$ is uniformly bounded. So

$$
\begin{aligned}
\mathbb{P}\left(\sup_{s\in[t,t+\delta]}\|\mathbf{B}\|\geq\frac{\epsilon}{2}\right) &\leq \mathbb{P}\left(\sum_{k=N(n,t,\eta)}^{N(n,s,\eta)}\eta_k\|\boldsymbol{b}_k\|\geq\frac{\epsilon}{2}\right)\\
&\leq \frac{4}{\epsilon^2}\mathbb{E}\left(\sum_{k=N(n,t,\eta)}^{N(n,s,\eta)}\eta_k\|\boldsymbol{b}_k\|\right)^2\\
&\leq \frac{4}{\epsilon^2}\left(\sum_{k=N(n,t,\eta)}^{N(n,s,\eta)}\eta_k\right)\mathbb{E}\left(\sum_{k=N(n,t,\eta)}^{N(n,s,\eta)}\eta_k\|\boldsymbol{b}_k\|^2\right)\\
&\leq \frac{4}{\epsilon^2}\sup_k\mathbb{E}\|\boldsymbol{b}_k\|^2\left(\sum_{k=N(n,t,\eta)}^{N(n,s,\eta)}\eta_k\right)^2\\
&\leq C\frac{(\delta+\eta_n)^2}{\epsilon^2}\overset{(a)}{\leq}\frac{\eta(\delta+\eta_n)}{4}\overset{(b)}{\leq}\frac{\eta\delta}{2}
\end{aligned}
\tag{36}
$$

Where (a) and (b) holds when we take $\delta+\eta_{n_0}<\frac{\eta\epsilon^2}{4C}$ and $\eta_{n_0}<\delta$.

On the other hand, thanks to the property of monotone interpolation, we have

$$
\|\boldsymbol{\Xi}\|\leq\max_{j\in\{0,1\}}\left\|\sum_{k=N(n,t,\eta)}^{N(n,s,\eta)-j}\sqrt{\eta_k}\xi_k-\sqrt{t-\underline{t}_n(\eta)}\xi_{N(n,t,\eta)}\right\|.
\tag{37}
$$

By leveraging the Doob's inequality and the assumption of bounded p-th moment of $\xi_k$, we can get

$$
\begin{aligned}
\mathbb{P}\left(\sup_{s\in[t,t+\delta]}\|\boldsymbol{\Xi}\|\geq\frac{\epsilon}{2}\right)&\leq\mathbb{P}\left(\max_{j\leq N(n,s,\eta)}\left\|\sum_{k=N(n,t,\eta)}^{j}\sqrt{\eta_k}\xi_k-\sqrt{t-\underline{t}_n(\eta)}\xi_{N(n,t,\eta)}\right\|\geq\frac{\epsilon}{2}\right)\\
&\leq\frac{2^p}{\epsilon^p}\mathbb{E}\left\|\sum_{k=N(n,t,\eta)}^{N(n,s,\eta)}\sqrt{\eta_k}\xi_k-\sqrt{t-\underline{t}_n(\eta)}\xi_{N(n,t,\eta)}\right\|^p\\
&\overset{(a)}{\leq}\frac{\mathcal{C}_p2^p}{\epsilon^p}\sum_{k=N(n,t,\eta)}^{N(n,s,\eta)}\eta_n^{p/2}\mathbb{E}\|\xi_k\|^p\\
&\leq\frac{\mathcal{C}}{\epsilon^p}\eta_n^{\frac{p}{2}-1}\sum_{k=N(n,t,\eta)}^{N(n,s,\eta)}\eta_k\\
&\overset{(b)}{\leq}\frac{\mathcal{C}}{\epsilon^p}\eta_n^{\frac{p}{2}-1}(\delta+\eta_n)\overset{(c)}{\leq}\frac{\eta(\delta+\eta_n)}{4}\leq\frac{\eta\delta}{2}
\end{aligned}
\tag{38}
$$

Where (a) holds by the Burkholder's inequality and (b), (c) hold when we choose $\eta_{n_0}^{\frac{p}{2}-1}\leq\frac{\epsilon^p\eta}{4\mathcal{C}}$ and $\eta_{n_0}\leq\delta$.

Combine (36) and (38), finally we can derive that

$$
\begin{aligned}
\mathbb{P}\left(\sup_{s\in[t,t+\delta]}\left\|\bar{\boldsymbol{u}}_s^{(n)}-\bar{\boldsymbol{u}}_t^{(n)}\right\|\geq\varepsilon\right)&\leq\mathbb{P}\left(\sup_{s\in[t,t+\delta]}\|\mathbf{B}\|\geq\frac{\epsilon}{2}\right)+\mathbb{P}\left(\sup_{s\in[t,t+\delta]}\|\boldsymbol{\Xi}\|\geq\frac{\epsilon}{2}\right)\\
&\leq\frac{\eta\delta}{2}+\frac{\eta\delta}{2}=\eta\delta
\end{aligned}
\tag{39}
$$

So far, we conclude the proof of Lemma 7. $\qquad\square$

**Lemma 8.** *Suppose Assumptions 1, 1 and 4 holds, and assume that there exists a positive number $p > 2$ such that $\sup\limits_{n \geq 0} \mathbb{E}\|\xi_n\|^p < \infty$. And suppose*

$$\mathbb{E}[\xi_t \xi_t^\top | \mathcal{F}_t] \overset{n \to \infty}{\longrightarrow} \Sigma \quad \text{in probability} \tag{40}$$

*Where $\Sigma$ is a positive definite $d \times d$-matrix. Then for any $C^2$ function $g : \mathbb{R}^d \to \mathbb{R}$, compactly supported with Lipschitz continuous second derivatives, we have*

$$\mathbb{E}[g(\check{\boldsymbol{u}}_{n+1}) - g(\check{\boldsymbol{u}}_n)|\mathcal{F}_n] = \eta_n \mathcal{L}g(\check{\boldsymbol{u}}_n) + \mathcal{R}_n^g \tag{41}$$

*Where $\frac{1}{\eta_n}\mathcal{R}_n^g \to 0$ in $L_1$ and $\mathcal{L}$ is the infinitesimal generator defined by*

$$\forall \phi \in \mathcal{C}^2(\mathbb{R}^p) \quad \mathcal{L}\phi(\boldsymbol{x}) = \left\langle -\mathcal{P}_{\boldsymbol{A}^\perp}\left(\nabla^2 f(\boldsymbol{x}^\star) - \frac{1}{2\eta_0}\mathbb{1}_{\{\alpha=1\}}\mathbf{I}_d\right)\mathcal{P}_{\boldsymbol{A}^\perp}\boldsymbol{x}, \nabla\phi\right\rangle + \frac{1}{2}\text{tr}\left(\nabla^2\phi(\boldsymbol{x})\Sigma\right) \tag{42}$$

*Proof.* (**Proof of Lemma 8**)

$\mathcal{C}$ will represent a universal constant whose value may change from line to line, for the sake of convenience. We use a Taylor expansion between $\boldsymbol{u}_n$ and $\boldsymbol{u}_{n+1}$

$$\begin{aligned}
g(\check{\boldsymbol{u}}_{n+1}) - g(\check{\boldsymbol{u}}_n) &= \langle\nabla g(\check{\boldsymbol{u}}_n), \check{\boldsymbol{u}}_{n+1} - \check{\boldsymbol{u}}_n\rangle + \frac{1}{2}(\check{\boldsymbol{u}}_{n+1} - \check{\boldsymbol{u}}_n)^\top \nabla^2 g(\check{\boldsymbol{u}}_n)(\check{\boldsymbol{u}}_{n+1} - \check{\boldsymbol{u}}_n) \\
&\quad + \underbrace{\frac{1}{2}(\check{\boldsymbol{u}}_{n+1} - \check{\boldsymbol{u}}_n)^\top \left(\nabla^2 g(\boldsymbol{\lambda}_n) - \nabla^2 g(\check{\boldsymbol{u}}_n)\right)(\check{\boldsymbol{u}}_{n+1} - \check{\boldsymbol{u}}_n)}_{\mathcal{R}_n^{(4)}}
\end{aligned} \tag{43}$$

Since $\nabla^2 g$ is Lipschitz continuous and compactly supported, $\nabla^2 g$ is also $\epsilon$-Hölder continuous for all $\epsilon \in (0,1]$. Then combine the equation (30) we can control the order of $\mathcal{R}_n^{(4)}$.

$$\begin{aligned}
\mathbb{E}\|\mathcal{R}_n^{(4)}\| &\precsim \mathbb{E}\|\check{\boldsymbol{u}}_{n+1} - \check{\boldsymbol{u}}_n\|^{2+\epsilon} \\
&\leq \mathbb{E}\left\|\eta_n \mathcal{P}_{\boldsymbol{A}^\perp}\left(\nabla^2 f(\boldsymbol{x}^\star) - \frac{1}{2\eta_0}\mathbb{1}_{\{\alpha=1\}}\mathbf{I}_d\right)\mathcal{P}_{\boldsymbol{A}^\perp}\check{\boldsymbol{u}}_n + \mathcal{R}_n^{(1)} + \mathcal{R}_n^{(2)} + \mathcal{R}_n^{(3)} + \sqrt{\eta_n}\xi_n\right\|^{2+\epsilon} \\
&\precsim \eta_n^{1+\frac{\epsilon}{2}}
\end{aligned}$$

So we deduce $\frac{1}{\eta_n}\mathcal{R}_n^{(4)} \to 0$ in $L_1$. Further, we make use of the update formula (30) again

$$\begin{aligned}
&\mathbb{E}[\langle\nabla g(\check{\boldsymbol{u}}_n), \check{\boldsymbol{u}}_{n+1} - \check{\boldsymbol{u}}_n\rangle|\mathcal{F}_n] \\
&= \mathbb{E}\left[\langle\nabla g(\check{\boldsymbol{u}}_n), -\eta_n \mathcal{P}_{\boldsymbol{A}^\perp}\left(\nabla^2 f(\boldsymbol{x}^\star) - \frac{1}{2\eta_0}\mathbb{1}_{\{\alpha=1\}}\mathbf{I}_d\right)\mathcal{P}_{\boldsymbol{A}^\perp}\check{\boldsymbol{u}}_n + \mathcal{R}_n^{(1)} + \mathcal{R}_n^{(2)} + \mathcal{R}_n^{(3)} + \sqrt{\eta_n}\xi_n\rangle|\mathcal{F}_n\right] \\
&= -\eta_n\mathbb{E}\left[\langle\nabla g(\check{\boldsymbol{u}}_n), \mathcal{P}_{\boldsymbol{A}^\perp}\left(\nabla^2 f(\boldsymbol{x}^\star) - \frac{1}{2\eta_0}\mathbb{1}_{\{\alpha=1\}}\mathbf{I}_d\right)\mathcal{P}_{\boldsymbol{A}^\perp}\check{\boldsymbol{u}}_n\rangle|\mathcal{F}_n\right] + \sum_{i=1}^{3}\mathbb{E}[\langle\nabla g(\check{\boldsymbol{u}}_n), \mathcal{R}_n^{(i)}\rangle|\mathcal{F}_n]
\end{aligned} \tag{44}$$

Note by Lemma 6, we have

$$\begin{aligned}
\mathbb{E}\left|\mathbb{E}\left[\left\langle\nabla g(\check{\boldsymbol{u}}_n), \frac{\mathcal{R}_n^{(i)}}{\eta_n}\right\rangle \Big| \mathcal{F}_n\right]\right| &\leq \mathbb{E}\left|\left\langle\nabla g(\check{\boldsymbol{u}}_n), \frac{\mathcal{R}_n^{(i)}}{\eta_n}\right\rangle\right| \\
&\precsim \mathbb{E}\left\|\frac{\mathcal{R}_n^{(i)}}{\eta_n}\right\| \leq \left(\mathbb{E}\left\|\frac{\mathcal{R}_n^{(i)}}{\eta_n}\right\|^2\right)^{\frac{1}{2}} = o(1)
\end{aligned} \tag{45}$$

And at last,

$$\frac{1}{2}\mathbb{E}\left[(\check{\boldsymbol{u}}_{n+1} - \check{\boldsymbol{u}}_n)^\top \nabla^2 g(\check{\boldsymbol{u}}_n)(\check{\boldsymbol{u}}_{n+1} - \check{\boldsymbol{u}}_n)|\mathcal{F}_n\right]$$

$$=\frac{\eta_n}{2}\mathbb{E}\left[\xi_n^\top \nabla^2 g(\check{\boldsymbol{u}}_n)\xi_n|\mathcal{F}_n\right] + \frac{\eta_n^{\frac{3}{2}}}{2}\mathbb{E}\left\langle \boldsymbol{b}_n, \nabla^2 g(\check{\boldsymbol{u}}_n)\xi_n \right\rangle \tag{46}$$

$$+\frac{\eta_n^2}{2}\mathbb{E}\langle \boldsymbol{b}_n, \nabla^2 g(\check{\boldsymbol{u}}_n)\boldsymbol{b}_n\rangle$$

Cause $g$ is compactly supported, the norm of $\nabla^2 g$ is bounded. And by Lemma 6, we can deduce $\mathbb{E}\|\boldsymbol{b}_n\|^2$ is uniformly bounded. Therefore, the last two terms of (46) are $o(\eta_n)$. Combine the above analysis, we have

$$\mathbb{E}[g(\check{\boldsymbol{u}}_{n+1}) - g(\check{\boldsymbol{u}}_n)|\mathcal{F}_n]$$
$$= -\eta_n \left\langle \nabla g(\check{\boldsymbol{u}}_n), \mathcal{P}_{\boldsymbol{A}^\perp}\left(\nabla^2 f(\boldsymbol{x}^\star) - \frac{1}{2\eta_0}\mathbb{1}_{\{\alpha=1\}}\mathbf{I}_d\right)\mathcal{P}_{\boldsymbol{A}^\perp}\check{\boldsymbol{u}}_n \right\rangle + \frac{\eta_n}{2}\left\langle \nabla^2 g(\check{\boldsymbol{u}}_n), \Sigma \right\rangle + \mathcal{R}_n^g \tag{47}$$

with $\mathbb{E}\|\mathcal{R}_n^g\| = o(\eta_n)$.

$\square$

*Proof.* (**Proof of Theorem 3.3**) The proof of this main results is divided into two steps. At first, we prove that every weak limits of sequence of random process $\{\bar{\boldsymbol{u}}^{(n)}\}$ is a solution of the martingale problem $(\mathcal{L}, \mathcal{C})$, where $\mathcal{C}$ denotes the class of $\mathcal{C}^2$-functions with compact support and Lipschitz continuous second derivatives. $\mathcal{L}$ is defined by (42). Then, from the property of Langevin dynamics, we know that (7) convergence to a unique invariant distribution $\pi^\star$. Further, by proving that the limit of every weakly converged subsequence equals to $\pi^\star$ and combining it with the Prokhorov's theorem, we conclude $\check{\boldsymbol{u}}_n$ converges to $\pi^\star$ weakly. Finally, repeat the first step of this proof, and we have $\{\bar{\boldsymbol{u}}_t^{(n)}\}$ converges to the solution of equation (7) with initial distribution $\pi^\star$.

**Step 1** Let $g$ belong to $\mathcal{C}$ and let $\mathcal{F}_t^{(n)}$ denote the natural filtration of $\bar{\boldsymbol{u}}_t^{(n)}$. We aim to derive the following equation, which can guarantee that every sub-limit of $\{\bar{\boldsymbol{u}}_t^{(n)}\}$ is a weak solution of the martingale problem $(\mathcal{L}, \mathcal{C})$.

$$\forall t \geq 0, \quad g\left(\bar{\boldsymbol{u}}_t^{(n)}\right) - g\left(\bar{\boldsymbol{u}}_0^{(n)}\right) - \int_0^t \mathcal{L}g\left(\bar{\boldsymbol{u}}_s^{(n)}\right) ds = \mathcal{M}_t^{(n,g)} + \mathcal{R}_t^{(n,g)} \tag{48}$$

Where $\mathcal{M}_t^{(n,g)}$ is a $\mathcal{F}_t^{(n)}$-martingale and $\mathcal{R}_t^{(n,g)}$ converges to zero in $L_1$.

In fact, we set

$$\mathcal{M}_t^{(n,g)} = \sum_{k=n+1}^{N(n,t,\eta)-1} \left\{g\left(\check{\boldsymbol{u}}_{k+1}\right) - g\left(\check{\boldsymbol{u}}_k\right) - \mathbb{E}\left[g\left(\check{\boldsymbol{u}}_{k+1}\right) - g\left(\check{\boldsymbol{u}}_k\right) \mid \mathcal{F}_k\right]\right\}$$

$$\mathcal{R}_t^{(n,g)} = g\left(\bar{\boldsymbol{u}}_t^{(n)}\right) - g\left(\bar{\boldsymbol{u}}_{\underline{t}_n}^{(n)}\right) - \int_{\underline{t}_n}^t \mathcal{L}g\left(\bar{\boldsymbol{u}}_s^{(n)}\right) ds \tag{49}$$

$$+ \int_0^{\underline{t}_n}\left(\mathcal{L}g\left(\bar{\boldsymbol{u}}_{\underline{s}_n}^{(n)}\right) - \mathcal{L}g\left(\bar{\boldsymbol{u}}_s^{(n)}\right)\right) ds + \sum_{k=n}^{N(n,t,\eta)-1} \mathcal{R}_k^g$$

From the definition of $\bar{\boldsymbol{u}}_t^{(n)}$ (6), we can get

$$\bar{\boldsymbol{u}}_t^{(n)} - \bar{\boldsymbol{u}}_{\underline{t}_n(\eta)}^{(n)} = (t - \underline{t}_n(\eta))\boldsymbol{b}_{N(n,t,\eta)} + \sqrt{t - \underline{t}_n(\eta)}\xi_{N(n,t,\eta)} \tag{50}$$

Which satisfies

$$\mathbb{E}\left\|\bar{\boldsymbol{u}}_t^{(n)} - \bar{\boldsymbol{u}}_{\underline{t}_n(\eta)}^{(n)}\right\| \leq (t - \underline{t}_n(\eta))\mathbb{E}\|\boldsymbol{b}_{N(n,t,\eta)}\| + \sqrt{t - \underline{t}_n(\eta)}\mathbb{E}\|\xi_{N(n,t,\eta)}\| \tag{51}$$
$$\precsim \sqrt{\eta_{N(n,t,\eta)}}$$

Plug the above bound into the residual $\mathcal{R}_t^{(n,g)}$, and note the Lipschitz continuity and boundedness of $g$, $\nabla g$ and $\nabla^2 g$,

$$\mathbb{E}\left|g(\bar{\boldsymbol{u}}_t^{(n)}) - g(\bar{\boldsymbol{u}}_{\underline{t}_n(\eta)}^{(n)})\right| \precsim \mathbb{E}\left\|\bar{\boldsymbol{u}}_t^{(n)} - \bar{\boldsymbol{u}}_{\underline{t}_n(\eta)}^{(n)}\right\| = o(1)$$

$$\mathbb{E}\left|\int_{\underline{t}_n(\eta)}^t \mathcal{L}g(\bar{\boldsymbol{u}}_s^{(n)})ds\right| \precsim \int_{\underline{t}_n(\eta)}^t \mathcal{C} \precsim \eta_{N(n,t,\eta)} = o(1)$$

$$\mathbb{E}\left|\int_0^{\underline{t}_n(\eta)} \mathcal{L}g(\bar{\boldsymbol{u}}_{\underline{s}_n(\eta)}^{(n)}) - \mathcal{L}g(\bar{\boldsymbol{u}}_s^{(n)})ds\right| \precsim \mathbb{E}\int_0^{\underline{t}_n(\eta)}\left\|\bar{\boldsymbol{u}}_{\underline{s}_n(\eta)}^{(n)} - \bar{\boldsymbol{u}}_s^{(n)}\right\|ds \tag{52}$$

$$\precsim \int_0^{\underline{t}_n(\eta)}\sqrt{\eta_{N(n,s,\eta)}}ds \leq \sqrt{\eta_n} = o(1)$$

Further, attributed to Lemma 8,

$$\mathbb{E}\left|\sum_{k=n}^{N(n,t,\eta)-1}\mathcal{R}_k^g\right| \leq \sum_{k=n}^{N(n,t,\eta)}\eta_k\mathbb{E}\left|\frac{\mathcal{R}_k^g}{\eta_k}\right| \tag{53}$$

$$\leq \sup_{k\geq n}\mathbb{E}\left|\frac{\mathcal{R}_k^g}{\eta_k}\right|\sum_{k=n}^{N(n,t,\eta)}\eta_k \precsim o(1)t = o(1)$$

So far we can say that $\mathbb{E}|\mathcal{R}_t^{(n,g)}| \to 0$, $n \to \infty$.

**Step 2**  Now we suppose that there exists a weakly convergent subsequence $\{\check{\boldsymbol{u}}_{n_k}\}_{k=1}^\infty$ with limit distribution $\tilde{\pi}$. We should introduce some new notations. For $n \in \mathbb{N}$ and $t \geq 0$, we define $M(n,t,\eta) = \min\left\{m \geq 0; \sum_{i=m}^{n-1}\eta_i \leq t\right\}$ and $\tilde{t}_n(\eta) = \Gamma_n - \Gamma_{M(n,t,\eta)}$. For the properties of step size sequence $\eta_n$, we can affirm $t - \tilde{t}_n(\eta) \to 0$ when $n \to \infty$.

By leveraging the Prokhorov's theorem, for any $T > 0$, we know that $\left\{\bar{\boldsymbol{u}}_t^{(M(n_k,T,\eta))}\right\}$ has a weakly convergent subsequence. Without loss of generality, we can assume that the subsequence $\left\{\bar{\boldsymbol{u}}_t^{(M(n_k,T,\eta))}\right\}$ itself converges weakly to a solution $\bar{\boldsymbol{u}}_t^{\tilde{\nu}^{(T)}}$ of the SDE (7) with initial distribution $\tilde{\nu}^{(T)}$. Owing to the tightness of the whole sequence $\{\bar{\boldsymbol{u}}^n\}$, for any given $\epsilon > 0$, there is a compact set $K_\epsilon \subset \mathbb{R}^d$ only depends on $\epsilon$ such that $\sup_n \mathbb{P}(\check{\boldsymbol{u}}_n \in K_\epsilon^c) \leq \epsilon$. This makes us find the following holds: $\tilde{\nu}^{(T)}(K_\epsilon) \geq 1 - \epsilon$ for any $T > 0$.

By the geometrical ergodicity of the dynamics (7), we can choose $T_\epsilon$ such that

$$\sup_{\boldsymbol{x}\in K_\epsilon}\sup_{g\in\mathcal{C}}\left|\mathcal{P}^{T_\epsilon}g(\boldsymbol{x}) - \langle\pi^\star, g\rangle\right| \leq \epsilon \tag{54}$$

Where $\mathcal{P}$ represents the Markov semigroup induced by the SDE (7). In virtue of the approximation of $\widetilde{(T_\epsilon)}_n(\eta)$ to $T_\epsilon$ and the tightness of the sequence $\bar{\boldsymbol{u}}^{(n)}$, we are able to deduce that $\check{\boldsymbol{u}}_{n_k}(= \bar{\boldsymbol{u}}_{\widetilde{(T_\epsilon)}_n(\eta)}^{M(n_k,T_\epsilon,\eta)})$ converges weakly to the limit random variable of the sequence $\bar{\boldsymbol{u}}_{T_\epsilon}^{M(n_k,T_\epsilon,\eta)}$ i.e., $\bar{\boldsymbol{u}}_{T_\epsilon}^{\tilde{\nu}^{(T_\epsilon)}}$. On the other hand, by assumption, $\check{\boldsymbol{u}}_{n_k}$ converges weakly to $\tilde{\pi}$. Thus $\bar{\boldsymbol{u}}_{T_\epsilon}^{\tilde{\nu}^{(T_\epsilon)}} \sim \tilde{\pi}$.

Given any $g \in \mathcal{C}$, it is not difficult to derive the following bounds

$$|\langle \tilde{\pi}, g \rangle - \langle \pi^\star, g \rangle| = \left| \mathbb{E} g \left( \bar{\boldsymbol{u}}_{T_\epsilon}^{(T_\epsilon)} \right) - \mathbb{E}_{\pi^\star} g \right| = \left| \int \left( \mathcal{P}^{T_\epsilon} g(\boldsymbol{x}) - \mathbb{E}_{\pi^\star} g \right) d\tilde{\nu}^{(T_\epsilon)}(\boldsymbol{x}) \right|$$

$$\leq \int \left| \mathcal{P}^{T_\epsilon} g(\boldsymbol{x}) - \mathbb{E}_{\pi^\star} g \right| d\tilde{\nu}^{(T_\epsilon)}(\boldsymbol{x})$$

$$= \int_{K_\epsilon} \left| \mathcal{P}^{T_\epsilon} g(\boldsymbol{x}) - \mathbb{E}_{\pi^\star} g \right| d\tilde{\nu}^{(T_\epsilon)}(\boldsymbol{x}) + \int_{K_\epsilon^c} \left| \mathcal{P}^{T_\epsilon} g(\boldsymbol{x}) - \mathbb{E}_{\pi^\star} g \right| d\tilde{\nu}(\boldsymbol{x}) \quad (55)$$

$$\leq \int_{K_\epsilon} \left| \mathcal{P}^{T_\epsilon} g(\boldsymbol{x}) - \mathbb{E}_{\pi^\star} g \right| d\tilde{\nu}^{(T_\epsilon)}(\boldsymbol{x}) + 2\|g\|_\infty \tilde{\nu}^{(T_\epsilon)}(K_\epsilon^c)$$

$$\overset{(a)}{\leq} \epsilon + 2\|g\|_\infty \epsilon$$

Where (a) holds for sake of $\tilde{\nu}^{(T_\epsilon)}(K_\epsilon) \geq 1 - \epsilon$ and (54). We obtain $\tilde{\pi} = \pi^\star$ by taking $\epsilon \to 0$. Finally, owing to the Prokhorov's theorem, we have proved that $\check{\boldsymbol{u}}_n$ converges weakly to $\pi^\star$. Further, the sequence of random process $\bar{\boldsymbol{u}}_t^{(n)}$ converges weakly to the dynamics (7) with stationary distribution $\pi^\star$ as initialization.

$\square$

## C.2  Proof of Case 2

We first complete the formulation of the recurrence relation for $\boldsymbol{v}_n$ that was omitted from the main text

$$\check{\boldsymbol{v}}_{(n+1)-} = \eta_n^{\beta-1} \mathcal{P}_{\boldsymbol{A}}(\boldsymbol{x}_n - \eta_n \nabla f(\boldsymbol{x}_n) + \eta_n \xi_n)$$

$$= \left( \frac{\eta_n}{\eta_{n-1}} \right)^{\beta-1} \check{\boldsymbol{v}}_n - \eta_n^\beta \mathcal{P}_{\boldsymbol{A}} \nabla f(\boldsymbol{x}_n) + \eta_n^\beta \mathcal{P}_{\boldsymbol{A}} \xi_n$$

$$= \left( \frac{\eta_n}{\eta_{n-1}} \right)^{\beta-1} \check{\boldsymbol{v}}_n - \eta_n^\beta \nabla f(\boldsymbol{x}^\star) - \eta_n^\beta \mathcal{P}_{\boldsymbol{A}} \left( \nabla f(\boldsymbol{x}_n) - \nabla f(\boldsymbol{u}_n) \right) \quad (56)$$

$$- \eta_n^\beta \mathcal{P}_{\boldsymbol{A}} (\nabla f(\boldsymbol{u}_n) - \nabla f(\boldsymbol{x}^\star)) + \eta_n^\beta \mathcal{P}_{\boldsymbol{A}} \xi_n$$

$$= \check{\boldsymbol{v}}_n - \eta_n^\beta \nabla f(\boldsymbol{x}^\star) - \mathcal{S}_n^{(1)} - \mathcal{S}_n^{(2)} - \mathcal{S}_n^{(3)} + \eta_n^\beta \xi_n^{(2)}$$

$$=: \check{\boldsymbol{v}}_n - \eta_n^\beta \boldsymbol{d}_n + \eta_n^\beta \xi_n^{(2)}$$

Where $\boldsymbol{d}_n = \nabla f(\boldsymbol{x}^\star) + \frac{1}{\eta_n^\beta} \mathcal{S}_n^{(1)} + \frac{1}{\eta_n^\beta} \mathcal{S}_n^{(2)} + \frac{1}{\eta_n^\beta} \mathcal{S}_n^{(3)}$ with higher order terms

$$\mathcal{S}_n^{(1)} = \left( 1 - \frac{\eta_n^\beta}{\eta_{n-1}^\beta} \right) \check{\boldsymbol{v}}_n$$

$$\mathcal{S}_n^{(2)} = \eta_n^\beta \mathcal{P}_{\boldsymbol{A}} (\nabla f(\boldsymbol{x}_n) - \nabla f(\boldsymbol{u}_n)) \quad (57)$$

$$\mathcal{S}_n^{(3)} = \eta_n^\beta \mathcal{P}_{\boldsymbol{A}} (\nabla f(\boldsymbol{u}_n) - \nabla f(\boldsymbol{x}^\star))$$

We can see from the Theorem 3.1 that both $\frac{\mathcal{S}_n^{(2)}}{\eta_n^\beta}$ and $\frac{\mathcal{S}_n^{(3)}}{\eta_n^\beta}$ are of order $o(\eta_n^{1-\beta})$ in $L_1$. Moreover, owing to the slow diminishing property of step size $\{\eta_n\}$, the following bound holds

$$1 - \frac{\eta_n^\beta}{\eta_{n-1}^\beta} = 1 - \left( 1 + \frac{\eta_n - \eta_{n-1}}{\eta_{n-1}} \right)^\beta$$

$$= 1 - (1 + \mathcal{O}(\eta_n))^\beta = 1 - (1 + \beta \mathcal{O}(\eta_n)) = \mathcal{O}(\eta_n). \quad (58)$$

So

$$\frac{1}{\eta_n^\beta} \mathbb{E} \left| \mathcal{S}_n^{(1)} \right| \precsim \frac{\mathcal{O}(\eta_n)}{\eta_n^\beta} = \mathcal{O}(\eta_n^{1-\beta}). \quad (59)$$

As in the derivation process of the first case, we first need to focus our attention on the discussion of tightness of the sequence of random process $\{\bar{\boldsymbol{v}}_t^{(n)}\}_{n=1}^\infty$. While the discontinuity of these processes constructed by (9) prevents the property 6 from being used to verify the tightness of $\{\bar{\boldsymbol{v}}_t^{(n)}\}$. Hence, we will leverage the following more general criterion for tightness proposed in [19].

**Proposition 7.** *Let $\{\mathbf{x}_n(t)\}$ be a sequence of $\mathbb{R}^d$-valued processes whose sample paths are càdlàg. Let*

$$\omega(\mathbf{x}_n, \delta, T) = \inf_{\{t_i\}} \max_i \sup_{t_{i-1} \le s < t < t_i} \|\mathbf{x}_t - \mathbf{x}_s\|. \tag{60}$$

*Where $\{t_i\}$ ranges over all finite partitions of the form $0 = t_0 \le t_1 < t_2 < \cdots < t_{r-1} < T \le t_r$ with $\min_{1 \le i \le r} (t_i - t_{i-1}) \ge \delta$. Then the sequence of processes $\{\mathbf{x}_n(t)\}$ is tight if and only if,*

1. *for every $T > 0$ and $\eta > 0$, there is a compact set $K$ such that*

$$\liminf_{n \to \infty} \mathbb{P}(\mathbf{x}_n(t) \in K; \ \forall t \in [0,T]) > 1 - \eta; \tag{61}$$

2. *for every $\epsilon, \eta > 0$, and $T > 0$, there is a $\delta > 0$ such that*

$$\limsup_{n \to \infty} \mathbb{P}(\omega(\mathbf{x}_n, \delta, T) \ge \epsilon) < \eta. \tag{62}$$

Denote $\mathcal{I}_n(T) = \left\{ N(n, t, \eta^\beta) : t \in [0, T] \right\} \subset \mathbb{N}$ and $\mathcal{L}_n(T) = \{\Gamma_{n+k} - \Gamma_n : \ k \in \mathcal{I}_n(T)\}$. From the update rule of parametric sequence $\{\boldsymbol{x}_n\}$ and the construction of the rescaling process $\{\bar{\boldsymbol{v}}_t^{(n)}\}$, an intuitive fact is that the discontinuous points of $\bar{\boldsymbol{v}}_t^{(n)}$ in the interval $[0, T]$ belong to $\mathcal{L}_n(T)$. And we have the following lemma to support the proof of tightness.

**Lemma 9.** *For the sequence of càdlàg processes $\{\bar{\boldsymbol{v}}^{(n)}\}$, consider the time point set $\mathcal{J}_n(T)$ such that*

$$\mathcal{J}_n(T) = \left\{ N(n, t, \eta^\beta) : t \in [0, T] \text{ and } \bar{\boldsymbol{v}}_{\underline{t}_n(\eta^\beta)}^{(n)} \ne \bar{\boldsymbol{v}}_{\underline{t}_n(\eta^\beta)-}^{(n)} \right\} \tag{63}$$

*and let*

$$\Delta(\mathcal{J}_n(T)) = \min \left\{ |\Gamma_{n+k}(\eta^\beta) - \Gamma_{n+l}(\eta^\beta)| : k, l \in \mathcal{J}_n(T) \text{ and } k \ne l \right\} \tag{64}$$

*Then there is a universal constant $C$ and an $n_0 \in \mathbb{N}$ subject to for any $\delta > 0$*

$$\mathbb{P}(\Delta(\mathcal{J}_n(T)) < \delta) \le C\delta \quad \forall n \ge n_0. \tag{65}$$

*Proof.* ( **Proof of Lemma 9**)

By the sub-additivity of probability and the jump scheme proposed in the Algorithm 1, we have

$$
\begin{aligned}
&\mathbb{P}(\Delta(\mathcal{J}_n(T)) < \delta) \\
&\le \sum_{k \in \mathcal{I}_n(T)} \mathbb{P} \left\{ \exists (k \le) l \in \mathcal{J}_n(T) \text{ s.t. } |\Gamma_{n+l}(\eta^\beta) - \Gamma_{n+k}(\eta^\beta)| < \delta; \ k \in \mathcal{J}_n(T) \right\} \\
&\le \sum_{k \in \mathcal{I}_n(T)} \sum_{0 < \Gamma_{n+l}(\eta^\beta) - \Gamma_{n+k}(\eta^\beta) < \delta} \mathbb{P} \left\{ l \in \mathcal{J}_n(T); k \in \mathcal{J}_n(T) \right\} \\
&\le \sum_{k \in \mathcal{I}_n(T)} p_{n+k} \left( \sum_{0 < \Gamma_{n+l}(\eta^\beta) - \Gamma_{n+k}(\eta^\beta) < \delta} p_{n+l} \right) \\
&\le \gamma^2 \sum_{k \in \mathcal{I}_n(T)} \eta_{n+k}^\beta \left( \sum_{0 < \Gamma_{n+l}(\eta^\beta) - \Gamma_{n+k}(\eta^\beta) < \delta} \eta_{n+l}^\beta \right) \\
&\overset{(a)}{\le} \gamma^2 (\delta + \eta_n^\beta) \sum_{k \in \mathcal{I}_n(T)} \eta_{n+k}^\beta \le 2\gamma^2 T\delta.
\end{aligned}
\tag{66}
$$

Where (a) holds when we let $\eta_n^\beta \le \delta$. We conclude the proof by letting $\mathcal{C} = 2\gamma^2 T$. $\qquad \square$

**Lemma 10.** *Suppose that Assumptions 1, 2 and 4 holds. Then the sequence of random processes $\{\bar{\boldsymbol{v}}^{(n)}\}$ is tight under the Skorokhod topology in finite interval.*

*Proof.* (**Proof of Lemma 10**) What we need to do now is to verify the conditions in the Property 7 one by one.

For a given path $\bar{\boldsymbol{v}}^{(n)}$, denote $t'_n = \max\{s \in \mathcal{L}_n(T) \cap [0,t]\} \cup \{0\}$. Then $\bar{\boldsymbol{v}}_{t'_n}^{(n)} = 0$ whenever $t'_n > 0$. Let $R > 0$, we have the following inequalities

$$
\begin{aligned}
&\mathbb{P}\left(\sup_{t\in[0,T]}\left\|\bar{\boldsymbol{v}}_t^{(n)}\right\| \geq R\right) \\
&\leq \mathbb{P}\left(\sup_{t\in[0,T]}\left\{\left\|\bar{\boldsymbol{v}}_t^{(n)} - \bar{\boldsymbol{v}}_{\underline{t}_n(\eta^\beta)}^{(n)}\right\| + \left\|\bar{\boldsymbol{v}}_{\underline{t}_n(\eta^\beta)}^{(n)} - \bar{\boldsymbol{v}}_{t'_n}^{(n)}\right\| + \left\|\bar{\boldsymbol{v}}_{t'_n}^{(n)}\right\|\right\} \geq R\right) \\
&\leq \frac{2}{R}\mathbb{E}\sup_{t\in[0,T]}\left\{(t - \underline{t}_n(\eta^\beta))\left\|\boldsymbol{d}_{\underline{t}_n(\eta^\beta)} - \xi_{\underline{t}_n(\eta^\beta)}^{(2)}\right\| + \left\|\bar{\boldsymbol{v}}_{\underline{t}_n(\eta^\beta)}^{(n)} - \bar{\boldsymbol{v}}_{t'_n}^{(n)}\right\|\right\} + \frac{2}{R}\mathbb{E}\left\|\bar{\boldsymbol{v}}_0^{(n)}\right\| \\
&\leq \frac{2}{R}\mathbb{E}\sup_{t\in\mathcal{L}_n(T)}\left\|\bar{\boldsymbol{v}}_{t-}^{(n)} - \bar{\boldsymbol{v}}_{(t-)'_n}^{(n)}\right\| + \frac{2}{R}\mathbb{E}\left\|\check{\boldsymbol{v}}_n\right\| \\
&\leq \frac{2}{R}\mathbb{E}\sup_{k\in\mathcal{I}_n(T)}\sum_{i=n}^{n+k-1}\eta_i^\beta\left\|\boldsymbol{d}_i + \xi_i^{(2)}\right\| + \frac{2}{R}\mathbb{E}\|\check{\boldsymbol{v}}_n\| \\
&= \frac{2}{R}\sum_{i=n}^{n+\sup\mathcal{I}_n(T)-1}\eta_i^\beta\mathbb{E}\left\|\boldsymbol{d}_i + \xi_i^{(2)}\right\| + \frac{2}{R}\mathbb{E}\|\check{\boldsymbol{v}}_n\| \\
&\leq \frac{2}{R}(T + \eta_n)\sup_{n\in\mathbb{N}}\mathbb{E}\left\|\boldsymbol{d}_i + \xi_i^{(2)}\right\| + \frac{2}{R}\mathbb{E}\|\check{\boldsymbol{v}}_n\| \\
&\stackrel{(a)}{\leq} \frac{\mathcal{C}(1 + T)}{R} \leq \eta.
\end{aligned}
$$

(67)

Where (a) holds for the uniform boundedness of $\mathbb{E}\|\boldsymbol{d}_n + \xi_n^{(2)}\| + \mathbb{E}\|\check{\boldsymbol{v}}_n\|$. And the final inequality holds when we take $\mathbb{R} > \frac{\mathcal{C}(1+T)}{\eta}$. Thus, the first condition of the Proposition 7 holds for $\bar{\boldsymbol{v}}^{(n)}$.

As for the second condition, what we should do is to construct an appropriate partition that makes $\omega(\bar{\boldsymbol{v}}^{(n)}, \delta, T)$ defined as (60) as small as possible.

For a given $\epsilon, \eta$ pair, let $\delta < \frac{\eta}{2\mathcal{C}}$, then from the Lemma 9 it can be seen $\mathbb{P}(\Delta(\mathcal{J}_n(T)) < \delta) < \frac{\eta}{2}$. Now given the event $\mathcal{E} = \{\Delta(\mathcal{J}_n(T)) \geq \delta\}$, we choose the partition points $\{\tau_k\} \in [0,T]$ recursively from the set $\mathcal{L}_n(T)$ such that the partition satisfies the following properties:

1. $\min_k\{\tau_k - \tau_{k-1}\} \in [\delta, 3\delta)$

2. $\mathcal{J}_n(T) \subset \{\tau_k\}$

Let $\tau_0 = 0$ and suppose we have constructed the partition points $\tau_0, \cdots, \tau_k \in [0,T]$ with inductive assumptions:

1. $\min_{i\leq k-1}\{\tau_{i+1} - \tau_i\} \in [\delta, 3\delta)$,

2. $\mathcal{J}_n(\tau_k) \subset \{\tau_0, \tau_1, \cdots, \tau_k\}$,

3. there is no discontinuous point in $(\tau_k, \tau_k + \delta)$, i.e. $\mathcal{J}_n(T) \cap (\tau_k, \tau_k + \delta) = \varnothing$.

We will use these results to find the next partition point $\tau_{k+1}$. Define $\tilde{\tau}_{k+1} = \min\{t : t - \tau_k \geq \delta, t \in \mathcal{L}_n(T)\}$, we use the following scheme:

$$
\tau_{k+1} = \begin{cases} s & \exists\, s \in (\tilde{\tau}_{k+1}, \tilde{\tau}_{k+1} + \delta) \cap \mathcal{J}_n(T) \\ \tilde{\tau}_{k+1} & \text{Otherwise} \end{cases}
$$

(68)

From the property of the event $\mathcal{E}$ we know there is at most one discontinuous point in $(\tilde{\tau}_{k+1}, \tilde{\tau}_{k+1}+\delta)$, which means the $\tau_{k+1}$ is always well-defined. Then we have $\delta \leq \tau_{k+1} - \tau_k \leq \tau_{k+1} - \tilde{\tau}_{k+1} + \tilde{\tau}_{k+1} - \tau_k \leq \delta + \eta_n + \delta \leq 3\delta$. Where the last inequality holds when we choose $n_0$ such that $\eta_{n_0} < \delta$. Thus $\tau_{k+1}$ satisfies the first inductive assumption.

By the third inductive assumption of $\tau_k$, we know there is no discontinuous point in $(\tau_k, \tilde{\tau}_{k+1})$. On the other hand, if $\tau_{k+1} \in \mathcal{J}_n(T)$, then $(\tau_{k+1} - \delta, \tau_{k+1}) \cap \mathcal{J}_n(T) = \varnothing$, and especially we have $[\tilde{\tau}_{k+1}, \tau_{k+1}) \cap \mathcal{J}_n(T) = \varnothing$. Hence, the second inductive assumption of $\tau_{k+1}$ has been proved.

Finally, if $\tau_{k+1} = \tilde{\tau}_{k+1}$, then from the recursive construction scheme (68) we know $(\tau_{k+1}, \tau_{k+1} + \delta) \cap \mathcal{J}_n(T) = (\tilde{\tau}_{k+1}, \tilde{\tau}_{k+1} + \delta) \cap \mathcal{J}_n(T) = \varnothing$. Else, $\tau_{k+1}$ must belong to $\mathcal{J}_n(T)$. Combining the definition of $\mathcal{E}$ we can make sure $(\tau_{k+1}, \tau_{k+1} + \delta) \cap \mathcal{J}_n(T) = \varnothing$. At this point, the proofs of the three inductive assumptions on $\tau_{k+1}$ are all complete.

$$
\begin{aligned}
\mathbb{P}(\omega(\bar{\boldsymbol{v}}^{(n)}, \delta, T) \geq \epsilon) &\leq \mathbb{P}(\omega(\bar{\boldsymbol{v}}^{(n)}, \delta, T) \geq \epsilon;\ \mathcal{E}) + \mathbb{P}(\mathcal{E}^c) \\
&\leq \mathbb{P}\left(\max_k \sup_{\tau_k \leq t < s < \tau_{k+1}} \left\| \bar{\boldsymbol{v}}_t^{(n)} - \bar{\boldsymbol{v}}_s^{(n)} \right\| \geq \epsilon;\ \mathcal{E}\right) + \frac{\eta}{2} \\
&\leq 2\mathbb{P}\left(\sup_{t \in [0,T]} \left\| \bar{\boldsymbol{v}}_t^{(n)} - \bar{\boldsymbol{v}}_{\underline{t}_n(\eta^\beta)}^{(n)} \right\| \geq \frac{\epsilon}{2}\right) + \frac{\eta}{2} \\
&\quad + \mathbb{P}\left(\max_k \sup_{\tau_k \leq t < s < \tau_{k+1}} \left\| \bar{\boldsymbol{v}}_{\underline{t}_n(\eta^\beta)}^{(n)} - \bar{\boldsymbol{v}}_{\underline{s}_n(\eta^\beta)}^{(n)} \right\| \geq \frac{\epsilon}{2};\ \mathcal{E}\right)
\end{aligned}
\tag{69}
$$

We will give the bound of two probabilities respectively. First,

$$
\begin{aligned}
\mathbb{P}\left(\sup_{t \in [0,T]} \left\| \bar{\boldsymbol{v}}_t^{(n)} - \bar{\boldsymbol{v}}_{\underline{t}_n(\eta^\beta)}^{(n)} \right\| \geq \frac{\epsilon}{2}\right) &= \mathbb{P}\left(\sup_{k \in \mathcal{I}_n(T)} \left\| \bar{\boldsymbol{v}}_{(\Gamma_{k+1} - \Gamma_n)-}^{(n)} - \bar{\boldsymbol{v}}_{\Gamma_k - \Gamma_n}^{(n)} \right\| \geq \frac{\epsilon}{2}\right) \\
&\leq \sum_{k \in \mathcal{I}_n(T)} \mathbb{P}\left(\left\| \check{\boldsymbol{v}}_{(n+k+1)-} - \check{\boldsymbol{v}}_{n+k} \right\| \geq \frac{\epsilon}{2}\right) \leq \sum_{k \in \mathcal{I}_n(T)} \frac{4}{\epsilon^2} \mathbb{E}\left\| \check{\boldsymbol{v}}_{(n+k+1)-} - \check{\boldsymbol{v}}_{n+k} \right\|^2 \\
&\leq \frac{4}{\epsilon^2} \sum_{k \in \mathcal{I}_n(T)} \eta_{n+k}^{2\beta} \mathbb{E}\| \boldsymbol{d}_{n+k} + \xi_{n+k}^{(2)} \|^2 \leq \frac{4\eta_n^\beta \sup_i \mathbb{E}\| \boldsymbol{d}_i + \xi_i^{(2)} \|^2}{\epsilon^2} \sum_{k \in \mathcal{I}_n(T)} \eta_{n+k}^\beta \\
&\leq \frac{\mathcal{C}(T + \eta_n^\beta)}{\epsilon^2} \eta_n^\beta < \frac{2\mathcal{C}T}{\epsilon} \eta_n^\beta < \frac{\eta}{8}.
\end{aligned}
\tag{70}
$$

Where the last inequality holds when we take $\eta_{n_0}^\beta < \frac{\epsilon\eta}{16\mathcal{C}T}$.

It is easy to see that we can use a bijection to link the elements in $\mathcal{I}_n(T)$ and that in $\mathcal{L}_n(T)$. Because the partition points $\{\tau_k\}$ are in $\mathcal{L}_n(T)$, we assume that every $\tau_k$ corresponds to an index $\varsigma_k \in \mathcal{I}_n(T)$. Then we have $\varsigma_{k+1} > \varsigma_k$. Denote $\mathfrak{S}_k = \mathcal{I}_n(T) \cap [\varsigma_k, \varsigma_{k+1})$. So far we are ready to bound the last

term in (69).

$$\mathbb{P}\left(\max_k \sup_{\tau_k \leq t < s < \tau_{k+1}} \left\|\bar{\boldsymbol{v}}^{(n)}_{\underline{t}_n(\eta^\beta)} - \bar{\boldsymbol{v}}^{(n)}_{\underline{s}_n(\eta^\beta)}\right\| \geq \frac{\epsilon}{2}; \mathcal{E}\right)$$

$$\leq \sum_k \mathbb{P}\left(\sup_{l,h \in \mathfrak{S}_k} \|\check{\boldsymbol{v}}_{n+l} - \check{\boldsymbol{v}}_{n+h}\| \geq \frac{\epsilon}{2}; \mathcal{E}\right)$$

$$\leq \sum_k \mathbb{P}\left(\sup_{l,h \in \mathfrak{S}_k} \sum_{i=l}^{h-1} \|\check{\boldsymbol{v}}_{n+i+1} - \check{\boldsymbol{v}}_{n+i}\| \geq \frac{\epsilon}{2}; \mathcal{E}\right) = \sum_k \mathbb{P}\left(\sum_{i=\varsigma_k}^{\varsigma_{k+1}-1} \|\check{\boldsymbol{v}}_{n+i+1} - \check{\boldsymbol{v}}_{n+i}\| \geq \frac{\epsilon}{2}; \mathcal{E}\right)$$

$$\overset{(a)}{\leq} \sum_k \mathbb{P}\left(\sum_{i=\varsigma_k}^{\varsigma_{k+1}-1} \eta^\beta_{n+i} \left\|\boldsymbol{d}_{n+i} + \xi^{(2)}_{n+i}\right\| \geq \frac{\epsilon}{2}\right) \leq \sum_k \frac{4}{\epsilon^2} \mathbb{E}\left(\sum_{i=\varsigma_k}^{\varsigma_{k+1}-1} \eta^\beta_{n+i} \left\|\boldsymbol{d}_{n+i} + \xi^{(2)}_{n+i}\right\|\right)^2$$

$$\leq \frac{4}{\epsilon^2} \sum_k \left\{\left(\sum_{i=\varsigma_k}^{\varsigma_{k+1}-1} \eta^\beta_{n+i}\right)\left(\sum_{i=\varsigma_k}^{\varsigma_{k+1}-1} \eta^\beta_{n+i} \mathbb{E}\left\|\boldsymbol{d}_{n+i} + \xi^{(2)}_{n+i}\right\|^2\right)\right\}$$

$$\leq \frac{4 \sup_i \mathbb{E}\|\boldsymbol{d}_i + \xi^{(2)}_i\|^2}{\epsilon^2} \sum_k \left(\sum_{i=\varsigma_k}^{\varsigma_{k+1}-1} \eta^\beta_{n+i}\right)^2 \leq \frac{\mathcal{C}}{\epsilon^2} \sum_k (\tau_{k+1} - \tau_k)^2$$

$$\overset{(b)}{\leq} \frac{3\mathcal{C}\delta}{\epsilon^2} \sum_k (\tau_{k+1} - \tau_k) \leq \frac{3\mathcal{C}T\delta}{\epsilon^2} < \frac{\eta}{4}.$$

(71)

Where (a) follows from the combination of the fact that the path $\bar{\boldsymbol{v}}^{(n)}$ is continuous in any interval $[\tau_{k+1}, \tau_k)$ when $\mathcal{E}$ holds and the update formula (9). And (b) is true by the property of the partition $\{\tau_k\}$ listed above. The last inequality holds when we take $\delta < \frac{\eta\epsilon^2}{12\mathcal{C}T}$.

Bring (70) and (71) into (69), we have,

$$\mathbb{P}(\omega(\bar{\boldsymbol{v}}^{(n)}, \delta, T) \geq \epsilon) \geq 2 \cdot \frac{\eta}{8} + \frac{\eta}{2} + \frac{\eta}{4} = \eta.$$

(72)

At this point, we have checked the two sufficient conditions in the Property 7. Hence, the tightness of $\{\bar{\boldsymbol{v}}^{(n)}\}$ has been proved. $\qquad\square$

**Lemma 11.** *Suppose Assumptions 1, 1 and 4 holds, and assume that there exists a positive number $p > 2$ such that $\sup_{n \geq 0} \mathbb{E}\|\xi_n\|^p < \infty$. When $p_n = \gamma\eta^\beta_n$ with $\gamma > 0$,*

*then for any $C^2$ function $g : \mathbb{R}^d \to \mathbb{R}$, compactly supported with Lipschitz continuous second derivatives, we have*

$$\mathbb{E}[g(\check{\boldsymbol{v}}_{n+1}) - g(\check{\boldsymbol{v}}_n)|\mathcal{F}_n] = \eta^\beta_n \mathcal{J}g(\check{\boldsymbol{v}}_n) + \mathcal{T}^g_n$$

(73)

*Where $\frac{1}{\eta^\beta_n}\mathcal{T}^g_n \to 0$ in $L_1$ and $\mathcal{J}$ is the infinitesimal generator defined by*

$$\forall \phi \in \mathcal{C}^2(\mathbb{R}^p) \quad \mathcal{J}\phi(\boldsymbol{x}) = \langle-\nabla f(\boldsymbol{x}^\star), \nabla\phi(\boldsymbol{x})\rangle + \gamma(\phi(\boldsymbol{0}) - \phi(\boldsymbol{x}))$$

(74)

*Proof.* (**Proof of Lemma 11**) We would like to say that the overall proof framework is similar to the proof of the Lemma 8. However, since $\check{\boldsymbol{v}}_{n+1}$ may suddenly jump to 0, we cannot directly use Taylor expansion to get the desired result. First, by the scheme on $\check{\boldsymbol{v}}_{n+1}$ jumping to zero, we have

$$\mathbb{E}[g(\check{\boldsymbol{v}}_{n+1}) - g(\check{\boldsymbol{v}}_n)|\mathcal{F}_n] = p_n(g(\boldsymbol{0}) - g(\check{\boldsymbol{v}}_n)) + (1 - p_t)\mathbb{E}[g(\check{\boldsymbol{v}}_{(n+1)-}) - g(\check{\boldsymbol{v}}_n)|\mathcal{F}_n]$$

(75)

Then we make use of the Taylor expansion between $\check{\boldsymbol{v}}_{(n+1)-}$ and $\check{\boldsymbol{v}}_n$.

$$
\begin{aligned}
g(\check{\boldsymbol{v}}_{(n+1)-}) - g(\check{\boldsymbol{v}}_n) &= \langle \nabla g(\check{\boldsymbol{v}}_n), \check{\boldsymbol{v}}_{(n+1)-} - \check{\boldsymbol{v}}_n \rangle \\
&+ \frac{1}{2} (\check{\boldsymbol{v}}_{(n+1)-} - \check{\boldsymbol{v}}_n)^\top \nabla^2 g(\varrho_n)(\check{\boldsymbol{v}}_{(n+1)-} - \check{\boldsymbol{v}}_n) \\
&= \eta^\beta \langle \nabla g(\check{\boldsymbol{v}}_n), \nabla f(\boldsymbol{x}^\star) + \xi_n^{(2)} \rangle + \eta_n^\beta \left\langle \nabla g(\check{\boldsymbol{v}}_n), \frac{1}{\eta_n^\beta} \sum_{i=1}^3 \mathcal{S}_n^{(i)} \right\rangle \\
&+ \frac{\eta_n^{2\beta}}{2} (\boldsymbol{d}_n + \xi_n^{(2)})^\top \nabla^2 g(\varrho_n)(\boldsymbol{d}_n + \xi_n^{(2)})
\end{aligned}
\tag{76}
$$

Substitute this equation into the second term of the right hand of the equation (75). It follows that

$$
\begin{aligned}
\mathbb{E}[g(\check{\boldsymbol{v}}_{n+1}) - g(\check{\boldsymbol{v}}_n) | \mathcal{F}_n] &= \eta_n^\beta \left( \gamma(g(\boldsymbol{0}) - g(\check{\boldsymbol{v}}_n)) + \langle \nabla g(\check{\boldsymbol{v}}_n), -\nabla f(\boldsymbol{x}^\star) \rangle \right) \\
&+ \eta_n^\beta \mathbb{E}\left[ \left\langle \nabla g(\check{\boldsymbol{v}}_n), \frac{1}{\eta_n^\beta} \sum_{i=1}^3 \mathcal{S}_n^{(i)} \right\rangle \Big| \mathcal{F}_n \right] + \frac{\eta_n^{2\beta}}{2} \mathbb{E}\left[ (\boldsymbol{d}_n + \xi_n^{(2)})^\top \nabla^2 g(\varrho_n)(\boldsymbol{d}_n + \xi_n^{(2)}) \Big| \mathcal{F}_n \right] \\
&- \gamma \eta_n^{2\beta} \mathbb{E}\left[ \langle \nabla g(\check{\boldsymbol{v}}_n, \boldsymbol{d}_n + \xi_n^{(2)} \rangle + \frac{\eta_n^\beta}{2} (\boldsymbol{d}_n + \xi_n^{(2)})^\top \nabla g(\varrho_n)(\boldsymbol{d}_n + \xi_n^{(2)}) \Big| \mathcal{F}_n \right] \\
&=: \eta_n^\beta \{ \gamma(g(\boldsymbol{0}) - g(\check{\boldsymbol{v}}_n)) - \langle \nabla g(\check{\boldsymbol{v}}_n), \nabla f(\boldsymbol{x}^\star) \rangle \} + \underbrace{\mathcal{T}_n^{(1)} + \mathcal{T}_n^{(2)} + \mathcal{T}_n^{(3)}}_{\mathcal{T}_n^g}
\end{aligned}
\tag{77}
$$

From the Theorem 3.1 and the equation (59), we have $\mathbb{E}|\mathcal{T}_n^{(1)}| = o(\eta_n^\beta)$. And $\mathbb{E}|\mathcal{T}_n^{(2)}| = \mathcal{O}(\eta_n^{2\beta})$ by leveraging that $\|\nabla^2 g(\boldsymbol{x})\|$ is bounded for all $\boldsymbol{x} \in \mathbb{R}^d$ and that $\mathbb{E}\|\boldsymbol{d}_n + \xi_n^{(2)}\|^2$ is bounded. Similar approaches can be used to show that $\mathbb{E}|\mathcal{T}_n^{(3)}| = \mathcal{O}(\eta_n^{2\beta})$. At this point, the result has been proved. $\quad\square$

From the Itô's formula for the semimartingales, we know that the infinitesimal generator $\mathcal{J}$ defined in the Lemma 11 corresponds to the following stochastic differential equation driven by the Poisson process with intensity $\gamma$.

$$
d\mathbf{Y}_t = -\nabla f(\boldsymbol{x}^\star)dt - \mathbf{Y}_t \cdot \mathbf{N}_\gamma(dt)
\tag{78}
$$

**Lemma 12.** *There exists a unique invariant measure $\boldsymbol{u}^\star$ for the Lévy process (78). Further, for any initial distribution $\nu_0$, we have $\mathcal{W}_2(\mathcal{G}^t \nu_0, \nu^\star) \to 0$; $t \to \infty$. Where $\mathcal{W}_2$ represents the Wasserstain-2 distance and $\{\mathcal{G}^t\}$ is the Markovian semigroup generated by the infinitesimal generator $\mathcal{J}$.*

*Proof.* (**Proof of Lemma 12**)

Consider the set of probability density functions $\left\{ h(\boldsymbol{x}) = p(t) \mathbb{1}_{\left\{ \boldsymbol{x} = \frac{\nabla f(\boldsymbol{x}^\star)}{\|f(\boldsymbol{x}^\star)\|} t \right\}} : p(t) \text{ is a p.d.f on } \mathbb{R} \right\}$
and denote it as $\mathcal{M}$. Then the distribution of any $\mathbf{Y}_t$ only has mass on the line $\left\{ \frac{\nabla f(\boldsymbol{x}^\star)}{\|f(\boldsymbol{x}^\star)\|} t : t \in \mathbb{R} \right\}$
if we choose the initial distribution in $\mathcal{M}$. In this case, we can suppose $\mathbf{Y}_t = -\frac{\nabla f(\boldsymbol{x}^\star)}{\|\nabla f(\boldsymbol{x}^\star)\|} \upsilon_t$.
Consequently, $\upsilon_t$ satisfies the following one dimensional stochastic differential equation,

$$
d\upsilon_t = \|\nabla f(\boldsymbol{x}^\star)\|dt - \upsilon_t \mathbf{N}_\gamma(dt).
\tag{79}
$$

Let $\varphi_t(\lambda) = \mathbb{E}_{p^\star} e^{i\lambda \upsilon_t}$ be the characteristic function of $\upsilon_t$ with stationary initialization $p^\star$. Then we have $\varphi_t(\lambda) = \varphi_s(\lambda)$; $\forall t \neq s$. On the other hand, consider the martingale problem corresponding to (79). It says that $e^{i\lambda \upsilon_t} - e^{i\lambda \upsilon_0} - \int_0^t \left( i\lambda \|\nabla f(\boldsymbol{x}^\star)\| e^{i\lambda \upsilon_s} + \gamma(1 - e^{i\lambda \upsilon_s}) \right) ds$ is a martingale with respect to the natural filtration generated by $\upsilon_t$. Take expectation we have

$$
\begin{aligned}
0 &= \varphi_t(\lambda) - \varphi_0(\lambda) - \int_0^t \{ i\lambda \|\nabla f(\boldsymbol{x}^\star)\| \varphi_s(\lambda) + \gamma(1 - \varphi_s(\lambda)) \} ds \\
&= -\int_0^t \{ i\lambda \|\nabla f(\boldsymbol{x}^\star)\| \varphi_s(\lambda) + \gamma(1 - \varphi_s(\lambda)) \} ds
\end{aligned}
\tag{80}
$$

Which means that

$$
i\lambda \|\nabla f(\boldsymbol{x}^\star)\| \varphi_s(\lambda) + \gamma(1 - \varphi_s(\lambda)) = 0; \quad \forall s > 0
\tag{81}
$$

i.e. $\varphi_s(\lambda) = \frac{1}{1 - i\|\nabla f(\boldsymbol{x}^\star)\|\lambda}$. So the invariant distribution of $\upsilon_t$ is $\mathcal{E}\left(\frac{\|\nabla f(\boldsymbol{x}^\star)\|}{\gamma}\right)$. As a result, the invariant distribution of $\mathbf{Y}_t$ is $\frac{\nabla f(\boldsymbol{x}^\star)}{\|\nabla f(\boldsymbol{x}^\star)\|} \cdot \mathcal{E}\left(\frac{\|\nabla f(\boldsymbol{x}^\star)\|}{\gamma}\right)$.

To show the mixing result, it is enough to prove the following fact,

$$\frac{1}{\|\boldsymbol{y}_0 - \boldsymbol{y}_1\|}\mathcal{W}_2(\mathcal{G}^t\delta_{\boldsymbol{y}_0}, \mathcal{G}^t\delta_{\boldsymbol{y}_1}) \longrightarrow 0; \quad t \to \infty \quad \forall \boldsymbol{y}_0 \neq \boldsymbol{y}_1. \tag{82}$$

Where $\delta_{\boldsymbol{y}}$ represents the Dirac measure at the point $\boldsymbol{y}$. Let $\mathbf{Y}_t^0$ and $\mathbf{Y}_t^1$ be the stochastic process generated by (78) with initial distribution $\delta_{\boldsymbol{y}_0}$ and $\delta_{\boldsymbol{y}_1}$ respectively. To give a bound for the Wasserstain-2 distance between $\mathbf{Y}_t^0$ and $\mathbf{Y}_t^1$, we compute the $L_2$ norm under the identical coupling, i.e., the two dynamics share all randomness in the Poisson process $\mathbf{N}_\gamma(s)$, $s \in [0, t]$. Owing to the property of the corresponding martingale problem of (78), we have

$$\begin{aligned}
0 &= \mathbb{E}\|\mathbf{Y}_t^0 - \mathbf{Y}_t^1\|^2 - \|\boldsymbol{y}_0 - \boldsymbol{y}_1\|^2 \\
&\quad - \int_0^t \mathbb{E}\left\{-\left(\nabla f(\boldsymbol{x}^\star)^\top, \nabla f(\boldsymbol{x}^\star)^\top\right)\begin{bmatrix} \mathbf{I} & -\mathbf{I} \\ -\mathbf{I} & \mathbf{I} \end{bmatrix}\begin{bmatrix} \mathbf{Y}_s^0 \\ \mathbf{Y}_s^1 \end{bmatrix} - \gamma\|\mathbf{Y}_s^0 - \mathbf{Y}_s^1\|^2\right\}ds \\
&= \mathbb{E}\|\mathbf{Y}_t^0 - \mathbf{Y}_t^0\|^2 - \|\boldsymbol{y}_0 - \boldsymbol{y}_1\|^2 + \gamma\int_0^1 \mathbb{E}\|\mathbf{Y}_s^0 - \mathbf{Y}_s^1\|^2 ds
\end{aligned} \tag{83}$$

Solving above integral equation we finally get $\mathbb{E}\|\mathbf{Y}_t^0 - \mathbf{Y}_t^1\|^2 = \|\boldsymbol{y}_0 - \boldsymbol{y}_1\|^2 e^{-\gamma t}$. Hence, the equation (82) has been proved.

$\square$

*Proof.* (**Proof of Theorem 3.4**)

This proof is basically modeled after the proof of Theorem 3.3. Therefore, for the sake of narrative simplicity, we will omit some details that overlap with the previous proofs. All symbols follow the meaning in the proof of Theorem 3.3 without special specification. Analogously, two steps are split to complete to proof.

**Step 1** Let $g$ belongs to $\mathcal{C}$ and let $\mathcal{D}_t^{(n)}$ denote the natural filtration of $\bar{\boldsymbol{v}}_t^{(n)}$. We aim to find the following martingale decomposition,

$$\forall t > 0, \quad g(\bar{\boldsymbol{v}}_t^{(n)}) - g(\bar{\boldsymbol{v}}_0^{(n)}) - \int_0^t \mathcal{J}g\left(\bar{\boldsymbol{v}}_s^{(n)}\right)ds = \mathcal{N}_t^{(n,g)} + \mathcal{T}_t^{(n,g)}. \tag{84}$$

Where $\mathcal{N}_t^{(n,g)}$ is a $\mathcal{D}_t^{(n)}$-martingale and $\mathcal{T}_t^{(n,g)}$ converges to zero in $L_1$. In fact, let

$$\begin{aligned}
\mathcal{N}_t^{(n,g)} &= \sum_{k=n+1}^{N(n,t,\eta^\beta)} \left\{g(\check{\boldsymbol{v}}_{k+1}) - g(\check{\boldsymbol{v}}_k) - \mathbb{E}[g(\check{\boldsymbol{v}}_{k+1}) - g(\check{\boldsymbol{v}}_k)|\mathcal{D}_k]\right\} \\
\mathcal{T}_t^{(n,g)} &= g(\bar{\boldsymbol{v}}_t^{(n)}) - g(\bar{\boldsymbol{v}}_{\underline{t}_n(\eta^\beta)}^{(n)}) - \int_{\underline{t}_n(\eta^\beta)}^t \mathcal{J}g(\bar{\boldsymbol{v}}_s^{(n)})ds \\
&\quad + \int_0^{\underline{t}_n(\eta^\beta)} \left(\mathcal{J}g\left(\bar{\boldsymbol{v}}_{\underline{s}_n(\eta^\beta)}^{(n)}\right) - \mathcal{J}g\left(\bar{\boldsymbol{v}}_s^{(n)}\right)\right)ds + \sum_{k=n}^{N(n,t,\eta^\beta)-1} \mathcal{T}_k^g.
\end{aligned} \tag{85}$$

Using the definition formula of $\bar{\boldsymbol{v}}_t^{(n)}$ (9) when $t \notin \mathcal{L}_n(T)$,

$$\begin{aligned}
\mathbb{E}\left\|\bar{\boldsymbol{v}}_t^{(n)} - \bar{\boldsymbol{v}}_{\underline{t}_n(\eta^\beta)}^{(n)}\right\|^2 &= (t - \underline{t}_n(\eta^\beta))^2 \mathbb{E}\|\boldsymbol{d}_{N(n,t,\eta^\beta)} - \xi_{N(n,t,\eta^\beta)}^{(2)}\|^2 \\
&\leq C\eta_n^{2\beta}.
\end{aligned} \tag{86}$$

This inequality combined with the Lipschitz continuity of g and its derivatives implies that the first three terms in the definition of $\mathcal{T}_t^{(n,g)}$ tend to 0 when $n \to \infty$. Further, by Lemma 11,

$$
\mathbb{E}\left| \sum_{k=n}^{N(n,t,\eta^\beta)-1} \mathcal{T}_k^g \right| \leq \sum_{k=n}^{N(n,t,\eta^\beta)-1} \eta_k^\beta \mathbb{E}\left| \frac{\mathcal{T}_k^g}{\eta_k^\beta} \right|
$$

$$
\leq \sup_{k \geq n} \mathbb{E}\left| \frac{\mathcal{T}_k^g}{\eta_k^\beta} \right| (t + \eta_n^\beta) \xrightarrow{n \to \infty} 0.
$$

(87)

**Step 2**  Suppose that there is a weakly convergent subsequence $\{\check{v}_{n_k}\}_{k=1}^\infty$ with limit distribution $\tilde{\nu}$. The definition of $M(n,t,\eta^\beta)$ and $\tilde{t}_n(\eta^\beta)$ can be extended intuitively from the first paragraph in the second step of the Theorem 3.3's proof.

Owing to the Prokhorov's theorem and Lemma 10, for any $T > 0$, there is a weakly convergent subsequence in $\left\{ \bar{v}_t^{M(n_k,T,\eta^\beta)} \right\}$. By the Theorem 1 in [20], we know that the weak limit of this sequence is a solution of the stochastic differential equation (78).And WLOG, we assume the sequence itself converges weakly to a solution $\bar{v}_t^{\tilde{\pi}^{(T)}}$ of (78) with initial distribution $\tilde{\pi}^{(T)}$ (the notations of $\tilde{\nu}$ and $\tilde{\pi}$ are independent with one in proof of Theorem 3.3). By Lemma 10, for any given $\epsilon > 0$, a compact set $K_\epsilon$ can be found such that $\sup_n \mathbb{P}(\check{v}_n \in K_\epsilon^c) \leq \epsilon$. Therefore, for all $T > 0$, $\tilde{\pi}^{(T)}(K_\epsilon) \geq 1 - \epsilon$.

Due to Lemma 12, a $T_\epsilon$ can be found such that

$$
\sup_{i \in K_\epsilon} \sup_{g \in \mathcal{C}} \left| \mathcal{G}^{T_\epsilon} g(\boldsymbol{x}) - \langle \nu^\star, g \rangle \right| \leq \epsilon.
$$

(88)

Where $\mathcal{G}$ is the Markov semigroup induced by the SDE (78). Because $\widetilde{(T_\epsilon)}_n(\eta^\beta)$ converges to $T_\epsilon$ when $n \to \infty$, we have $\check{v}_{n_k}\left( = \bar{v}_{\widetilde{(T_\epsilon)}_n(\eta^\beta)}^{M(n_k,T_\epsilon,\eta^\beta)} \right)$ converges weakly to the limit random variable of the sequence $\bar{v}_{T_\epsilon}^{M(n_k,T_\epsilon,\eta^\beta)}$,i.e., $\bar{v}_{T_\epsilon}^{\tilde{\pi}^{(T_\epsilon)}}$. On the other hand, by assumption, $\check{v}_{n_k}$ converges weakly to $\tilde{\nu}$. Thus $\bar{v}_{T_\epsilon}^{\tilde{\pi}^{(T_\epsilon)}} \sim \tilde{\nu}$. Combining all result we have obtained, the inequality corresponded to (55) can be derived. Consequently, we obtain $\tilde{\nu} = \nu^\star$. Finally, by the Prokhorov's theorem, $\check{v}_n$ convergence weakly to $\nu^\star$. Further, $\{\bar{v}_t^{(n)}\}$ converges weakly to the dynamics (78) with stationary distribution $\nu^\star$ as initialization. □

Before we start proving the Corollary 1, we need to use the following lemma.

**Lemma 13.** *Let $\{r_t\}$ be the sequence defined in Lemma 1. If a positive sequence $\{x_t\}$ satisfies:*

$$
x_{t+1} \leq (1 - r_t)x_t + o(r_t),
$$

(89)

*then $\lim_{t \to \infty} x_t \longrightarrow 0$.*

*Proof.* (**Proof of Lemma 13**) By the recursive inequality (89), for any given $\epsilon > 0$ there is a $t_0$ such that $\forall t > t_0$, $o(r_t) < \epsilon r_t$. Iterate the relation (89) and combine Lemma 1. Consequently, we have,

$$
x_t \leq \prod_{k=t_0}^{t}(1 - r_k)x_{t_0} + \sum_{k=t_0+1}^{t} \epsilon r_k \prod_{s=k+1}^{t}(1 - r_s) \longrightarrow 0 + \epsilon; \ \ t \to \infty.
$$

(90)

Because $\epsilon$ can be chosen arbitrarily, the final limit of $x_t$ is zero. □

*Proof.* (**Proof of Corollary 1**)

We prove the target conclusion in two steps. First we show that the mean of $\hat{u}_n$ converges to a constant non-zero vector. Next, we will see that the asymptotic variance of $\{\hat{u}_n\}$ is zero for $\beta \in \left(\frac{1}{2}, 1\right)$.

**Step 1** The first thing we need to do is to derive the recurrence relation for $\hat{u}_n$,

$$
\hat{u}_{n+1} = \mathcal{P}_{\boldsymbol{A}^\perp} \frac{\boldsymbol{x}_{(n+1)-} - \boldsymbol{x}^\star}{\eta_n^{1-\beta}} = \frac{\mathcal{P}_{\boldsymbol{A}^\perp}(\boldsymbol{x}_n - \boldsymbol{x}^\star - \eta_n \nabla f(\boldsymbol{x}_n) + \eta_n \xi_n)}{\eta_n^{1-\beta}}
$$

$$
= \left(\frac{\eta_{n-1}}{\eta_n}\right)^{1-\beta} \hat{u}_n - \eta_n^\beta \mathcal{P}_{\boldsymbol{A}^\perp} \nabla f(\boldsymbol{x}_n) + \eta_n^\beta \xi_n^{(1)}
$$

$$
= \hat{u}_n - \eta_n^\beta \mathcal{P}_{\boldsymbol{A}^\perp} \left\{ \nabla^2 f(\boldsymbol{x}^\star)(\boldsymbol{x}_n - \boldsymbol{u}_n) + [\nabla^2 f(\vartheta_n^{\boldsymbol{v}}) - \nabla^2 f(\boldsymbol{x}^\star)](\boldsymbol{x}_n - \boldsymbol{u}_n) \right\}
$$

$$
+ \left( \left(\frac{\eta_{n-1}}{\eta_n}\right)^{1-\beta} - 1 \right) \hat{u}_n + \eta_n^\beta \xi_n^{(1)}
$$

$$
- \eta_n^\beta \mathcal{P}_{\boldsymbol{A}^\perp} \left\{ \nabla^2 f(\boldsymbol{x}^\star)(\boldsymbol{u}_n - \boldsymbol{x}^\star) + [\nabla^2 f(\vartheta_n^{\boldsymbol{u}}) - \nabla^2 f(\boldsymbol{x}^\star)](\boldsymbol{u}_n - \boldsymbol{x}^\star) \right\}
$$

$$
= \left( \boldsymbol{I} - \eta_n \mathcal{P}_{\boldsymbol{A}^\perp} \nabla^2 f(\boldsymbol{x}^\star) \mathcal{P}_{\boldsymbol{A}^\perp} \right) \hat{u}_n - \eta_n \mathcal{P}_{\boldsymbol{A}^\perp} \nabla^2 f(\boldsymbol{x}^\star) \check{v}_n + \eta_n^\beta \xi_n^{(1)} \tag{91}
$$

$$
+ (\eta_n - \eta_n^\beta \eta_{n-1}^{1-\beta}) \mathcal{P}_{\boldsymbol{A}^\perp} \nabla^2 f(\boldsymbol{x}^\star) \hat{u}_n + (\eta_n - \eta_n^\beta \eta_{n-1}^{1-\beta}) \mathcal{P}_{\boldsymbol{A}^\perp} \nabla^2 f(\boldsymbol{x}^\star) \check{v}_n
$$

$$
- \left\{ \eta_n^\beta \mathcal{P}_{\boldsymbol{A}^\perp} \left( \nabla^2 f(\vartheta_n^{\boldsymbol{v}}) - \nabla^2 f(\boldsymbol{x}^\star) \right) (\boldsymbol{x}_n - \boldsymbol{u}_n) \right\}
$$

$$
- \left\{ \eta_n^\beta \mathcal{P}_{\boldsymbol{A}^\perp} [\nabla^2 f(\vartheta_n^{\boldsymbol{u}}) - \nabla^2 f(\boldsymbol{x}^\star)](\boldsymbol{u}_n - \boldsymbol{x}^\star) \right\}
$$

$$
+ \left( \left(\frac{\eta_{n-1}}{\eta_n}\right)^{1-\beta} - 1 \right) \hat{u}_n
$$

$$
=: \left( \boldsymbol{I} - \eta_n \mathcal{P}_{\boldsymbol{A}^\perp} \left( \nabla^2 f(\boldsymbol{x}^\star) - \frac{1-\beta}{\eta_0} \mathbb{1}_{\{\alpha=1\}} \boldsymbol{I}_d \right) \mathcal{P}_{\boldsymbol{A}^\perp} \right) \hat{u}_n
$$

$$
- \eta_n \mathcal{P}_{\boldsymbol{A}^\perp} \nabla^2 f(\boldsymbol{x}^\star) \check{v}_n + \eta_n^\beta \xi_n^{(1)} + \eta_n \mathcal{R}_n^{\boldsymbol{u}}
$$

Where $\vartheta_n^{\boldsymbol{u}}$ and $\vartheta_n^{\boldsymbol{v}}$ are two entrywise interpolation point between $\boldsymbol{u}_n$ and $\boldsymbol{x}_n$ ; $\boldsymbol{u}_n$ and $\boldsymbol{x}^\star$ respectively. And,

$$
\mathcal{R}_n^{\boldsymbol{u}} = \left( 1 - \left(\frac{\eta_{n-1}}{\eta_n}\right)^{1-\beta} \right) \mathcal{P}_{\boldsymbol{A}^\perp} \nabla^2 f(\boldsymbol{x}^\star)(\hat{u}_n + \check{v}_n) + \frac{1}{\eta_n} \left( \left(\frac{\eta_{n-1}}{\eta_n}\right)^{1-\beta} - 1 - \frac{1-\beta}{n} \mathbb{1}_{\{\alpha=1\}} \right) \hat{u}_n
$$

$$
- \left(\frac{\eta_{n-1}}{\eta_n}\right)^{1-\beta} \mathcal{P}_{\boldsymbol{A}^\perp} \left\{ (\nabla^2 f(\vartheta_n^{\boldsymbol{v}}) - \nabla^2 f(\boldsymbol{x}^\star)) \check{v}_n + (\nabla^2 f(\vartheta_n^{\boldsymbol{u}}) - \nabla^2 f(\boldsymbol{x}^\star)) \hat{u}_n \right\}
$$

$$
\tag{92}
$$

The properties of the step size sequence $\{\eta_n\}$ tell us that, when $\alpha < 1$,

$$
1 - \left(\frac{\eta_{n-1}}{\eta_n}\right)^{1-\beta} = 1 - \left( 1 + \frac{\eta_{n-1} - \eta_n}{\eta_n} \right)^{1-\beta}
$$

$$
= 1 - (1 + o(\eta_n))^{1-\beta} = (1-\beta)o(\eta_n) = o(1)\eta_n. \tag{93}
$$

And when $\alpha = 1$,

$$
1 + \frac{1-\beta}{n} - \left(\frac{n}{n-1}\right)^{1-\beta} = 1 + \frac{1-\beta}{n} - \left( 1 + \frac{1-\beta}{n} + \mathcal{O}(n^{-2}) \right) = o(\eta_n)
$$

The result can be used to guarantee the first line of (92) being $o(1)$ in $L_2$. By the assumption 3 and 1, $\nabla^2 f(\cdot)$ is Lipschitz continuous and uniformly bounded. Then for any $\delta \in (0,1)$, $\nabla^2 f(\cdot)$ is $\delta$-Höder continuous. Similar to the proof of the Lemmas 4 and 5, by leveraging the Taylor expansion for $\|\cdot\|^p$, $3 > p > 2$, we can deduce the following analogous bounds,

$$
\mathbb{E}\|\boldsymbol{u}_n - \boldsymbol{x}^\star\|^p \precsim \eta_n^{p(1-\beta)};
$$
$$
\mathbb{E}\|\boldsymbol{v}_n\|^p \precsim \eta_n^{p(1-\beta)} \tag{94}
$$

when $\beta \in [\frac{1}{2}, 1)$. Based on these preparations, take $\delta = p/2 - 1$ and use the Young's inequality,

$$
\begin{aligned}
\mathbb{E} \left\| (\nabla^2 f(\vartheta_n^{\boldsymbol{v}}) - \nabla^2 f(\boldsymbol{x}^\star)) \check{\boldsymbol{v}}_n \right\|^2 &\precsim \mathbb{E} \left\| \vartheta_n^{\boldsymbol{u}} - \boldsymbol{x}^\star \right\|^{2\delta} \left\| \check{\boldsymbol{v}}_n \right\|^2 \\
&\precsim \mathbb{E} \left( \left\| \boldsymbol{v}_n \right\|^{2\delta} + \left\| \boldsymbol{u}_n - \boldsymbol{x}^\star \right\|^{2\delta} \right) \left\| \check{\boldsymbol{v}}_n \right\|^2 \\
&= \frac{1}{\eta_{n-1}^{2(1-\beta)}} \mathbb{E} \left\| \boldsymbol{v}_n \right\|^p + \frac{1}{\eta_{n-1}^{2(1-\beta)}} \mathbb{E} \left\| \boldsymbol{u}_n - \boldsymbol{x}^\star \right\|^{2\delta} \left\| \boldsymbol{v}_n \right\|^2 \qquad (95) \\
&\precsim \frac{1}{\eta_{n-1}^{2(1-\beta)}} \left( \mathbb{E} \left\| \boldsymbol{v}_n \right\|^p + \mathbb{E} \left\| \boldsymbol{u}_n - \boldsymbol{x}^\star \right\|^p \right) \precsim \eta_n^{(p-2)(1-\beta)}.
\end{aligned}
$$

The same bound can be derived for $(\nabla^2 f(\vartheta_n^{\boldsymbol{u}}) - \nabla^2 f(\boldsymbol{x}^\star)) \check{\boldsymbol{u}}_n$. These two results enable the second line of (92) to be $o(1)$ in $L_2$. To simplify our writing, we denote $\boldsymbol{\nu} = -\frac{1}{\gamma} \nabla f(\boldsymbol{x}^\star)$ and $\boldsymbol{\mu} = \frac{1}{\gamma} \left( \mathcal{P}_{\boldsymbol{A}^\perp} \left( \nabla^2 f(\boldsymbol{x}^\star) - \frac{1-\beta}{\eta_0} \mathbb{1}_{\{\alpha=1\}} \mathbf{I}_d \right) \mathcal{P}_{\boldsymbol{A}^\perp} \right)^\dagger \mathcal{P}_{\boldsymbol{A}^\perp} \nabla^2 f(\boldsymbol{x}^\star) \nabla f(\boldsymbol{x}^\star)$. Taking the expectation on both sides of (91) yields

$$
\begin{aligned}
\mathbb{E} \hat{\boldsymbol{u}}_{n+1} = &\left( \mathbf{I} - \eta_n \left( \mathcal{P}_{\boldsymbol{A}^\perp} \nabla^2 f(\boldsymbol{x}^\star) - \frac{1-\beta}{\eta_0} \mathbb{1}_{\{\alpha=1\}} \mathbf{I}_d \right) \mathcal{P}_{\boldsymbol{A}^\perp} \right) \mathbb{E} \hat{\boldsymbol{u}}_n \\
&- \eta_n \mathcal{P}_{\boldsymbol{A}^\perp} \nabla^2 f(\boldsymbol{x}^\star) \mathbb{E} \check{\boldsymbol{v}}_n + \eta_n \mathbb{E} \mathcal{R}_n^{\boldsymbol{u}}
\end{aligned} \qquad (96)
$$

Subtract $\mathcal{P}_{\boldsymbol{A}^\perp} \nabla^2 f(\boldsymbol{x}^\star) \boldsymbol{\nu}$ from both sides of (96) and we have,

$$
\begin{aligned}
\mathbb{E} \hat{\boldsymbol{u}}_{n+1} - \boldsymbol{\mu} = &\left( \mathbf{I} - \eta_n \mathcal{P}_{\boldsymbol{A}^\perp} \left( \nabla^2 f(\boldsymbol{x}^\star) - \frac{1-\beta}{\eta_0} \mathbb{1}_{\{\alpha=1\}} \mathbf{I}_d \right) \mathcal{P}_{\boldsymbol{A}^\perp} \right) (\mathbb{E} \hat{\boldsymbol{u}}_n - \boldsymbol{\mu}) \\
&- \eta_n \left( \mathcal{P}_{\boldsymbol{A}^\perp} \nabla^2 f(\boldsymbol{x}^\star) (\mathbb{E}(\check{\boldsymbol{v}}_n - \boldsymbol{\nu}) + \mathbb{E} \mathcal{R}_n^{\boldsymbol{u}}) \right)
\end{aligned} \qquad (97)
$$

According to Theorem 3.4 we know $\|\mathbb{E} \check{\boldsymbol{v}}_n - \boldsymbol{\nu}\| = o(1)$. As a result of this and $\|\mathbb{E} \mathcal{R}_n^{\boldsymbol{u}}\| \leq \mathbb{E} \|\mathcal{R}_n^{\boldsymbol{u}}\| = o(1)$, the last term in the right hand of (97) is $o(\eta_n)$. Therefore, using Lemma 13, it holds that $\|\mathbb{E} \hat{\boldsymbol{u}}_n - \boldsymbol{\mu}\| = o(1)$, which means $\mathbb{E} \hat{\boldsymbol{u}}_n \xrightarrow{n \to \infty} \boldsymbol{\mu}$.

**Step 2** Before we calculate the asymptotic variance of $\hat{\boldsymbol{u}}_n$, consider the inner product $\left| \mathbb{E} \left\langle \hat{\boldsymbol{u}}_n - \mathbb{E} \hat{\boldsymbol{u}}_n, \nabla^2 f(\boldsymbol{x}^\star)(\check{\boldsymbol{v}}_n - \mathbb{E} \check{\boldsymbol{v}}_n) \right\rangle \right|$,

$$
\begin{aligned}
&\left| \mathbb{E} \left\langle \hat{\boldsymbol{u}}_{n+1} - \mathbb{E} \hat{\boldsymbol{u}}_{n+1}, \nabla^2 f(\boldsymbol{x}^\star)(\check{\boldsymbol{v}}_{n+1} - \mathbb{E} \check{\boldsymbol{v}}_{n+1}) \right\rangle \right| \\
=&(1 - \gamma\eta_n^\beta) \left| \mathbb{E} \left\langle \hat{\boldsymbol{u}}_{n+1} - \mathbb{E} \hat{\boldsymbol{u}}_{n+1}, \nabla^2 f(\boldsymbol{x}^\star)(\check{\boldsymbol{v}}_{(n+1)-} - \mathbb{E} \check{\boldsymbol{v}}_{n+1}) \right\rangle \right| + \gamma\eta_n^\beta \left| \mathbb{E} \left\langle \hat{\boldsymbol{u}}_{n+1} - \mathbb{E} \hat{\boldsymbol{u}}_{n+1}, \nabla^2 f(\boldsymbol{x}^\star) \mathbb{E} \check{\boldsymbol{v}}_{n+1} \right\rangle \right| \\
=&(1 - \gamma\eta_n^\beta) \left| \mathbb{E} \left\langle \hat{\boldsymbol{u}}_{n+1} - \mathbb{E} \hat{\boldsymbol{u}}_{n+1}, \nabla^2 f(\boldsymbol{x}^\star)(\check{\boldsymbol{v}}_{(n+1)-} - \mathbb{E} \check{\boldsymbol{v}}_{n+1}) \right\rangle \right| \\
=&(1 - \gamma\eta_n^\beta) \left| \mathbb{E} \left\langle \left( \mathbf{I} - \eta_n \mathcal{P}_{\boldsymbol{A}^\perp} \left( \nabla^2 f(\boldsymbol{x}^\star) - \frac{1-\beta}{\eta_0} \mathbb{1}_{\{\alpha=1\}} \mathbf{I}_d \right) \mathcal{P}_{\boldsymbol{A}^\perp} \right) (\hat{\boldsymbol{u}}_n - \mathbb{E} \hat{\boldsymbol{u}}_n) - \eta_n \mathcal{P}_{\boldsymbol{A}^\perp} \nabla^2 f(\boldsymbol{x}^\star)(\check{\boldsymbol{v}}_n - \mathbb{E} \check{\boldsymbol{v}}_n) \right. \right. \\
&\left. \left. + \eta_n^\beta \xi_n^{(1)} + \eta_n (\mathcal{R}_n^{\boldsymbol{u}} - \mathbb{E} \mathcal{R}_n^{\boldsymbol{u}}), \nabla^2 f(\boldsymbol{x}^\star) \left\{ (\check{\boldsymbol{v}}_n - \mathbb{E} \check{\boldsymbol{v}}_n) + \eta_n^\beta \xi_n^{(2)} + \eta_n^\beta (\mathcal{R}_n^{\boldsymbol{v}} - \mathbb{E} \mathcal{R}_n^{\boldsymbol{v}}) \right\} \right\rangle \right| \\
\leq&(1 - \gamma\eta_n^\beta) \left| \mathbb{E} \left\langle \hat{\boldsymbol{u}}_n - \mathbb{E} \hat{\boldsymbol{u}}_n, \nabla^2 f(\boldsymbol{x}^\star)(\check{\boldsymbol{v}}_n - \mathbb{E} \check{\boldsymbol{v}}_n) \right\rangle \right| + \eta_n^\beta \left| \mathbb{E} \left\langle \hat{\boldsymbol{u}}_n - \mathbb{E} \hat{\boldsymbol{u}}_n, \nabla^2 f(\boldsymbol{x}^\star)(\mathcal{R}_n^{\boldsymbol{v}} - \mathbb{E} \mathcal{R}_n^{\boldsymbol{v}}) \right\rangle \right| + \mathcal{O}(\eta_n) \\
&\hspace{15cm} (98)
\end{aligned}
$$

From the fact $\mathbb{E}\|\mathcal{R}_n^{\boldsymbol{v}}\|^2 = o(1)$, we have

$$
\left| \mathbb{E} \left\langle \hat{\boldsymbol{u}}_{n+1} - \mathbb{E} \hat{\boldsymbol{u}}_{n+1}, \nabla^2 f(\boldsymbol{x}^\star)(\check{\boldsymbol{v}}_{n+1} - \mathbb{E} \check{\boldsymbol{v}}_{n+1}) \right\rangle \right| \leq (1 - \gamma\eta_n^\beta) \left| \mathbb{E} \left\langle \hat{\boldsymbol{u}}_n - \mathbb{E} \hat{\boldsymbol{u}}_n, \nabla^2 f(\boldsymbol{x}^\star)(\check{\boldsymbol{v}}_n - \mathbb{E} \check{\boldsymbol{v}}_n) \right\rangle \right| + o(\eta_n^\beta) \qquad (99)
$$

We can obtain from Lemma 13 that $\left| \mathbb{E} \left\langle \hat{\boldsymbol{u}}_n - \mathbb{E} \hat{\boldsymbol{u}}_n, \nabla^2 f(\boldsymbol{x}^\star)(\check{\boldsymbol{v}}_n - \mathbb{E} \check{\boldsymbol{v}}_n) \right\rangle \right| \xrightarrow{n \to \infty} 0$.

Back to the main result's proof, we can write down the recursive rule for the variance of $\hat{\boldsymbol{u}}_n$,

$$
\begin{aligned}
&\mathbb{E} \left\| \hat{\boldsymbol{u}}_{n+1} - \mathbb{E} \hat{\boldsymbol{u}}_{n+1} \right\|^2 \\
=&\mathbb{E} \left\| \left( \mathbf{I} - \eta_n \mathcal{P}_{\boldsymbol{A}^\perp} \left( \nabla^2 f(\boldsymbol{x}^\star) - \frac{1-\beta}{\eta_0} \mathbb{1}_{\{\alpha=1\}} \mathbf{I}_d \right) \mathcal{P}_{\boldsymbol{A}^\perp} \right) (\hat{\boldsymbol{u}}_n - \mathbb{E} \hat{\boldsymbol{u}}_n) \right. \\
&\left. - \eta_n \mathcal{P}_{\boldsymbol{A}^\perp} \nabla^2 f(\boldsymbol{x}^\star)(\check{\boldsymbol{v}}_n - \mathbb{E} \check{\boldsymbol{v}}_n) + \eta_n^\beta \xi_n^{(1)} + \eta_n (\mathcal{R}_n^{\boldsymbol{u}} - \mathbb{E} \mathcal{R}_n^{\boldsymbol{u}}) \right\|^2 \\
\leq&(1 - \mu\eta_n) \mathbb{E} \left\| \hat{\boldsymbol{u}}_n - \mathbb{E} \hat{\boldsymbol{u}}_n \right\|^2 - \eta_n \mathbb{E} \left\langle \hat{\boldsymbol{u}}_n - \mathbb{E} \hat{\boldsymbol{u}}_n, \nabla^2 f(\boldsymbol{x}^\star)(\check{\boldsymbol{v}}_n - \mathbb{E} \check{\boldsymbol{v}}_n) \right\rangle + \eta_n^{2\beta} \mathbb{E} \left\langle \xi_n^{(1)}, \xi_n^{(2)} \right\rangle + o(\eta_n) \\
\leq&(1 - \mu\eta_n) \mathbb{E} \left\| \hat{\boldsymbol{u}}_n - \mathbb{E} \hat{\boldsymbol{u}}_n \right\|^2 + o(\eta_n).
\end{aligned}
$$
$$\hspace{15cm} (100)$$

Where the last equation follows from the diminish correlation we derived just now and the pre-condition $\beta > \frac{1}{2}$. Finally, the Lemma is completed from Lemma 13 and the fact $\mathbb{E} \|\hat{\boldsymbol{u}}_n - \boldsymbol{\mu}\|^2 = \mathbb{E} \|\hat{\boldsymbol{u}}_n - \mathbb{E}\hat{\boldsymbol{u}}_n\|^2 + \|\mathbb{E}\hat{\boldsymbol{u}}_n - \boldsymbol{\mu}\|^2 \to 0$. $\qquad\square$

## D  Experimental Details

In this section, we present the experimental details and the complete results on three different datasets.

### D.1  Datasets

We have introduced two datasets in Section 4 in the FL setting. We will restate them and add a new dataset for the general linearly constrained problem.

**IID** There are $K$ clients and the sample $(\boldsymbol{x}_k, z_k)$ on the $k$-th client is modeled as $\boldsymbol{x}_k \sim \mathcal{N}(\nu_k, \Lambda)$ and $z_k = \mathrm{argmax}(\mathrm{softmax}(\boldsymbol{W}_k \boldsymbol{x}_k + \boldsymbol{b}_k))$ where $\Lambda \in \mathbb{R}^{d \times d}$ is diagonal with the entry $(j, j)$ equal to $j^{-1.2}$, all the clients share the same $\boldsymbol{W}_k \in \mathbb{R}^{C \times d}$ and $\boldsymbol{b}_k \in \mathbb{R}^C$ and their entries are modeled as $\mathcal{N}(0, 1)$. We set $K = 100$, $d = 60$ and $C = 10$. For this dataset, there is no heterogeneity between the optimal local parameters. The heterogeneity is all from the diversity of the distributions of $\boldsymbol{x}_k$. For each client, the sample size is around $100$.

**Synthetic** $(a, b)$ There are $K$ clients and the sample $(\boldsymbol{x}_k, z_k)$ on the $k$-th client is modeled as $\boldsymbol{x}_k \sim \mathcal{N}(\nu_k, \Lambda)$ and $z_k = \mathrm{argmax}(\mathrm{softmax}(\boldsymbol{W}_k \boldsymbol{x}_k + \boldsymbol{b}_k))$ where $\Lambda \in \mathbb{R}^{d \times d}$ is diagonal with the entry $(j, j)$ equal to $j^{-1.2}$, each entry of $\boldsymbol{W}_k$ and $\boldsymbol{b}_k$ is modeled as $\mathcal{N}(\mu_k, 1)$ with $\mu_k \sim \mathcal{N}(0, a)$ and $\nu_k \sim \mathcal{N}(\zeta_k, \boldsymbol{I})$ with $\zeta_k \sim \mathcal{N}(\boldsymbol{0}, b\boldsymbol{I}_d)$. We set $K = 20$, $d = 10$ and $C = 5$. $a$ controls how many local models differ from each other and $b$ controls how much the local data for each client differs from that of other clients. They are the two sources of heterogeneity. For each client, the sample size is around $50$. In this paper, we let $a = b = 1$.

The last dataset aims to solve the general linearly constrained problem (1).

**Lincons** The data are generated by the same way in IID. Since in IID, all the clients share the same $\boldsymbol{W}_k$ and $\boldsymbol{b}_k$, we can combine all the samples and obtain the dataset Lincons. Then we generate the matrix $\boldsymbol{A} \in \mathbb{R}^{610 \times 400}$ whose entries are independent and modeled as $\mathcal{N}(0, 1)$.

For all the three datasets, the loss function is defined as the sum of cross entropy loss and $\ell_2^2$ regularization.

### D.2  Parameters

For all the datasets, the mini-batch size is $4$. As for the probability $p_n$, we reparameterize it as $p_0 n^{-\alpha\beta}$ with $p_0 < 1$. The value of $\alpha$ is from $\{1, 0.8, 0.6\}$ and the value of $\beta$ is from $\{0, 0.2, 0.4, 0.6, 0.8\}$. For $\beta = 0$, we set $p_0 = 0.2$; for $\beta > 0$, we set $p_0 = 0.5$. And we run gradient descent $1000$ steps to obtain the value of $\boldsymbol{x}^\star$.

**IID** The parameter of $\ell_2^2$ regularization is $0.005$. For $\alpha = 1$, we set $\eta_0 = 200$; for $\alpha = 0.8$, we set $\eta_0 = 40$ in Appendices D.3 and D.4 and set $\eta_0 = 200$ in Appendices D.5 and D.6; for $\alpha = 0.6$, we set $\eta_0 = 20$.

**Synthetic** $(1, 1)$ The parameter of $\ell_2^2$ regularization is $0.5$. For $\alpha = 1$, we set $\eta_0 = 1$; for $\alpha = 0.8$, we set $\eta_0 = 0.3$; for $\alpha = 0.6$, we set $\eta_0 = 0.1$.

**Lincons** The parameter of $\ell_2^2$ regularization is $0.05$. For $\alpha = 1$, we set $\eta_0 = 8$; for $\alpha = 0.8$, we set $\eta_0 = 2$; for $\alpha = 0.6$, we set $\eta_0 = 0.8$.

### D.3  Convergence Rates

We plot the log-log scale graphs of averaged MSEs over $5$ repetitions on IID vs iterations in Figure 1 and the log-log scale graphs of averaged MSEs over $10$ repetitions on IID and Lincons vs

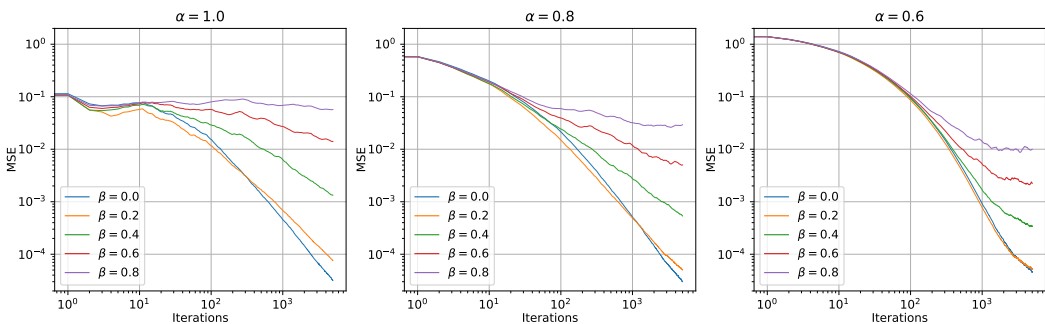

Figure 4: The log-log scale graphs of averaged MSE on `Synthetic` $(1,1)$ over 10 repetitions vs iterations.

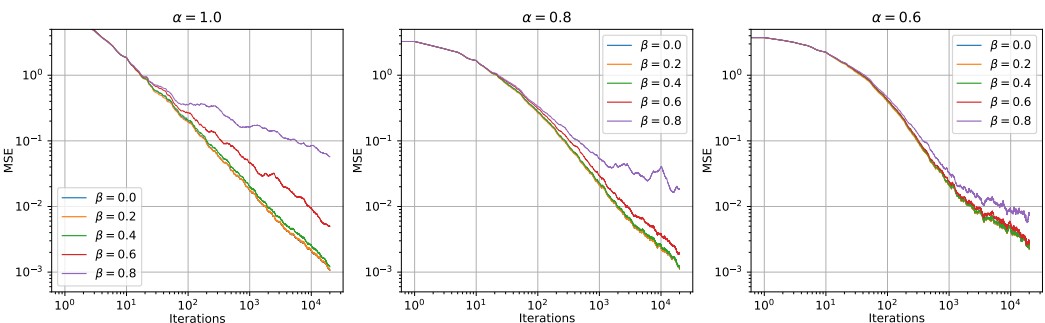

Figure 5: The log-log scale graphs of averaged MSE on `Lincons` over 10 repetitions vs iterations.

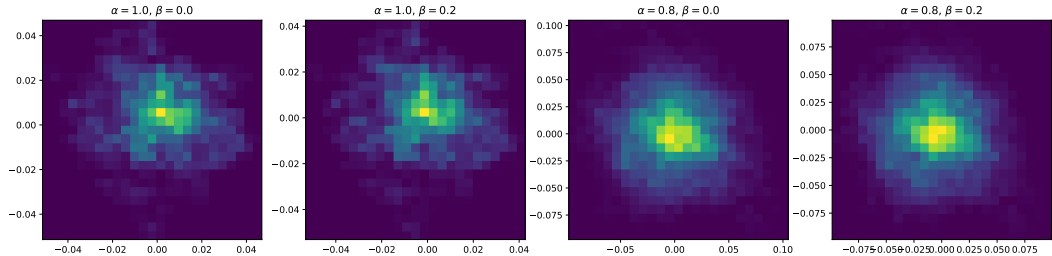

Figure 6: The heatmaps of $\check{\boldsymbol{u}}_n$ across two orthogonal directions over 100 repetition on `IID`.

iterations in Figures 4 and 5. When $\beta < 1/2$, the value of $\beta$ hardly affects the convergence rate; when $\beta > 1/2$, both larger $\beta$ and smaller $\alpha$ lead to a slower convergence rate. This is consistent to the result of Theorem 3.1.

### D.4 Heatmaps

We plot the heatmaps for $\alpha = 1, 0.8$ and $\beta = 0, 0.2$. The results for the three datasets are shown in Figures 6, 7 and 8. For `IID`, we run 2000 steps of LPSA over 100 repetitions and pick up the last 200 iterates. For `Synthetic` $(1,1)$, we run 3000 steps of LPSA over 100 repetitions for $\alpha = 1$ and 4000 steps for $\alpha = 0.8$. Then we pick up the last 800 iterates to plot the heatmap. For `Lincons`, we run 2000 steps of LPSA over 100 repetitions and pick up the last 800 iterates. All the heatmaps show that the cells near the origin have lighter colors, which agrees with Theorem 3.3.

### D.5 Trajectories

For $\alpha = 1, 0.8$ and $\beta = 0.6, 0.8$, we plot the trajectories of $\check{\boldsymbol{v}}_n$ along two random directions $\boldsymbol{e}_1$ and $\boldsymbol{e}_2$ vs accumulation of $\eta_n$ in Figures 9, 10 and 11. Note that the directions vectors $\boldsymbol{e}_1$ and $\boldsymbol{e}_2$ are distinct

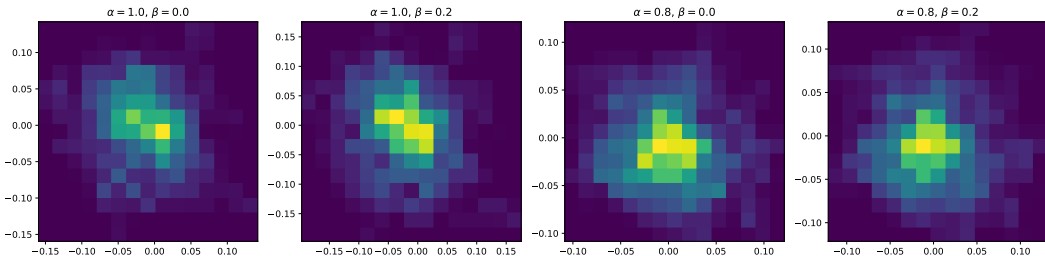

Figure 7: The heatmaps of $\check{u}_n$ across two orthogonal directions over 100 repetition on `Synthetic` $(1, 1)$.

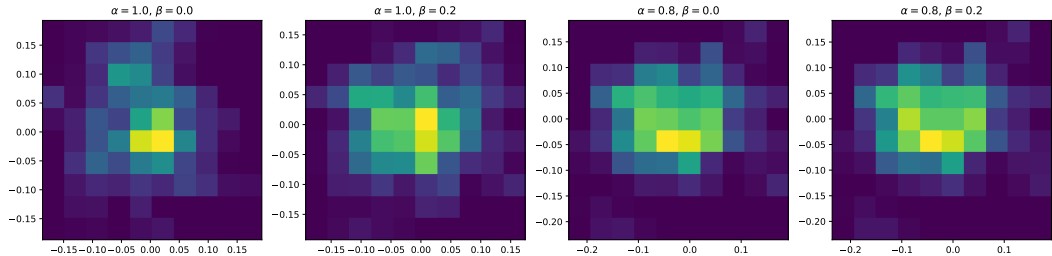

Figure 8: The heatmaps of $\check{u}_n$ across two orthogonal directions over 100 repetition on `Lincons`.

for different datasets. The value of the horizontal coordinate is $\sum_{i=s}^{n} \eta_i^{\beta}$, where $s$ is the start point of the trajectory that aims to eliminate the irregular behavior in the early stage of the optimization process. In Figure 9, we run 20000 steps of LPSA and set $s = 2000$; in Figure 10, we run 50000 steps of LPSA and set $s = 1000$; in Figure 11, we run 5000 steps of LPSA and set $s = 2000$.

We take Figure 10 as an example. Observe that the trajectories in Figure 10 come in a jagged manner and the peak value does not vanish or explode. This is because we have chosen a suitable rescaled version $\check{v}_n$ of $v_n$ and such a behavior can be captured by Theorem 3.4. The same discussion also applies for Figure 11. As for Figure 9, where the dimension of $v_n$ is 61000, the rate of the weak convergence mentioned in Theorem 3.4 is much slower than the low-dimensional counterparts depicted in Figures 10 and 11, where the dimension of $v_n$ is of hundreds or around 1000. When the number of iterations is not so large, the influence of gradient noise can not be ignored. As a result, the trajectories in Figure 9 keep fluctuating a lot and are not so smooth as those in Figures 10 and 11.

### D.6 Bias

For $\alpha = 1, 0.8$ and $\beta = 0.6, 0.8$, we plot the trajectories of $\hat{u}_n$ along two random directions $e_1$ and $e_2$ (or $e_3$) for three datasets in Figures 12, 13 and 14 to show the asymptotic biased of $\hat{u}_n$. Note that the directions vectors $e_1$ and $e_2$ are distinct for different datasets. The scale of coordinate axis in Figure 13 is different from that in Figure 3. This is due to that we omit the influence of $\eta_0$ in Figure 3, whose value does not affect the shape of the trajectories. Moreover, we choose a different direction $e_3$ in Figure 13 instead of $e_2$ in Figure 3 for a better illustration. We observe that although some trajectories have not converged yet, they stay away from the blue horizontal dashed line, which denotes the value 0. This verifies the result of Corollary 1.

### D.7 Convergence Rates in terms of the Number of Projections

Recall that in Section 3.1 and Appendix D.3, we establish the convergence rates in terms of the number of iterations and provide the log-log scale graphs of averages MSEs vs. the number of iterations. To better capture the influence of projections, in this subsection, we consider the convergence rates in terms of the number of projections and plot the the log-log scale graphs of averages MSEs vs. the number of projections.

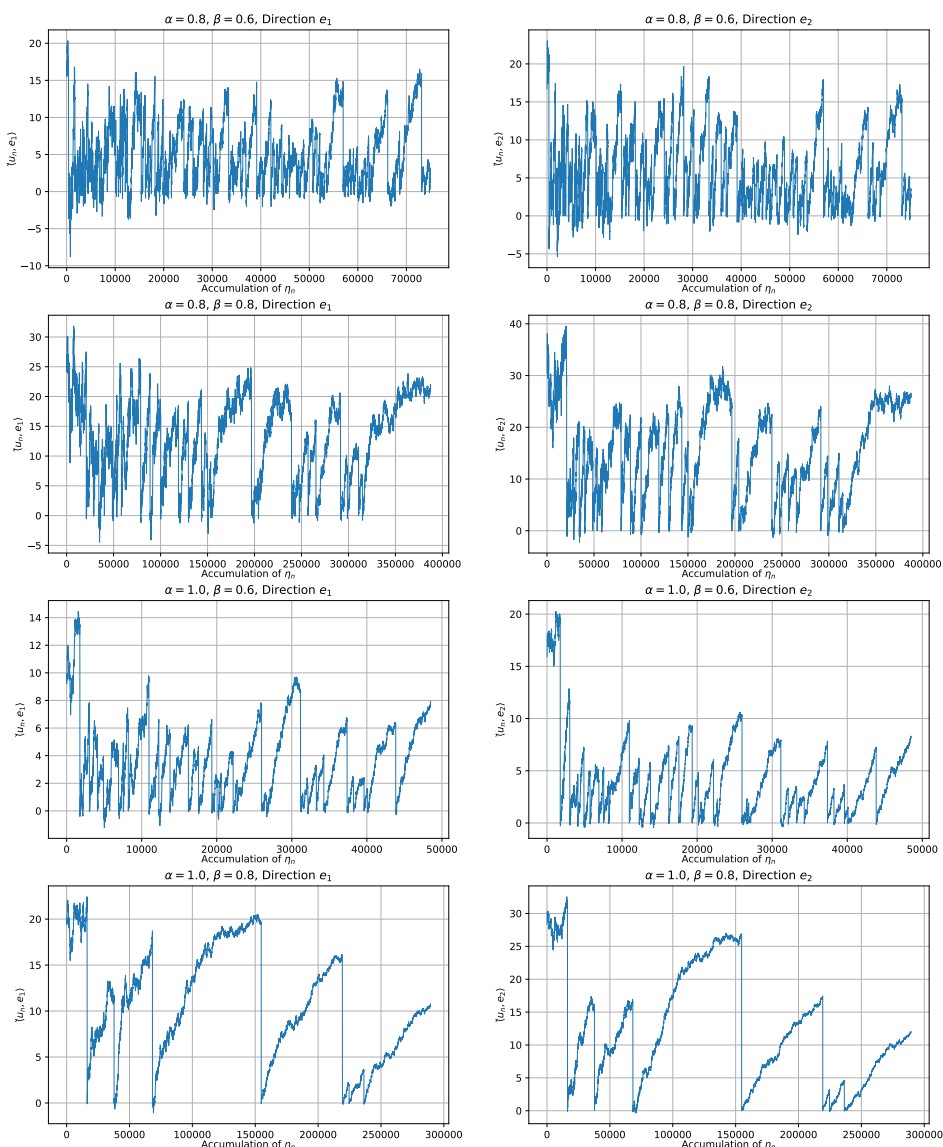

Figure 9: Trajectories of $\check{\boldsymbol{v}}_n$ along two random directions vs accumulation of $\eta_n$ on `IID`.

In our method, at the $n$-th iteration, the projection probability is $p_n = \min\{\eta_n^\beta, 1\} = \Theta(n^{-\alpha\beta})$. As a result, after $n$ steps of iterations, the number of projections $m$ should be of the order $\Theta(n^{1-\alpha\beta})$. Suppose that after $m$ steps of projections, we obtain the variable $\boldsymbol{x}_m$ and $\boldsymbol{u}_m = \mathcal{P}_{\boldsymbol{A}^\perp}(\boldsymbol{x}_m)$. By Theorem 3.1, we have $\mathbb{E}\left\|\boldsymbol{u}_m - \boldsymbol{x}^\star\right\|^2 = \mathcal{O}\left(m^{-\frac{\alpha\min\{1,2-2\beta\}}{1-\alpha\beta}}\right)$. For $0 \le \beta < 0.5$, the rate is of the order $\mathcal{O}\left(m^{-\frac{\alpha}{1-\alpha\beta}}\right)$ and a larger $\beta$ leads to a faster rate. For $0.5 \le \beta < 1$, the rate is of the order $\mathcal{O}\left(m^{-\frac{2\alpha(1-\beta)}{1-\alpha\beta}}\right) = \mathcal{O}\left(m^{-2\left(1-\frac{1-\alpha}{1-\alpha\beta}\right)}.\right)$ and a larger $\beta$ leads to a slower rate.

To conclude, if we only focus on the complexity of projection steps and ignore the cost of gradient computation, $\beta = 0.5$ is the best choice.

Then we plot the log-log scale graphs of averages MSEs vs. the number of projections on two datasets `IID` and `Lincons` over 5 repetitions in Figures 15 and 16.

For `IID`, the value of $\alpha$ is from $\{1.0, 0.8, 0.6\}$ and the value of $\beta$ is from $\{0, 0.2, 0.4, 0.5, 0.6\}$. When $\beta = 0, 0.2$, we run 10000 steps pf LPSA; when $(\alpha, \beta) = (0.6, 0.4)$, we run 20000 steps pf LPSA;

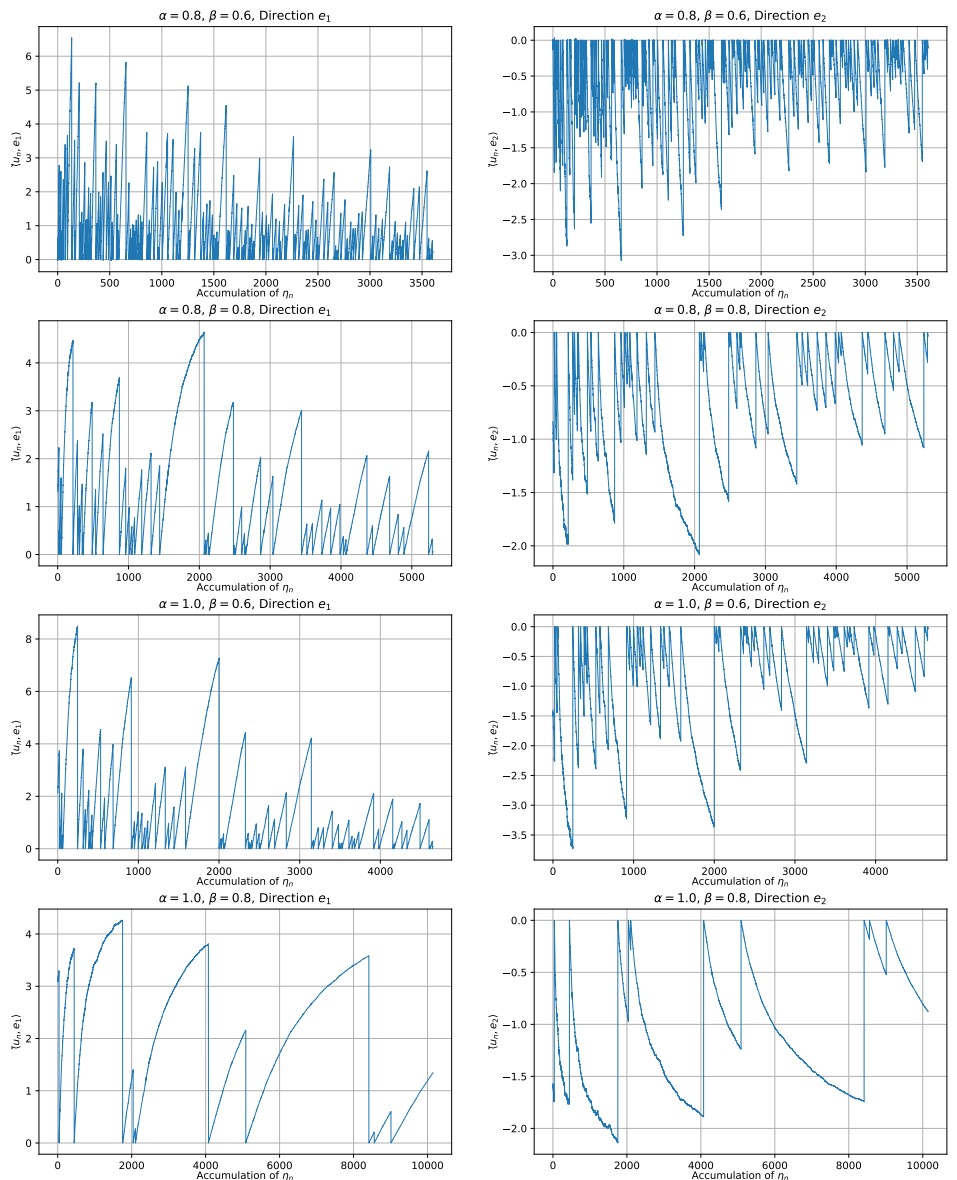

Figure 10: Trajectories of $\check{v}_n$ along two random directions vs accumulation of $\eta_n$ on `Synthetic` $(1, 1)$.

when $(\alpha, \beta) = (0.8, 0.4)$, we run 30000 steps pf LPSA; when $\beta = 0.5, 0.6$ or $(\alpha, \beta) = (1.0, 0.4)$, we run 50000 steps pf LPSA.

For `Lincons`, the value of $\alpha$ is from $\{1.0, 0.8, 0.6\}$ and the value of $\beta$ is from $\{0, 0.2, 0.4, 0.6, 0.8\}$. When $\beta = 0, 0.2$, we run 10000 steps pf LPSA; when $\beta = 0.4$ or $(\alpha, \beta) = (0.6, 0.6)$, we run 50000 steps pf LPSA; when $(\alpha, \beta) = (1.0, 0.6), (0.8, 0.6)$, we run 100000 steps pf LPSA; when $(\alpha, \beta) = (0.6, 0.8)$, we run 500000 steps pf LPSA; when $(\alpha, \beta) = (1.0, 0.8), (1.0, 0.6)$, we run 1000000 steps pf LPSA.

We find that in both Figures 15 and 16, when $\beta$ is closer to $0.5$, the lines of convergence rates are steeper. This is consistent with our analysis above. It is worth noting that for a fixed number of iterations, a larger $\beta$ implies a smaller number of projections. As a result, for $\beta$ larger than $0.5$, the interval between two adjacent projections is pretty large and to get a target number of projections (e.g., 1000), the number of iterations can be undesirable. To reduce the computational cost, we take a predetermined number of iterations, so the lines corresponding the larger $\beta$ can be shorter than others.

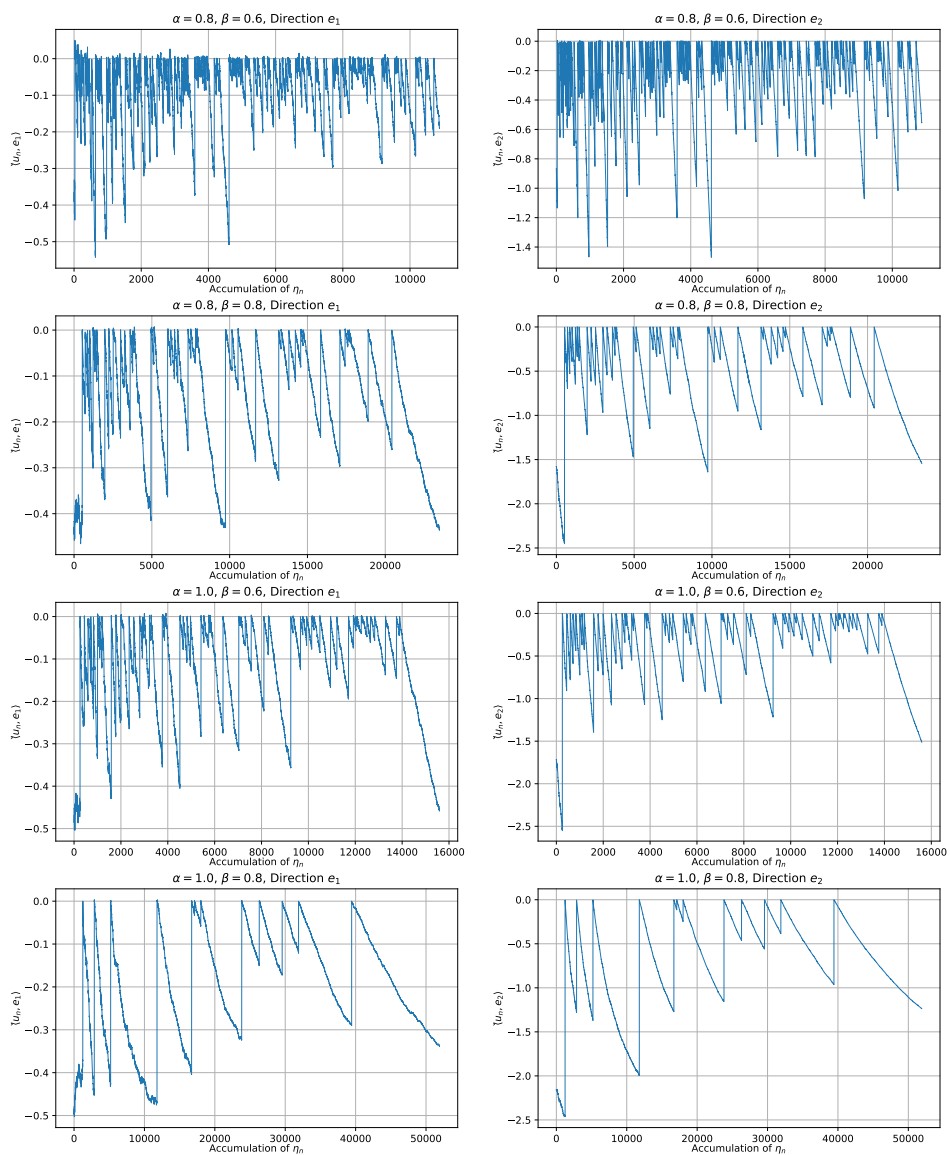

Figure 11: Trajectories of $\check{v}_n$ along two random directions vs accumulation of $\eta_n$ on `Lincons`.

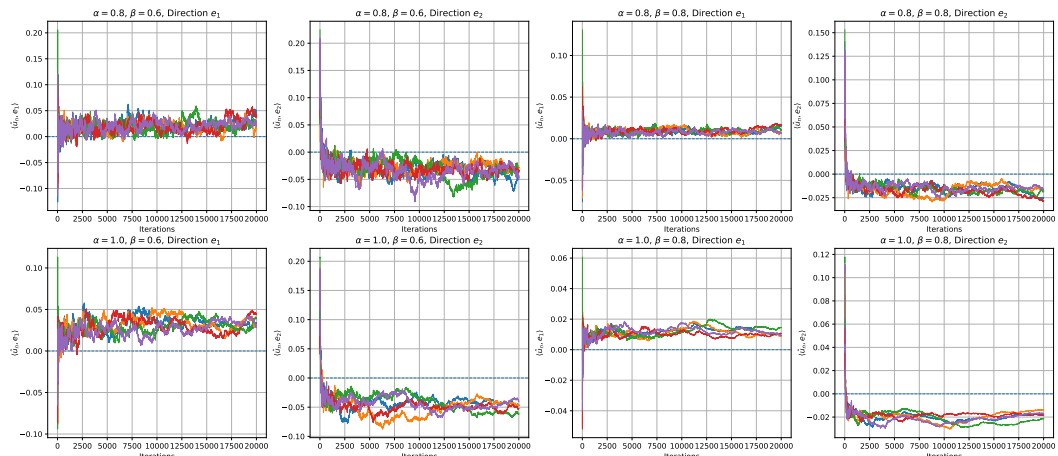

Figure 12: Trajectories of $\hat{\boldsymbol{u}}_n$ along two random directions over 5 repetitions on IID.

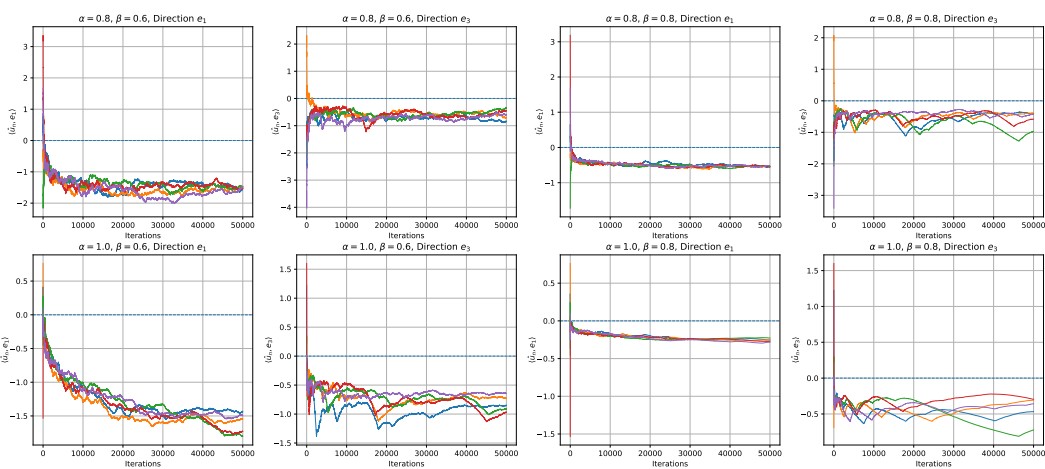

Figure 13: Trajectories of $\hat{\boldsymbol{u}}_n$ along two random directions over 5 repetitions on Synthetic $(1, 1)$.

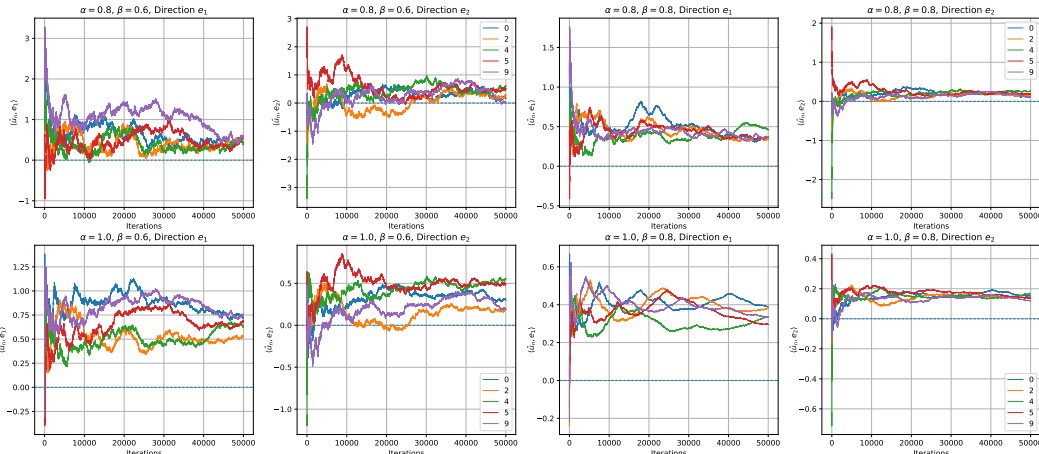

Figure 14: Trajectories of $\hat{\boldsymbol{u}}_n$ along two random directions over 5 repetitions on Lincons.

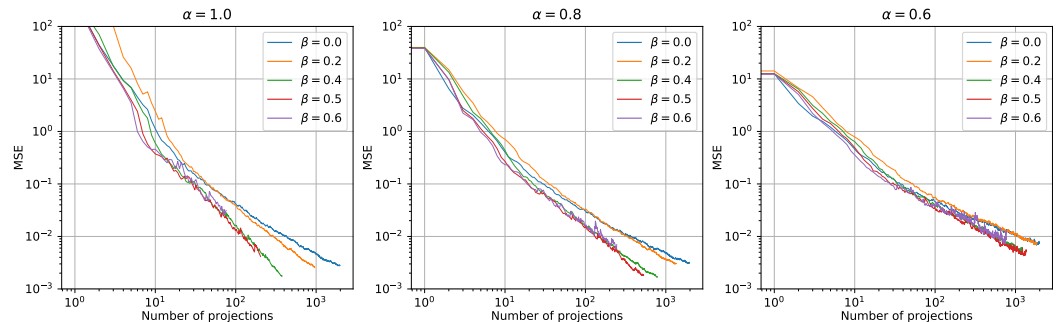

Figure 15: The log-log scale graphs of averaged MSE over 5 repetitions on `IID` vs. the number of projections.

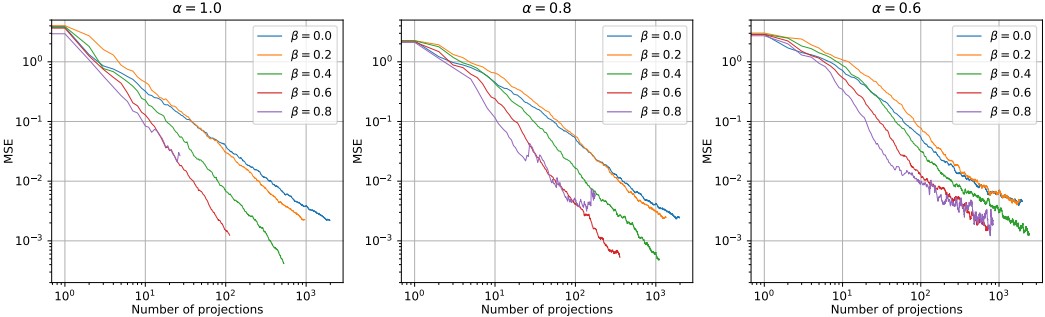

Figure 16: The log-log scale graphs of averaged MSE over 5 repetitions on `Lincons` vs. the number of projections.