# OpenReview forum: "Asymptotic Behaviors of Projected Stochastic Approximation: A Jump Diffusion Perspective"
_NeurIPS.cc/2022/Conference — NeurIPS 2022 Accept_

### Official Review · Reviewer_ikRt · 2022-06-28

**Rating:** 8
**Confidence:** 3
**Soundness:** 4 excellent
**Presentation:** 4 excellent
**Contribution:** 3 good

**Summary:**

This paper analyzes the performance of projected SGD in solving constrained optimization problems. It shows that a bias-variance tradeoff appears in the choice of the frequency of the projection. This bias-variance tradeoff is shown is two ways:
- a non-asymptotic analysis
- an asymptotic description with continuous-time stochastic processes.
Understanding this bias-variance tradeoff enables to use the optimal communication rate in federated learning.

**Questions:**

Minor comments:
(*) l.782: Can you explain "entry-wise linear interpolation"?
(*) L.780, 4th equality: why is there no more projection on the last term?
(*) eq. (16): should it be $v_{n+1}$ on the r.h.s.?
(*) l.58: did you mean $p_n$ instead of $\beta_n$?
(*) Could you explain (or give a ref) for the sentence l.185-186?
(*) l.232: should it be $-Y_{t-}$ instead of $-Y_{t}$?

**Limitations:**

I have none.

**Strengths And Weaknesses:**

The continuous-time perspective provides an elegant perspective on the choice of the projection frequency: either we have convergence to a diffusion process (variance dominates) or to a process with Poisson jumps (bias dominates).
In general, this paper is well-written, elegant and rigorous.
I do not master all the stochastic approximation results used in this paper. However, the authors introduce well the necessary tools, and provide all necessary references.
It seems that the techniques used in this paper could be used to understand other stochastic approximation algorithms.

---

> ### Author Response · Authors · 2022-08-01
> **Thank you very much for appreciating our work.**
>
> Thank you very much for appreciating our work, here is our response to your review comments.
>
> > *Can you explain "entry-wise linear interpolation"?*
>
> We are very sorry that there is a certain amount of abuse of notation here. What we essentially want to express is that by the mean value theorem, $\nabla^2 f(\theta_n)(u_n - x^*) = \nabla f(x^*) - \nabla f(u_n)$. In fact, $\nabla^2 f(\theta_n)$ is not a Hessian matrix on some point in $\mathbb{R}^d$. Actually, by using the mean value theorems for integrals, $\nabla^2 f(\theta_n) = \int^1_0 \nabla^2 f((1-t)x^* + tu_n)dt$. We have revised this area in the rebuttal revision and the proofs have been adjusted accordingly. Please check eq 30, 31 and the proof of Lemma 6.
>
> > *Could you explain (or give a ref) for the sentence l.185-186?*
>
> Please refer to Lemma 1 in [CZ Wei]. Let
> $ G = H = -\mathcal{P} _ { \mathbf{A} ^ \bot } \left( \nabla ^ 2 f ( x ^ * ) - ( 2 \eta _ 0 ) ^ {-1} \mathrm{1} _ { \alpha = 1 } \mathbf{I} _ d \right) \mathcal{P} _ { \mathbf{A} ^ \bot }$
> and
> $ Y = - \mathcal{P} _ { \mathbf{A} ^ \bot } \Sigma ( x ^ * ) \mathcal{P} _ { \mathbf{A} ^ \bot } $.
>
> We can check that in the subspace $\mathbf{A}^\bot$, all eigenvalues of $G,~H$ and $Y$ is negative.
> Then the matrix equation below line 184 has a unique solution on $\mathbf{A}^\bot$,
> $ \tilde{ \Sigma } = \int ^ \infty _ 0 \exp ( G t ) \mathcal{P} _ { \mathbf{A} ^ \bot} \Sigma ( x ^ * ) \mathcal{P} _ { \mathbf{A} ^ \bot } \exp ( Gt ) d t$.
> We also add the reference into the revised version.
>
> [CZ Wei] Multivariate adaptive stochastic approximation
>
> > *L780, eq(16), L58, L232*
>
> We apologise for these typos which have affected your reading. We have fixed them all in the revision.

---

> > ### Comment · Reviewer_ikRt · 2022-08-09
> > **Thank you**
> >
> > Thank you for these details / corrections.

---

### Official Review · Reviewer_L3VQ · 2022-07-08

**Rating:** 6
**Confidence:** 2
**Soundness:** 3 good
**Presentation:** 2 fair
**Contribution:** 3 good

**Summary:**

This paper proposes a jump diffusion perspective for a loopless variant of stochastic approximation schemes, essentially replacing the loop-length ratios by flipping a coin at each time step and performing projections accordingly. The main result is that there is a phase transition depending on the parameter , which controls the projection rate. The authors prove that in the frequent projection regime (), the asymptotic behavior of the algorithm is close to a diffusion, while in the occasional projection regime the asymptotic behavior goes to a discontinuous jump diffusion.


**Questions:**

Since I'm not an expert in the field, I find the constant drift outcome to be counterintuitive. Usually in stochastic approximation, the algorithms converge to either the gradient flow or, in the stochastic case, Langevin diffusion. Is there any intuition why we should expect a constant drift for both the frequent and occasionally projection regimes?


**Limitations:**

In my opinion, the most significant limitation of the work is the assumption of strong convexity; in the non-federated setting, we know that strong convexity is not necessary; see [Kushner-Yin]. It would therefore be great if the authors can point out potential avenues to improving/relaxing these restrictive assumptions.

[Kushner-Yin] Stochastic Approximation and Recursive Algorithms and Applications


**Strengths And Weaknesses:**


Strength: The analysis technique is fairly straightforward: The proof goes by decomposing the target sequences into a first-order term and then high-order terms that vanish in the long run. This technique is standard in stochastic approximation but seems adequate for the authors' purposes.

Weakness: The analysis relies heavily on strong convexity and does not seem to easily generalize to the non-strongly convex case; the latter, however, is of higher interest.

---

> ### Author Response · Authors · 2022-08-01
> **Thanks for your feedback and constructive advice 1/2**
>
> Thanks for your feedback and constructive advice.
> We list our response to your comments / questions below.
>
> > *The proof goes by decomposing the target sequences ... in stochastic approximation but seems adequate for the authors' purposes.*
>
> As you say, in the analysis of the rescaled sequence, the first step of our approach is to decompose the target sequence into the sum of several terms including a linear term, a quadratic term and higher order terms.
> Though the higher order terms can be eventually controlled, our contribution in proof is not here.
>
> To show that the target sequence approximates jump diffusion, we need to verify that the rescaled sequence weakly converges to the solution of a martingale problem which admits a specific discontinuity structure (i.e., the jump process) and has not been analyzed in previous similar works.
> A necessary step is to verify the tightness of rescaled sequences.
> Previous works (such as ref[9, 10, 35]) all obtain it by verifying properties similar to equidegree continuity (see Chapt 7 in ref[4]).
> Note that the discontinuity of $\bar{v}^{(n)}_t$ make all the previous attempts fail.
> Therefore, we devise a more general criterion and give an easy-to-follow proof procedure for loopless type algorithms (see Property 7 and proofs of Lemma 9, 10 for details).
> We believe that the proposed theoretical techniques are novel and might inspire the development and research of other algorithms.
> We highlight this point in the Proof idea of Theorem 3.4.
>
>
> > *Is there any intuition why we should expect a constant drift for both the frequent and occasionally projection regimes?*
>
> This is a good question and we present a more intuitive explanation here.
> Suppose the objective function has a global optimal point $x_*$ in the unconstrained case, then when no projection operation is performed, the sequence $x_n$ obtained by gradient descent of course would go towards to $x_*$. However, $x_*$ is typically not $x^*$, the optimal point under a linear constraint.
> This causes a bias (or a drift) in the optimization direction, which we correct by projection.
> During the excuting of LPSA, $x_n$ will converge to $x^*$.
> Therefore, the bias in each update's gradients will also converge to a specific direction, which is in fact the direction of $\nabla f(x^*)$ as shown in (10).
> This is the source of what you call "constant drift".  As the frequency of the projection decreases, the accumulation of drift grows until its impact exceeds that of fluctuation and becomes the dominant factor for the convergence efficiency.

---

> > ### Author Response · Authors · 2022-08-01
> > **Thanks for your feedback and constructive advice 2/2**
> >
> >
> > > *The analysis relies heavily on strong convexity and does not seem to easily generalize to the non-strongly convex case; the latter, however, is of higher interest.*
> >
> > First, we would like to say that the strong convexity assumption is common in works on the asymptotic performance of stochastic algorithms (see ref[10, 27], [Chen et.al.] and Section 4 of [Polyak - Juditsky]).
> > As a work that focuses on the asymptotic and non-asymptotic nature of loopless projection methods, we adopt the strong convexity assumption in order to focus the reader's attention on the implications of loopless mechanism and the associated analysis of jump diffusion approximation.
> >
> > On the other hand, we are able to relax the global strong convexity assumption to an assumption that requires global convexity and local strong convexity around the optimal point.
> > We denote the relaxed assumption by $\mathcal{A}^*$ and state it formally as following
> >
> > $
> > \mathcal{A}^*:~~ \langle\nabla f(x) - \nabla f(x^*), ~x-x^*\rangle \ge 0 ~~\forall x\in\mathbb{R}^d; \text{ and }
> > $
> >
> > $
> > \exists M>0,~~\forall x\in \mathcal{B}(x^*,M)
> > $
> >
> > $\langle\nabla f(x) - \nabla f(x^*), ~x-x^*\rangle \ge \mu\|x-x^ * \|^2$
> >
> > We believe that under the assumption $\mathcal{A}^*$ and given $\Gamma(x^*):= \{u_n \to x^*\}$,
> > the results in Thm 3.1 and 3.2 still hold.
> > In the following, we provide a high-level idea of a potential proof under relaxed assumptions, which follows the main spirit of the proof of Thm 1 in ref[35].
> > Define
> >
> > $\Gamma_N(x^*):= \Gamma(x^*)\cap \{ \sup_{n\ge N}\|u_n-x^*\| < M \}$.
> >
> > Then we have $\lim_{N\to\infty}\Gamma_N(x^*) = \Gamma(x^*)$.
> > To prove the weak convergence of $\bar{u}_t^{(n)}$ and $\bar{v}_t^{(n)}$ given $\Gamma(x^*)$, it is sufficient to prove it given $\Gamma_N(x^*)$ for all $N$.
> >
> > For a given $N$, consider a surrogate sequence
> > $x^\prime_n$ defined as
> > $x^\prime_N = x_N \mathrm{1}_{ \ { \| \mathcal{P} _ { \mathbf{A} ^ \bot } x_N - x ^ * \| <M \ } }$
> > and
> >
> > $
> > x ^ \prime _ { n + 1/2 } = x _ n ^ \prime - \eta _ n \left ( \nabla f( x ^ \prime _ n ) \mathrm{1} _ { \{ \| \mathcal{P} _ { \mathbf{A} ^ \bot } x _ n - x ^ * \| < M \} } + K (x ^ \prime _ n - x ^ * ) \mathrm{1} _ { \{ \| \mathcal{P} _ { \mathbf {A} ^ \bot} x _ n - x ^ * \| \ge M \} } \right ) + \eta _ n \xi _ n \mathrm{1} _ { \{ \| \mathcal{P} _ { \mathbf {A} ^ \bot } x _ n - x ^ * \| < M \} } , n \ge N.
> > $
> >
> > Where $K$ is a positive number to be determined. At the same time, $x^\prime_{n+1}$ is assigned $ x ^ \prime _ {n + 1/2}$ or $ \mathcal{P} _ { \mathbf {A} ^ \bot } x ^ \prime _ { n + 1/2 }$ with the same randomness as $x_{n+1}$. Define $ u ^ \prime _ n = \mathcal {P} _ { \mathbf {A} ^ \bot } x ^ \prime _ { n } $ and $v ^ \prime _ n = \mathcal {P} _ { \mathbf {A} } x ^ \prime _ { n }$ . Then $ (u ^ \prime _ n , v ^ \prime _ n ) = ( u _ n , v _ n )$ given $ \Gamma _ N ( x ^ * )$. We can prove the counterpart results of Lemma 4, 5 on $ u ^ \prime _ n , ~ v ^ \prime _ n $. This is owing to that for $ v ^ \prime _ n$, the proof of Lemma 5 does not essentially require the assumption of strong convexity, and for $ u ^ \prime _ n $, its update rule is essentially the same as that investigated in ref[35].
> > And in the subsequent approximation sessions, we do not use any of the other bonuses arising from the strong convexity assumption other than Lemmas 4 and 5 and the positive definite property of $ \nabla ^ 2 f ( x ^ * ) $.
> >
> > However, for the sake of $\Gamma_N(x^*)\in \mathcal{F}_\infty$, the martingale difference property of $\xi_n$ would be disrupted given $\Gamma_N(x^*)$, which prevents us from using original tools to verify the tightness of the target sequence. Pelletier spent a considerable amount of space in [35] dealing with this issue, and finally proposed a universal proof procedure.
> > These techniques go beyond the core of what we want to discuss in this article, so we will leave them as a future work.
> >
> > Finally, we would like to say that we are sorry that we are not familiar with [Kushner-Yin] listed by the reviewer.
> > After a quick reading, we still don't know whether our above weaknesses meet your expectations.
> > If there is a difference, we would welcome the reviewer to point out where the book mentions diffusion approximation results without the strong convexity assumption.
> > In our proof, the analytical influences of the strong convexity assumption and the loopless projection are independent of each other.
> > Therefore, we think it's promising to generalize the results under the original non-strongly convex setting to ones with loopless projection.
> >
> > [Chen et.al.] Statistical Inference for Model Parameters in Stochastic Gradient Descent
> >
> > [Polyak - Juditsky] Acceleration of Stochastic Approximation by Averaging
> >
> > [Kushner - Yin] Stochastic Approximation and Recursive Algorithms and Applications

---

### Official Review · Reviewer_RZGN · 2022-07-11

**Rating:** 7
**Confidence:** 2
**Soundness:** 3 good
**Presentation:** 3 good
**Contribution:** 3 good

**Summary:**

The paper considers the problem of stochastic convex optimization with linear constraints. They proposed the so-called loopless projected stochastic approximation (LPSA) that randomly performs or skip the project step at each iteration. The authors provide the non-asymptotic convergence rate of the resulting algorithms. They also study the asymptotic behaviors of the algorithms under different conditions and parameters of the algorithm. Moreover, their asymptotic analysis provides insights of the effect of projection frequency on the convergence rates.


**Questions:**

The paper proposed a loopless stochastic approximation algorithm. The main results are obtained for stochastic convex optimization. For general stochastic approximation, the mean field is not necessary a gradient function. I wonder would the proofs of Theorem 3.3 and Theorem 3.4 be easily adapted to this case?


**Limitations:**

Yes

**Strengths And Weaknesses:**

The paper is well written and technically strong. Analyzing asymptotic convergence rate of stochastic approximation algorithms with projection is generally difficult because of the discontinuous dynamics (which people usually resort to stochastic inclusion for establishing convergence). I did not check the proofs in the appendix, but outlines of proofs seems plausible.
The results they obtained reveal the relationship between convergence rate and the tuning parameter that controls projection frequency, which is useful in practice.


Weakness:
Currently, the experiment focuses on validating convergence rate of the proposed algorithm. Since the paper consider stochastic optimization with projection. It could help readers understand the effect of loopless projection better by providing plots of number of projections or computational complexity for each parameter setting, along with the MSE figure.

---

> ### Author Response · Authors · 2022-08-01
> **Thank you for your appreciation and recognition of our work.**
>
> Thank you for your appreciation and recognition of our work. Here are our responses to your questions.
>
> > *It could help readers understand the effect of ... along with the MSE figure.*
>
> We have added the log-log scale figures of averaged MSEs vs. the number of projections in Figures 15 and 16 in Appendix D.7 in revision (see page 51).  We find that in both figures, when $\beta$ is closer to $0.5$, the lines of convergence rates are steeper. It is worth noting that a larger $\beta$ implies a smaller number of projections. As a result, for $\beta$ larger than $0.5$, the interval between two adjacent projections is pretty large and to get a target number of projections (e.g., $1000$), the number of iterations can be undesirable. To reduce the computational cost, we take a predetermined number of iterations and depict figures of averaged MSEs vs. the number of projections, so the lines corresponding the larger $\beta$ can be shorter than others.
>
> > *The main results are obtained for stochastic convex optimization. For general ... the proofs of Theorem 3.3 and Theorem 3.4 be easily adapted to this case?*
>
> In fact, it is possible to extend current theories to the case of non-gradient mean fields.
> This is because our proof relies only on the Lipschitz continuity and strong monotony of its gradient $\nabla f$.
> More specifically, if the problem we want to solve is $\mathcal{R}(x) = 0,~~ \text{ s.t. }\mathbf{A}^\top x = 0$, then we only need to assume the following assumptions.
>
> 1. $$
>    \exists L > 0, \text{ s.t. } \| \mathcal{R} ( x ) - \mathcal{R} ( y ) \| \le L \| x - y \| ; ~~ \forall x , y \in \mathbb{R} ^ d
>    $$
>
> 2. $$
>    \exists \mu > 0 , \text{ s.t. } \langle \mathcal{R} ( x ) - \mathcal{R} ( y ) , ~ x - y \rangle \ge \mu \| x - y \| ^ 2 ; ~ \forall x , y \in \mathbb{R} ^ d
>    $$
>
> 3. $$
>    \exists \tilde{L} > 0 , \text{ s.t. } \| \nabla \mathcal{R} ( x ) - \nabla \mathcal{R} ( y ) \| _ 2 \le \tilde{L} \| x - y \| ; ~ \forall x , y \in \mathbb{R} ^ d
>    $$
>
> It is easy to check that, our Lemmas 4 and 5 still hold, and so do the approximation results.
> In this way, we relax the assumption that $\mathcal{R}$ should be in the form of function gradients and extend our result to more general non-linear stochastic approximation.

---

### Meta-Review · Area_Chair_d4SE · 2022-08-26

**Recommendation:** Accept
**Confidence:** Certain

**Metareview:**

All the reviewers agreed that the paper is novel, interesting, and the results are mathematically rigorous. Some of the concerns have been aptly addressed by the authors during the rebuttal period. I congratulate the authors for the nice work and recommend an acceptance.

**Award:**

No

---

### Decision · Program_Chairs · 2022-09-14

Accept